# Sequence Modeling with Spectral Mean Flows

**Jinwoo Kim**
KAIST

**Max Beier**
TU Munich

**Petar Bevanda**
TU Munich

**Nayun Kim**
KAIST

**Seunghoon Hong**
KAIST

## Abstract

A key question in sequence modeling with neural networks is how to represent and learn highly nonlinear and probabilistic state dynamics. Operator theory views such dynamics as linear maps on Hilbert spaces containing mean embedding vectors of distributions, offering an appealing but currently overlooked perspective. We propose a new approach to sequence modeling based on an operator-theoretic view of a hidden Markov model (HMM). Instead of materializing stochastic recurrence, we embed the full sequence distribution as a tensor in the product Hilbert space. A generative process is then defined as maximum mean discrepancy (MMD) gradient flow in the space of sequences. To overcome challenges with large tensors and slow sampling convergence, we introduce **spectral mean flows**, a novel tractable algorithm integrating two core concepts. First, we propose a new neural architecture by leveraging spectral decomposition of linear operators to derive a scalable tensor network decomposition of sequence mean embeddings. Second, we extend MMD gradient flows to time-dependent Hilbert spaces and connect them to flow matching via the continuity equation, enabling simulation-free learning and faster sampling. We demonstrate competitive results on a range of time-series modeling datasets.[1]

## 1 Introduction

A fundamental question in sequence modeling with neural networks is how to represent and learn highly nonlinear and probabilistic state dynamics throughout a sequence. A popular method is to model each step with a stochastic nonlinear network [5, 23, 47, 62, 98]. Since this requires serialization over steps at test time, there has been recent interest in linear recurrence [36, 37, 77] that offers more parallelizability. Yet, their underlying theory often only covers deterministic linear dynamics, requiring probabilistic dynamics to be handled in a post hoc manner via autoregression [36, 37] or stepwise variational inference [82, 104, 72]. This leaves room for a new method that computationally uses linear recurrence while natively handling the nonlinear and probabilistic nature of state dynamics.

Operator theory suggests an appealing approach in this context: embedding probability distributions in Hilbert spaces in which nonlinear dynamics become linear [68]. Specifically, consider Markovian dynamics $(Z^n)_{n \in \mathbb{N}}$ on a state space $\mathcal{Z}$. The theory states that, with a proper choice of Hilbert space $\mathbb{G}$, the distribution of each state $Z^n$ can be uniquely embedded as a vector $\mu_{Z^n}$ in $\mathbb{G}$. Furthermore, the state dynamics $\mathbb{P}[Z^{n+1}|Z^n]$ can be represented as a linear operator $U : \mathbb{G} \to \mathbb{G}$ such that:

$$\mu_{Z^{n+1}} = U \mu_{Z^n}. \tag{1.1}$$

This enables the use of standard linear algebraic tools, such as spectral decomposition of linear operators, to implicitly but correctly represent and learn nonlinear probabilistic dynamics. This approach has been extensively explored in kernel algorithms for learning dynamical systems [51, 55]. However, its application in neural sequence modeling has been limited.

In this work, we propose a new approach to neural sequence modeling that leverages this perspective. We begin with the classical notion of a hidden Markov model (HMM), which is a probabilistic model

---

[1] Code is available at `https://github.com/jw9730/spectral-mean-flow`.

39th Conference on Neural Information Processing Systems (NeurIPS 2025).

for sequences $X^{1:N} = (X^1, ..., X^N)$ based on hidden states $Z^{1:N} = (Z^1, ..., Z^N)$ under Markovian dynamics $\mathbb{P}[Z^{n+1}|Z^n]$ and local observation model $\mathbb{P}[X^n|Z^n]$. While HMMs offer a structured and universal way to model sequences, directly modeling their nonlinear stochastic recurrence can be challenging. This motivates the aforementioned operator-theoretic perspective.

Specifically, our first step is to embed the distribution of the entire sequence $X^{1:N}$ as an element in a tensor product Hilbert space. This provides a basic setup for sequence modeling in the operator-theoretic context, allowing us to exploit linearity (1.1). The setup also allows us to define a generative process as a maximum mean discrepancy (MMD) gradient flow [4] in the space of sequences. Then, to overcome practical challenges associated with high-dimensional tensors and slow convergence of standard MMD flows, we introduce **spectral mean flows**, a novel method with two core contributions:

- We propose a new neural architecture that utilizes spectral decomposition of the linear operator underlying an HMM. This leads to a scalable tensor network decomposition of the sequence distribution embedding, making computations tractable.

- We propose an extension of MMD gradient flows to time-dependent Hilbert spaces, which may be of independent interest. This connects to flow matching [60] via the continuity equation, enabling end-to-end, simulation-free learning and faster sampling convergence.

We test spectral mean flows on a range of time-series datasets, demonstrating competitive results.

The remainder of this paper is organized as follows. In Section 2, we introduce necessary background. Section 3 introduces spectral mean flows. Section 4 discusses related work, and Section 5 presents experiments on synthetic data and time-series modeling datasets. Section 6 draws conclusions. All theoretical arguments are formally stated and proven in the appendix.

## 2 Background

In this section, we introduce key mathematical tools that enable us to work with probability distributions using linear algebraic methods. More details can be found in Appendix A.

**Reproducing kernel Hilbert spaces (RKHSs)** offer a framework for treating functions as vectors in Hilbert spaces [83]. Consider a domain $\mathcal{X} \subset \mathbb{R}^d$, and let $\mathbb{H}$ be an RKHS induced by a kernel $k : \mathcal{X} \times \mathcal{X} \to \mathbb{R}$. The space $\mathbb{H}$ contains functions $f : \mathcal{X} \to \mathbb{R}$ and is equipped with an inner product $\langle \cdot, \cdot \rangle_{\mathbb{H}}$ and a corresponding norm $\| \cdot \|_{\mathbb{H}}$. Its canonical feature map $\phi : \mathcal{X} \to \mathbb{H}$ is defined by $\phi(\mathbf{x}) := k(\cdot, \mathbf{x})$, and satisfies the reproducing property: $f(\mathbf{x}) = \langle \phi(\mathbf{x}), f \rangle_{\mathbb{H}}$ for any $f \in \mathbb{H}$.

To handle dependencies between variables, we also consider a second domain $\mathcal{Z} \subset \mathbb{R}^m$ with its own RKHS $\mathbb{G}$ induced by a kernel $l : \mathcal{Z} \times \mathcal{Z} \to \mathbb{R}$ with canonical feature map $\varphi : \mathcal{Z} \to \mathbb{G}$.

**Mean embeddings** provide a way to represent probability distributions as vectors in RKHS [70]. For a random variable $X$ with distribution $\pi$ on $\mathcal{X}$, the mean embedding is defined as:

$$\mu_\pi := \mathbb{E}[\phi(X)] \in \mathbb{H}. \tag{2.1}$$

We write $\mu_X = \mu_\pi$ when the distribution is clear from context. For a characteristic RKHS [91], the map $\pi \mapsto \mu_\pi$ is injective, meaning each distribution has a unique embedding. This property underlies the maximum mean discrepancy (MMD) distance of distributions, $\mathrm{MMD}(\nu, \pi) = \|\mu_\nu - \mu_\pi\|_{\mathbb{H}}$ [34].

**Conditional mean embeddings** extend the above to conditional distributions [70]. Consider a random variable $Z$ on $\mathcal{Z}$, defining $\mathbb{P}[X|Z]$. Under standard assumptions [68, 74], the mean embedding of $X$ given $Z = \mathbf{z}$ can be expressed with the conditional mean embedding (CME) operator $U_{X|Z} : \mathbb{G} \to \mathbb{H}$:

$$\mu_{X|Z=\mathbf{z}} := \mathbb{E}[\phi(X) \mid Z = \mathbf{z}] = U_{X|Z}\, \varphi(\mathbf{z}) \in \mathbb{H}. \tag{2.2}$$

Notably, $U_{X|Z}$ is a linear operator. The linearity allows the use of spectral methods, and also yields useful properties such as $\mu_X = U_{X|Z}\, \mu_Z$ (1.1) via the law of total expectation.

**MMD gradient flows** enable sampling from distributions specified by mean embeddings [4]. Given a target distribution $\pi$ with embedding $\mu_\pi$, the flow is driven by a time-dependent vector field $(v_t)_{t \geq 0}$:

$$v_t(\mathbf{x}) = -\nabla_{\mathbf{x}}(\mu_{p_t} - \mu_\pi)(\mathbf{x}) = -\nabla_{\mathbf{x}}\langle \phi(\mathbf{x}), \mu_{p_t} - \mu_\pi \rangle_{\mathbb{H}}, \tag{2.3}$$

where $(p_t)_{t \geq 0}$ is a probability path induced by the continuity equation $\partial_t p_t + \mathrm{div}(p_t v_t) = 0$. Starting at any initial distribution $p_0$, the path converges towards the target distribution: $p_t \to \pi$ as $t \to \infty$. In practice, one can generate samples $\mathbf{x} \sim \pi$ by starting at an initial sample $\mathbf{x}_0 \sim p_0$ and numerically integrating the ordinary differential equation (ODE) $d\mathbf{x}_t = v_t(\mathbf{x}_t)\, dt$ to obtain $\mathbf{x}_t \sim p_t$.

## 3 Spectral Mean Flows

**Problem setup**  We formulate sequence modeling in an operator-theoretic context using hidden Markov models (HMM). We consider a sequence distribution $X^{1:N} \sim \rho$, where $X^n \in \mathcal{X}$, and model it as an HMM with hidden states $Z^{1:N}$ where $Z^n \in \mathcal{Z}$. The hidden states evolve under time-invariant Markovian dynamics $\mathbb{P}[Z^{n+1}|Z^n]$, and determine $X^{1:N}$ via a local observation model $\mathbb{P}[X^n|Z^n]$. The HMM can express any sequence distribution if the hidden Markov chain is sufficiently expressive.

Our approach begins by embedding the entire sequence distribution $X^{1:N} \sim \rho$ in an appropriate Hilbert space, such that an MMD flow can be defined on the space of sequences $\mathcal{X}^N$. We leverage the fact that tensor product $\mathbb{H}^{\otimes N} := \mathbb{H} \otimes \cdots \otimes \mathbb{H}$ of a characteristic RKHS $\mathbb{H}$ remains characteristic under mild assumptions (Lemma B.1). The canonical feature map of this tensor product space is $\mathbf{x}^{1:N} \mapsto \phi(\mathbf{x}^1) \otimes \cdots \otimes \phi(\mathbf{x}^N)$, which naturally leads to the sequence mean embedding $\mu_\rho = \mu_{X^{1:N}}$:

$$\mu_\rho := \mathbb{E}[\phi(X^1) \otimes \cdots \otimes \phi(X^N)] \in \mathbb{H}^{\otimes N}. \tag{3.1}$$

With the above concepts in hand, we can extend the MMD flow (2.3) to operate on entire sequences. Specifically, we define a time-dependent vector field $(v_t)_{t\geq 0}$ that induces a probability path $(p_t)_{t\geq 0}$ of sequences $X_t^{1:N} \sim p_t$, converging towards the target distribution $X^{1:N} \sim \rho$ over time $t$:[2]

$$v_t(\mathbf{x}^{1:N}) = -\nabla_{\mathbf{x}^{1:N}} \langle \phi(\mathbf{x}^1) \otimes \cdots \otimes \phi(\mathbf{x}^N), \mu_{p_t} - \mu_\rho \rangle_{\mathbb{H}^{\otimes N}}. \tag{3.2}$$

Samples $\mathbf{x}_t^{1:N} \sim p_t$ can then be generated by solving the ODE $d\mathbf{x}_t^{1:N} = v_t(\mathbf{x}_t^{1:N})\, dt$ from $\mathbf{x}_0^{1:N} \sim p_0$.

While theoretically sound, the idea in its naïve form faces practical challenges. The mean embedding $\mu_\rho$ (3.1) is a higher-order tensor whose size grows exponentially with the sequence length $N$, making training and sampling intractable. In addition, sampling with MMD flow is typically slow, because its convergence $p_t \to \rho$ is only guaranteed asymptotically as $t \to \infty$ [4].

To overcome the challenges, we propose **spectral mean flows**, a novel approach that is tractable, easy to train, and converges fast at sampling. In Section 3.1, we derive a scalable tensor network decomposition of mean embeddings (3.1) using spectral decomposition of linear operators underlying the HMM. In Section 3.2 we present an extension of MMD flow (3.2) to time-dependent RKHS that connects to flow matching and converges within a finite time $t = 1$. Section 3.3 integrates these ideas into a neural sequence architecture and learning algorithm, and discusses implementation.

### 3.1 Tractability with Spectral Decomposition

To make the MMD flows (3.2) with sequence mean embeddings (3.1) computationally tractable, we exploit the linear operator structure underlying the HMM by applying spectral decomposition.

**Linear operators underlying HMMs**  For the conditional distributions $\mathbb{P}[Z^{n+1}|Z^n]$ and $\mathbb{P}[X^n|Z^n]$ underlying our HMM for sequences $X^{1:N}$, we can define the respective transition and observation CME operators $U : \mathbb{G} \to \mathbb{G}$ and $O : \mathbb{G} \to \mathbb{H}$ as follows (see Section 2 and Appendix A.3):

$$U\varphi(\mathbf{z}) = \mathbb{E}[\varphi(Z^{n+1}) \mid Z^n = \mathbf{z}] \in \mathbb{G}, \tag{3.3}$$
$$O\varphi(\mathbf{z}) = \mathbb{E}[\phi(X^n) \mid Z^n = \mathbf{z}] \in \mathbb{H}. \tag{3.4}$$

For the MMD flow generative process $(X_t^{1:N})_{t\geq 0}$ (3.2), we make a modeling choice that the hidden dynamics $\mathbb{P}[Z^{n+1}|Z^n]$ remain fixed over time $t$, and only the observation model $\mathbb{P}[X_t^n|Z^n]$ evolves over $t$. Then we obtain a time-dependent observation CME operator $(S_t)_{t\geq 0}$ where $S_t : \mathbb{G} \to \mathbb{H}$:

$$S_t\varphi(\mathbf{z}) = \mathbb{E}[\phi(X_t^n) \mid Z^n = \mathbf{z}] \in \mathbb{H}. \tag{3.5}$$

**Mean embedding decomposition**  To make MMD flow (3.2) tractable, our key idea is to decompose mean embeddings $\mu_\rho$ and $\mu_{p_t}$ into tractable forms by leveraging linearity of CME operators $U, O, S_t$.

As a first step, we decompose the mean embeddings $\mu_\rho$ and $\mu_{p_t}$ into their respective observation operators $O$ and $S_t$, along with the mean embedding $\mu_{Z^{1:N}}$ of the hidden Markov chain defined as:

$$\mu_{Z^{1:N}} := \mathbb{E}[\varphi(Z^1) \otimes \cdots \otimes \varphi(Z^N)] \in \mathbb{G}^{\otimes N}. \tag{3.6}$$

---

[2]We have two different notions of time: $n = 1...N$ over sequence length, and $t \geq 0$ over generative process.

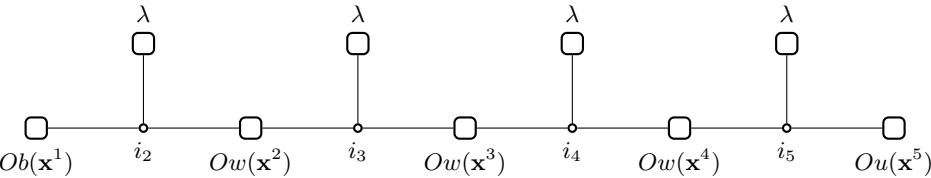

Figure 1: A tensor diagram [11, 2] of the inner product $\langle \phi(\mathbf{x}^1) \otimes \cdots \otimes \phi(\mathbf{x}^N), \mu_\rho \rangle$ (3.10) for $N = 5$, showing how its evaluation involving a tensor $\mu_\rho$ decomposes into matrix and vector multiplications.

Defining the $N$-fold tensor product operator by $O^{\otimes N} := O \otimes \cdots \otimes O : \mathbb{G}^{\otimes N} \to \mathbb{H}^{\otimes N}$, we obtain:

$$\mu_\rho := \mathbb{E}[\phi(X^1) \otimes \cdots \otimes \phi(X^N)] \qquad \text{(mean embedding (3.1))}$$
$$= \mathbb{E}\left[ \mathbb{E}[\phi(X^1) \mid Z^1] \otimes \cdots \otimes \mathbb{E}[\phi(X^N) \mid Z^N] \right] \qquad \text{(HMM structure)}$$
$$= \mathbb{E}[(O\varphi(Z^1)) \otimes \cdots \otimes (O\varphi(Z^N))] \qquad \text{(CME operator (3.4))}$$
$$= O^{\otimes N} \mathbb{E}[\varphi(Z^1) \otimes \cdots \otimes \varphi(Z^N)] \qquad \text{(linearity)}$$
$$= O^{\otimes N} \mu_{Z^{1:N}}. \qquad (3.7)$$

Similarly, we have $\mu_{p_t} = S_t^{\otimes N} \mu_{Z^{1:N}}$ with $S_t^{\otimes N} : \mathbb{G}^{\otimes N} \to \mathbb{H}^{\otimes N}$. Formal proof is in Proposition B.2.

The remaining challenge is decomposing the hidden chain embedding $\mu_{Z^{1:N}}$. We use the fact that the linearity of the transition operator $U : \mathbb{G} \to \mathbb{G}$ enables a decomposition of the following form:

$$U = \sum_{i \in \mathbb{N}} \lambda_i h_i \otimes g_i. \qquad (3.8)$$

In practice, we use rank-$r$ decomposition by restricting to $i \in [r]$, where $r$ controls expressiveness.

A possible choice of (3.8) is singular value decomposition (SVD) with singular values $\lambda_i \in \mathbb{R}$ and left/right singular functions $g_i, h_i : \mathcal{Z} \to \mathbb{R}$. Or, assuming $U$ is normal with a discrete spectrum, we can use eigenvalue decomposition (EVD) with eigenvalues $\lambda_i \in \mathbb{C}$ and eigenfunctions $g_i, h_i : \mathcal{Z} \to \mathbb{C}$. While both are valid, we use EVD that efficiently encodes long-range interactions via oscillations [12].

Then, to decompose $\mu_{Z^{1:N}} \in \mathbb{G}^{\otimes N}$, we draw inspiration from a result in Koopman operator theory that eigenfunctions are closed under pointwise multiplication, i.e., they form a multiplicative algebra, with products yielding new eigenfunctions whose eigenvalues multiply [66, 12, 10]. As we show, encoding this closure in finite-rank representations makes instantiating $\mathbb{G}^{\otimes N}$ redundant, as nonlinear or higher-order interactions are already encoded in the algebra generated by the span of eigenfunctions.

Specifically, we make a technical assumption that the RKHS $\mathbb{G}$ is closed under pointwise multiplication, $f, g \in \mathbb{G} \implies f \cdot g \in \mathbb{G}$. This holds for Sobolev spaces $H^s(\mathbb{R}^d)$ of order $s > d/2$ [1], which are characteristic and thus suitable for our framework. Under this assumption and leveraging a decomposition of $U$ (3.8) with $g_i, h_i \in \mathbb{G}$, we derive the following novel decomposition of $\mu_{Z^{1:N}}$:

$$\mu_{Z^{1:N}} = \sum_{i_2, \ldots, i_N} b_{i_2} \otimes \lambda_{i_2} w_{i_2, i_3} \otimes \cdots \otimes \lambda_{i_{N-1}} w_{i_{N-1}, i_N} \otimes \lambda_{i_N} h_{i_N}, \qquad (3.9)$$

where $b_i, w_{i,j}$ are fixed elements in $\mathbb{G}$ determined by $g_i, h_i$ and the initial state distribution $\mu_{Z^1}$. Complete details and proof are given in Proposition B.6.

**Tractability** With the decompositions in (3.7) and (3.9), we can convert the intractable computations in MMD flow (3.2) into a tractable form. Specifically, in (3.2), the inner product between feature of sequence data $\mathbf{x}^{1:N}$ and mean embedding $\mu_\rho$ dramatically simplify (Corollary B.7):

$$\langle \phi(\mathbf{x}^1) \otimes \cdots \otimes \phi(\mathbf{x}^N), \mu_\rho \rangle = \langle \phi(\mathbf{x}^1) \otimes \cdots \otimes \phi(\mathbf{x}^N), O^{\otimes N} \mu_{Z^{1:N}} \rangle$$
$$= \sum_{i_2, \ldots, i_N} Ob_{i_2}(\mathbf{x}^1) \cdot \lambda_{i_2} \cdot Ow_{i_2, i_3}(\mathbf{x}^2) \cdots \lambda_{i_N} \cdot Oh_{i_N}(\mathbf{x}^N), \quad (3.10)$$

where we use the shorthand $\langle \cdot, \cdot \rangle = \langle \cdot, \cdot \rangle_{\mathbb{H}^{\otimes N}}$. For $\mu_{p_t}$, it is only required to substitute $O$ with $S_t$.

Importantly, the reformulation in (3.10) eliminates the requirement to materialize exponentially large tensors in $\mathbb{H}^{\otimes N}$ when computing (3.2). Instead, the computation reduces to a standard tensor network contraction (illustrated in Figure 1), achieving linear complexity in sequence length $N$ and polynomial complexity in rank $r$. This makes the approach tractable for realistic sequence lengths.

## 3.2 Faster Convergence with Time-dependent RKHS and Flow Matching

Having established tractable mean embeddings, we now address the remaining challenge of slow sampling convergence in standard MMD flows.

**Time-dependent RKHS**   Recall MMD gradient flow $(p_t)_{t \geq 0}$ for a target distribution $\pi$ on $\mathcal{X}$ within an RKHS $\mathbb{H}$ (Section 2). The flow follows the steepest direction of the MMD between the current distribution $p_t$ and target $\pi$ measured in $\mathbb{H}$ [4]. Since the RKHS $\mathbb{H}$ determines the geometric structure underlying the gradient flow via MMD, convergence behavior is essentially tied to the fixed $\mathbb{H}$.

To improve the convergence of MMD flow, a natural way is to use time-dependent RKHS $\mathbb{H}_t$ [30]. This creates a time-evolving geometry where the steepest direction of the MMD changes over time, allowing more flexibility of the flow. Let $(\mathbb{H}_t)_{t \geq 0}$ be a time-dependent family of characteristic RKHS, with canonical feature map $(\phi_t)_{t \geq 0}$ where $\phi_t : \mathcal{X} \to \mathbb{H}_t$. The mean embedding of a distribution $\pi$ is:

$$\mu_{\pi,t} := \mathbb{E}[\phi_t(X)] \in \mathbb{H}_t. \tag{3.11}$$

The MMD distance in $\mathbb{H}_t$ is accordingly given as $\mathrm{MMD}_t(\nu, \pi) = \|\mu_{\nu,t} - \mu_{\pi,t}\|_{\mathbb{H}_t}$.

Then, MMD flow with time-dependent RKHS $\mathbb{H}_t$ can be obtained from standard construction [27]. The flow defines a probability path $(p_t)_{t \geq 0}$ driven by a vector field $(v_t)_{t \geq 0}$ via the continuity equation $\partial_t p_t + \mathrm{div}(p_t v_t) = 0$ that, at each time $t$, follows the steepest direction of the MMD measured in $\mathbb{H}_t$:

$$v_t(\mathbf{x}) = -\nabla_{\mathbf{x}} \langle \phi_t(\mathbf{x}), \mu_{p_t,t} - \mu_{\pi,t} \rangle_{\mathbb{H}_t}, \tag{3.12}$$

and samples $\mathbf{x}_t \sim p_t$ can be obtained via the ODE $d\mathbf{x}_t = v_t(\mathbf{x}_t)\,dt$. More details are in Appendix B.3.

**Connection to flow matching**   With time-dependent RKHS, [30] used discriminator learning of $\phi_t$ to empirically improve the convergence of MMD flow. Instead, by connecting to flow matching [60], we propose a way to achieve convergence at finite time $t = 1$ rather than asymptotically as $t \to \infty$.

We hinge on the fact that we can often find an analytic probability path $q_t$ and a vector field $u_t$ defined in $t \in [0,1]$ that satisfy the continuity equation $\partial_t q_t + \mathrm{div}(q_t u_t) = 0$ and also $q_1 \approx \pi$.[3] This is the idea of flow matching (FM) [60], which allows us to borrow established pairs of $(q_t, u_t)$.

That is, for a choice of $(q_t, u_t)$, we aim to identify the components of the MMD flow (3.12) such that its vector field $v_t$ regresses $u_t$. If we have $v_t = u_t$ and $p_0 = q_0$, it follows from the continuity equation that $p_t = q_t$ and the MMD flow converges closely to the target $p_1 = q_1 \approx \pi$ at $t = 1$.

Here, an important caveat is that our vector field $v_t$ is always a gradient field with zero curl. For the target equality $v_t = u_t$ to be well-specified, the target $u_t$ must also be a gradient field. While this is not always true, we fortunately find that many commonly used vector fields in FM are gradient fields. This includes the optimal transport (OT) field and the diffusion-types, as we prove in Appendix B.4.

**Sequence instantiation**   We now apply the aforementioned ideas to HMM for sequences $X^{1:N} \sim \rho$ based on hidden states $Z^{1:N}$. We consider time-dependent RKHS $(\mathbb{H}_t)_{t \in [0,1]}$. By extending (3.12), the vector field $v_t$ of the corresponding MMD flow $X_t^{1:N} \sim p_t$ can be written as follows:

$$v_t(\mathbf{x}^{1:N}) = -\nabla_{\mathbf{x}^{1:N}} \langle \phi_t(\mathbf{x}^1) \otimes \cdots \otimes \phi_t(\mathbf{x}^N), \mu_{p_t,t} - \mu_{\rho,t} \rangle_{\mathbb{H}_t^{\otimes N}}. \tag{3.13}$$

Then, assuming that the transition operator $U : \mathbb{G} \to \mathbb{G}$ for $\mathbb{P}[Z^{n+1}|Z^n]$ does not depend on time $t$, we extend the observation operators $O_t, S_t : \mathbb{G} \to \mathbb{H}_t$ for $\mathbb{P}[X^n|Z^n]$ and $\mathbb{P}[X_t^n|Z^n]$, respectively, as:

$$O_t \varphi(\mathbf{z}) = \mathbb{E}[\phi_t(X^n) \mid Z^n = \mathbf{z}] \in \mathbb{H}_t, \tag{3.14}$$

$$S_t \varphi(\mathbf{z}) = \mathbb{E}[\phi_t(X_t^n) \mid Z^n = \mathbf{z}] \in \mathbb{H}_t. \tag{3.15}$$

---

[3] For example, $q_1$ is a smoothing of $\pi$ with a Gaussian filter with a sufficiently small standard deviation.

With this, by extending the decomposition in (3.7), we obtain:

$$\mu_{\rho,t} = O_t^{\otimes N} \mu_{Z^{1:N}}, \quad \mu_{p_t,t} = S_t^{\otimes N} \mu_{Z^{1:N}}. \tag{3.16}$$

And by extending the decomposition in (3.10), we obtain:

$$\langle \phi_t(\mathbf{x}^1) \otimes \cdots \otimes \phi_t(\mathbf{x}^N), \mu_{\rho,t} \rangle = \sum_{i_2 \ldots i_N} O_t b_{i_2}(\mathbf{x}^1) \cdot \lambda_{i_2} O_t w_{i_2,i_3}(\mathbf{x}^2) \cdots \lambda_{i_N} O_t h_{i_N}(\mathbf{x}^N) \tag{3.17}$$

where we use the shorthand $\langle \cdot, \cdot \rangle = \langle \cdot, \cdot \rangle_{\mathbb{H}_t^{\otimes N}}$. For $\mu_{p_t,t}$, it is only required to substitute $O_t$ by $S_t$. This retains the tensor network structure, and hence the tractability of the MMD flow as in Section 3.1.

## 3.3 Neural Parameterization and Learning

Having established spectral mean flows in RKHSs, we now develop a neural sequence architecture and learning algorithm. In (3.13) and (3.17), we can see that the operators $O_t, S_t : \mathbb{G} \to \mathbb{H}_t$, together with the (Sobolev) RKHS elements $b_i, w_{ij}, h_i \in \mathbb{G}$ and $\lambda_i$ for $i, j \in [r]$, determine the MMD flow $v_t$. We describe how to parameterize and learn these components as neural networks to pursue $v_t = u_t$.

**Neural parameterization**   To develop a neural parameterization, we adopt neural tangent kernel (NTK) theory [42] as a theoretical footing that connects neural networks and Sobolev spaces, e.g., $\mathbb{G}$. Specifically, the function space defined by a multi-layer perceptron (MLP) in the infinite-width limit is norm-equivalent to the RKHS of a Sobolev kernel [8]. In addition, any function in a Sobolev space can be approximated to arbitrary precision with a finite-width MLP, where the theoretical scaling law predicts decreasing approximation error as the network size increases [86, Theorems 1 and 3].

These results justify, e.g., parameterizing each $b_i, w_{ij}, h_i \in \mathbb{G}$ as a scalar-valued MLP. Yet, instead of separating $O_t, S_t : \mathbb{G} \to \mathbb{H}_t$, we use an end-to-end parameterization [39] of each $O_t b_i, S_t b_i, \ldots \in \mathbb{H}_t$ as a $t$-conditioned scalar-valued MLP on $\mathcal{X}$. While this $t$-conditioning is inspired by [30], a difference is that we directly parameterize elements of $\mathbb{H}_t$ without using linear kernel approximation.

**Learning and sampling**   Under the aforementioned neural parameterization, the MMD flow (3.13) becomes a neural gradient field $v_t(\cdot; \theta)$ with learnable components $\theta := (O_t b_i, S_t b_i, O_t w_{ij}, \ldots, \lambda_i)$.

Then, using a known pair $(q_t, u_t)_{t \in [0,1]}$ of probability path $q_t$ and vector field $u_t$ satisfying $q_1 \approx \rho$, we can use flow matching (FM) objective [60] to achieve the equality $v_t(\cdot; \theta) = u_t(\cdot)$ (Section 3.2):

$$\theta^* = \arg\min_\theta \mathbb{E}_{t \sim \mathcal{U}[0,1], q_t(\mathbf{x}^{1:N})} \left[ \|v_t(\mathbf{x}^{1:N}; \theta) - u_t(\mathbf{x}^{1:N})\|_2^2 \right], \tag{3.18}$$

While (3.18) is intractable, thanks to compatibility with FM (Section 3.2), we can use an equivalent tractable objective named conditional FM [60] using conditional counterparts $q_t(\cdot | \mathbf{x}_1^{1:N}), u_t(\cdot | \mathbf{x}_1^{1:N})$:

$$\theta^* = \arg\min_\theta \mathbb{E}_{t \sim \mathcal{U}[0,1], \rho(\mathbf{x}_1^{1:N}), q_t(\mathbf{x}^{1:N} | \mathbf{x}_1^{1:N})} \left[ \|v_t(\mathbf{x}^{1:N}; \theta) - u_t(\mathbf{x}^{1:N} | \mathbf{x}_1^{1:N})\|_2^2 \right]. \tag{3.19}$$

This leaves us with a tractable learning algorithm for the neural MMD flow $v_t(\cdot; \theta)$. The algorithm is simulation-free, i.e., no ODE solving with $v_t(\cdot; \theta)$ is necessary during training, and uses double backpropagation, i.e., $v_t(\cdot; \theta)$ internally invokes backpropagation during forward pass. In our experiments, we use the OT flow path and field $(q_t, u_t)$ for training (Proposition B.11) due to their simplicity.

After training, we can sample $\mathbf{x}_1^{1:N} \sim q_1 \approx \rho$ via the ODE $d\mathbf{x}_t^{1:N} = v_t(\mathbf{x}_t^{1:N}; \theta^*), \mathbf{x}_0^{1:N} \sim q_0$.

**Implementation**   We discuss implementation of spectral mean flow used in our experiments. More details are in Appendix C.1. First of all, we use rank-$r$ EVD as discussed in Section 3.1, requiring complex eigenvalues $\lambda_i \in \mathbb{C}$ and complex-valued MLPs that parameterize $O_t b_i, S_t b_i, \ldots : \mathcal{X} \to \mathbb{C}$.

To parameterize $O_t b_i, O_t w_{ij}, O_t h_i$ for $i, j \in [r]$, we employ a shared feature extractor $\mathrm{MLP}_O(\cdot, t) : \mathcal{X} \to \mathbb{C}^{d_f}$ with feature dimension $d_f$ and combine it with readout heads $\mathbf{B}, \mathbf{L}, \mathbf{R}, \mathbf{H} \in \mathbb{C}^{d_f \times r}$:

$$O_t b_i(\mathbf{x}) \approx [\mathbf{hB}]_i \quad O_t w_{ij}(\mathbf{x}) \approx [\mathbf{L}\mathrm{diag}(\mathbf{h})\mathbf{R}]_{ij} \quad O_t h_i(\mathbf{x}) \approx [\mathbf{hH}]_i \quad \mathbf{h} := \mathrm{MLP}_O(\mathbf{x}, t). \tag{3.20}$$

We note that similar approaches are found in operator-theoretic learning of dynamical systems [57]. To parameterize $S_t b_i, S_t w_{ij}, S_t h_i$, we only change the feature extractor, denoted as $\mathrm{MLP}_S(\cdot, t) : \mathcal{X} \to \mathbb{C}^{d_f}$. We design the two MLPs to share most of their parameters through a switching scheme:

$$\mathrm{MLP}_O(\cdot, t) := \mathrm{MLP}(\cdot, \mathbf{o}, t), \quad \mathrm{MLP}_S(\cdot, t) := \mathrm{MLP}(\cdot, \mathbf{s}, t), \tag{3.21}$$

where $\mathrm{MLP}(\cdot, \cdot, t) : \mathcal{X} \times \mathbb{R}^{d_h} \to \mathbb{C}^{d_f}$ is shared and $\mathbf{o}, \mathbf{s} \in \mathbb{R}^{d_h}$ are trainable switching parameters.

The design of readout heads in (3.20) offers a computational benefit. With the linearity of the tensor network (3.17), we can rearrange matrix multiplications to avoid materializing $r \times r$ matrix states $\mathbf{L}\mathrm{diag}(\mathbf{h})\mathbf{R}$. With this, we achieve $\mathcal{O}(Nr + r^2)$ space complexity of (3.17), avoiding the $Nr^2$ term.

To parameterize and initialize the complex-valued parameters $(\lambda, \mathbf{B}, \mathbf{L}, \mathbf{R}, \mathbf{H})$, we take inspiration from the universal spectrum argument of [7, Proposition 3], which proposes to take eigenvalues from the uniform distribution on the complex unit disk. We employ the exponential parameterization of [75, Lemma 3.2] for this, which has an additional benefit of stabilizing the training.

We lastly discuss the design of $\mathrm{MLP}(\cdot, \cdot, t)$. Since MMD flow (3.13) uses its gradient with respect to input, non-differentiable components such as ReLU can cause discontinuities [26]. We design $\mathrm{MLP}(\cdot, \cdot, t)$ to be fully differentiable with squared ReLU activation [87] and root mean square (RMS) normalization [103]. For $t$-conditioning, we use sinusoidal embedding [96]. We find these design choices to work robustly while being simpler over alternatives such as SwiGLU [85] and adaLN [79].

## 4 Related Work

**Operator theory**   Operator theory lifts the idea of vector spaces from points to functions [80]. The framework was initially developed for solving equations of infinitely many variables. This point of view is particularly advantageous in high-dimensional spaces, as maps of functions describe the map of every points simultaneously instead of each individual point. [52, 54] used the theory of linear operators to describe physical evolution equations [24] of probability measures and their dual observable functions. In a modern context, linear operator theory for evolution equations is often used to find practical linear algebra techniques to minimize, typically unsupervised, learning objectives. Following [40], we can categorize them into four goals: revealing underlying structure [20, 28, 51, 55], learning representations [64, 57, 81], modeling data distributions for prediction [43, 68, 7], and lastly, generating points from a distribution over sequential data. To the best of our knowledge, the fourth task of generating new points from a distribution over sequences has not been approached yet.

**Mean embeddings**   Hilbert space embeddings of statistical quantities are a long-term staple in statistical learning [6]. In particular, RKHS mean embeddings have proven to be powerful tools [70]. Regressing the embedding of conditional expectations is studied with conditional mean embeddings [89, 35, 94]. This framework generalizes linear regression in RKHS from the space of vectors to the space of distributions, naturally giving rise to so-called conditional expectation operators that map distributions to their expectations of a function. They appear in the kernel Bayes rule [29] and are used in filtering [65], where a recursive application of conditional expectation operators over sequences first appeared in [88]. Recently, conditional expectation operators were connected to the MMD [68], the transfer operators of Markov models [43, 88, 9, 55], and have been integrated with the linear operator theory for evolution equations [55, 41]. Again, none of the methods studies generation with transfer operators, especially generating an entire coherent sequence, which is of our interest.

**Generating samples from mean embeddings**   A key underlying idea of our work is generating samples from a distribution specified as a mean embedding. A classical approach for this is matching a mean embedding against those of a parametric family of distributions and recovering the minimizing distribution [90]. Another approach is to estimate the solution to an inverse problem from data to obtain operators that directly recover densities from mean embeddings [84]. Our work is more closely related to approaches that evolve an empirical distribution to match the mean embedding, which includes kernel herding [14, 58], and particularly MMD gradient flows [4] that form the basis of our work. A downside is that fast convergence is only guaranteed for a small class of kernels [4, 30, 38], which we improve by connecting to flow matching [60] via the continuity equation.

## 5 Experiments

We demonstrate spectral mean flows on two synthetic setups and generative modeling on a range of time-series datasets. Details of the experiments and supplementary results are in Appendix C.

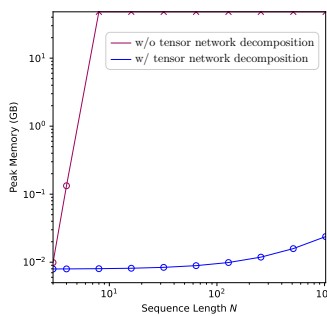

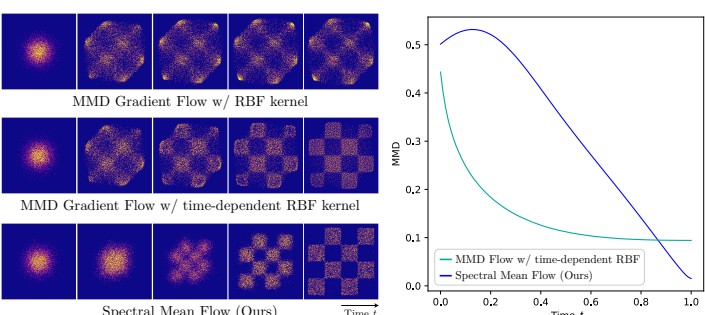

Figure 2: Peak GPU memory of inner product $\langle \mathbf{x}^1 \otimes \cdots \otimes \mathbf{x}^N, \mu \rangle$ depending on the use of tensor network decomposition (3.17).

Figure 3: 2D checkerboard experiment. Left: Intermediate distributions over sampling timesteps (**zoom in** for a better view). Right: MMD between the intermediate and target distributions over sampling timesteps, measured with an RBF kernel of bandwidth 1.

## 5.1 Synthetic Experiments

We first verify our claims on tractability (Section 3.1), focusing on how the tensor network decomposition in equation (3.17) and Figure 1 contributes to tractability and scalability under the hood. We run a numerical experiment of evaluating the inner product $\langle \mathbf{x}^1 \otimes \cdots \otimes \mathbf{x}^N, \mu \rangle$ between a rank-one tensor product $\mathbf{x}^1 \otimes \cdots \otimes \mathbf{x}^N$ of vectors $\mathbf{x}^n \in \mathbb{R}^d$ and a higher-order tensor $\mu \in \mathbb{R}^{d^N}$ for $d = 32$. We measure the peak GPU memory usage across sequence lengths $N$, depending on the availability of tensor network decomposition in the form of (3.17). The results are in Figure 2, showing that without the tensor network decomposition, it is almost impossible to evaluate the inner product for $N > 4$.

Then, we verify our claims on faster convergence than MMD gradient flows (Section 3.2). For ease of visualization, we use a 2D checkerboard dataset with scale $[-4.5, +4.5]$. We compare against MMD flows based on radial basis function (RBF) kernel $k(\mathbf{x}, \mathbf{x}') = \exp(-0.5\|\mathbf{x} - \mathbf{x}'\|^2/\sigma^2)$ with bandwidth $\sigma = 1$, and time-dependent RBF kernel $(k_t)_{t \in [0,1]}$ [30] with $\sigma(t) = (1 - t) + 0.1t$. For MMD flows, we use 10,000 particles and 100 sampling steps with step size 1. For spectral mean flow, we treat each data as a sequence of length $N = 2$ and use 100 sampling steps with a `midpoint` ODE solver. The results are in Figure 3. MMD flows explicitly minimize MMD measured by RBF kernels, and their samples move rapidly towards the target near $t = 0$. However, after some steps they face stagnation, suggesting much longer sampling is required. In contrast, spectral mean flow accurately converges to the target at $t = 1$. This supports our claims in Section 3.2 that spectral mean flow can converge arbitrarily close to the target at fixed time $t = 1$, faster than naïve MMD flow that requires $t \to \infty$. We note that, in the context of generative modeling, convergence at $t = 1$ is sufficient and we need not worry about near $t = 0$ where spectral mean flow slightly increases MMD initially.

## 5.2 Time-Series Modeling

We demonstrate spectral mean flow on unconditional generation of time-series data. The baselines include autoregressive models [33, 15], generative adversarial networks (GANs) [67, 25, 100, 99], variational autoencoders (VAEs) [21, 104, 72], and diffusion models [53, 16, 101, 71].

**Regular time series** For the first experiment, we follow [100, 101] and use four real-world datasets Stocks, ETTh, Energy, fMRI and two simulated datasets Sines, MuJoCo of length-24 time series. We use four existing metrics to measure the quality of generated sequences. Context-Fréchet distance (context-FID) score [44] measures the discrepancy between the distributions of features of the real and generated data encoded by a pre-trained TSVec model [102]. Correlational score [59] measures the absolute error between cross-correlation matrices of the real and generated data. Discriminative score [101] trains a GRU classifier to distinguish between the real and generated data, and measures how close its accuracy is to the chance level $50\%$. Predictive score [101] trains a GRU next-step predictor on the generated data, and measures the mean absolute error (MAE) on the real data.

The main results are in Table 1. Spectral mean flow achieves the best metric in 18 out of 24 cases, showing that it is competitive with the state-of-the-art Diffusion-TS [101]. The gain in the quality metric is often significant, e.g., our approach improves previous best context-FID, correlational score,

Table 1: Time-series generative modeling.

| Metric | Methods | Sines | Stocks | ETTh | MuJoCo | Energy | fMRI |
|---|---|---|---|---|---|---|---|
| Context-FID Score ↓ | Ours | **0.004±.001** | **0.008±.003** | **0.058±.007** | 0.018±.002 | **0.051±.009** | **0.116±.004** |
|  | Diffusion-TS | 0.013±.001 | 0.169±.021 | 0.126±.007 | **0.015±.001** | 0.113±.011 | 0.118±.007 |
|  | DiffTime | 0.006±.001 | 0.236±.074 | 0.299±.044 | 0.188±.028 | 0.279±.045 | 0.340±.015 |
|  | Diffwave | 0.014±.002 | 0.232±.032 | 0.873±.061 | 0.393±.041 | 1.031±.131 | 0.244±.018 |
|  | TimeGAN | 0.101±.014 | 0.103±.013 | 0.300±.013 | 0.563±.052 | 0.767±.103 | 1.292±.218 |
|  | TimeVAE | 0.307±.060 | 0.215±.035 | 0.805±.186 | 0.251±.015 | 1.631±.142 | 14.449±.969 |
|  | Cot-GAN | 1.337±.068 | 0.408±.086 | 0.980±.071 | 1.094±.079 | 1.039±.028 | 7.813±.550 |
| Correlational Score ↓ | Ours | 0.027±.012 | 0.010±.007 | **0.040±.015** | **0.173±.016** | **0.732±.107** | **0.737±.021** |
|  | Diffusion-TS | **0.016±.005** | 0.010±.009 | 0.049±.013 | 0.188±.035 | 0.788±.075 | 1.252±.070 |
|  | DiffTime | 0.017±.004 | **0.006±.002** | 0.067±.005 | 0.218±.031 | 1.158±.095 | 1.501±.048 |
|  | Diffwave | 0.022±.005 | 0.030±.020 | 0.175±.006 | 0.579±.018 | 5.001±.154 | 3.927±.049 |
|  | TimeGAN | 0.045±.010 | 0.063±.005 | 0.210±.006 | 0.886±.039 | 4.010±.104 | 23.502±.039 |
|  | TimeVAE | 0.131±.010 | 0.095±.008 | 0.111±020 | 0.388±.041 | 1.688±.226 | 17.296±.526 |
|  | Cot-GAN | 0.049±.010 | 0.087±.004 | 0.249±.009 | 1.042±.007 | 3.164±.061 | 26.824±.449 |
| Discriminative Score ↓ | Ours | **0.006±.006** | **0.022±.013** | **0.027±.010** | **0.005±.004** | 0.161±.021 | **0.136±.207** |
|  | Diffusion-TS | 0.030±.006 | 0.085±.026 | 0.075±.007 | 0.012±.006 | **0.154±.012** | 0.158±.020 |
|  | DiffTime | 0.013±.006 | 0.097±.016 | 0.100±.007 | 0.154±.045 | 0.445±.004 | 0.245±.051 |
|  | Diffwave | 0.017±.008 | 0.232±.061 | 0.190±.008 | 0.203±.096 | 0.493±.004 | 0.402±.029 |
|  | TimeGAN | 0.011±.008 | 0.102±.021 | 0.114±.055 | 0.238±.068 | 0.236±.012 | 0.484±.042 |
|  | TimeVAE | 0.041±.044 | 0.145±.120 | 0.209±.058 | 0.230±.102 | 0.499±.000 | 0.476±.044 |
|  | Cot-GAN | 0.254±.137 | 0.230±.016 | 0.325±.099 | 0.426±.022 | 0.498±.002 | 0.492±.018 |
|  | RNN-AR | 0.495±.001 | 0.226±.035 | - | - | 0.483±.004 | - |
| Predictive Score ↓ | Ours | **0.093±.000** | **0.037±.000** | 0.123±.005 | 0.008±.001 | **0.251±.000** | **0.100±.000** |
|  | Diffusion-TS | 0.095±.000 | **0.037±.000** | **0.121±.002** | **0.007±.001** | **0.251±.000** | **0.100±.000** |
|  | DiffTime | **0.093±.000** | 0.038±.001 | **0.121±.004** | 0.010±.001 | 0.252±.000 | **0.100±.000** |
|  | Diffwave | **0.093±.000** | 0.047±.000 | 0.130±.001 | 0.013±.000 | **0.251±.000** | 0.101±.000 |
|  | TimeGAN | **0.093±.019** | 0.038±.001 | 0.124±.001 | 0.025±.003 | 0.273±.004 | 0.126±.002 |
|  | TimeVAE | **0.093±.000** | 0.039±.000 | 0.126±.004 | 0.012±.002 | 0.292±.000 | 0.113±.003 |
|  | Cot-GAN | 0.100±.000 | 0.047±.001 | 0.129±.000 | 0.068±.009 | 0.259±.000 | 0.185±.003 |
|  | RNN-AR | 0.150±.022 | 0.038±.001 | - | - | 0.315±.005 | - |
|  | Original | 0.094±.001 | 0.036±.001 | 0.121±.005 | 0.007±.001 | 0.250±.003 | 0.090±.001 |

Table 2: Time-series modeling in larger model regime.

| Metric | Methods | Sines | Stocks | MuJoCo |
|---|---|---|---|---|
| Context-FID Score ↓ | Ours | **0.002±.000** | **0.004±.001** | 0.013±.001 |
|  | SDFormer-AR | 0.008±.001 | 0.006±.001 | **0.008±.000** |
|  | SDFormer-M | 0.010±.002 | 0.034±.008 | 0.030±.003 |
|  | ImagenTime | 0.009±.001 | 0.011±.002 | 0.017±.002 |
| Discriminative Score ↓ | Ours | **0.007±.008** | 0.012±.013 | **0.009±.009** |
|  | SDFormer-AR | 0.016±.010 | **0.006±.006** | **0.009±.006** |
|  | SDFormer-M | 0.008±.004 | 0.020±.011 | 0.025±.007 |
|  | ImagenTime | 0.016±.010 | 0.010±.007 | 0.011±.005 |
| Predictive Score ↓ | Ours | **0.093±.000** | **0.037±.000** | 0.008±.001 |
|  | SDFormer-AR | **0.093±.000** | **0.037±.000** | 0.008±.002 |
|  | SDFormer-M | **0.093±.000** | **0.037±.000** | **0.007±.001** |
|  | ImagenTime | 0.095±.000 | **0.037±.000** | 0.033±.002 |

Table 3: Long time-series modeling.

| Metric | Methods | FRED-MD | NN5 Daily |
|---|---|---|---|
| Marginal Score ↓ | Ours | **0.019±n.a.** | **0.006±n.a.** |
|  | ImagenTime | 0.022±n.a. | 0.009±n.a. |
|  | LS4 | 0.022±n.a. | 0.007±n.a. |
|  | SaShiMi-AR | 0.048±n.a. | 0.020±n.a. |
| Classification Score ↑ | Ours | **1.338±.753** | **0.950±.257** |
|  | ImagenTime | 0.755±.343 | 0.560±.174 |
|  | LS4 | 0.544±n.a. | 0.636±n.a. |
|  | SaShiMi-AR | 0.001±n.a. | 0.045±n.a. |
| Predictive Score ↓ | Ours | **0.030±.006** | 0.539±.196 |
|  | ImagenTime | 0.034±.020 | 0.584±.188 |
|  | LS4 | 0.037±n.a. | **0.241±n.a.** |
|  | SaShiMi-AR | 0.232±n.a. | 0.849±n.a. |

and discriminative score in ETTh by around 50% on average, while predictive score does not lag much behind Diffusion-TS. The results are notable considering that previous methods often make use of carefully designed auxiliary losses specialized for time-series, e.g., Diffusion-TS uses a Fourier domain loss to improve the handling of periodicity. In contrast, our models only use standard flow matching loss (3.19) for training, which illustrates the advantages of principled architecture design. Specifically, spectral expansion induces the use of complex-valued parameters, and tensor network decomposition induces a multiplicative parameter sharing in the length direction (Section 3.1), offering a natural inductive bias to handle periodicity without specialized loss or components.

To provide further comparisons in a larger parameter regime, we conduct an additional experiment following [15, 71], comparing against larger models than ones in Table 1. We use Sines, Stocks, and MuJoCo considering resource constraints. The results are in Table 2. Spectral mean flow achieves the best in 6 out of 9 cases, showing that it is competitive with the state-of-the-art SDFormer-AR [15] and generally outperforms SDFormer-M [15] and ImagenTime [71]. Similarly as in Table 1, we note that SDFormer-AR uses a sophisticated two-stage training of a vector-quantized codebook and then an autoregressive model on top of it, while our models are end-to-end trained with flow matching.

Table 4: Irregular time-series modeling based on Stocks dataset, evaluated with discriminative score ↓.

| Task | Methods | 0% Drop | 30% Drop | 50% Drop | 70% Drop |
|---|---|---|---|---|---|
| Irregular → Regular | Ours | **0.009±.008** | **0.020±.011** | **0.019±.008** | **0.015±.007** |
| | Koopman VAE | 0.021±.022 | 0.109±.051 | 0.067±.038 | 0.049±.052 |
| | GT-GAN | 0.077±.031 | 0.251±.097 | 0.265±.073 | 0.230±.053 |
| | TimeGAN | 0.102±.021 | 0.411±.040 | 0.477±.021 | 0.485±.022 |
| | RCGAN | 0.196±.027 | 0.436±.064 | 0.478±.049 | 0.381±.086 |
| | C-RNN-GAN | 0.399±.028 | 0.500±.000 | 0.500±.000 | 0.500±.000 |
| | RNN-AR | 0.226±.035 | 0.305±.002 | 0.308±.010 | 0.317±.019 |
| Irregular → Irregular | Ours | **0.009±.008** | **0.049±.017** | **0.044±.017** | **0.138±.137** |
| | Koopman VAE | 0.021±.022 | 0.227±.096 | 0.211±.078 | 0.187±.075 |

Table 5: Physics-informed modeling of a nonlinear pendulum.

| Methods | Corr. Score ↓ |
|---|---|
| Ours w/ stability loss | **0.0005±.0004** |
| KoVAE w/ stability loss | 0.0030±.0004 |
| Ours w/o stability loss | **0.0029±.0008** |
| KoVAE w/o stability loss | 0.0040±.0005 |

**Long time series**   For additional demonstrations of modeling longer time series than in Table 1, we use FRED-MD and NN5 Daily from the Monash repository [32], containing time series of lengths 728 and 791, respectively. Our models are designed as in Section 3, equipped with time-delay observables used in operator-theoretic methods for dynamical systems [49]. We use three metrics following [71]. Marginal score [59] measures the absolute error between the empirical probability densities of the real and generated data. Classification score [104] trains an S4 [37] classifier to distinguish between the real and generated data, and measures its loss on the generated data. Predictive score [104] trains an S4 10-step predictor on the generated data, and measures the prediction error on the real data. The results are in Table 3. Spectral mean flow shows a strong result, achieving the best metric in 5 out of 6 cases and generally outperforming the previous best methods ImagenTime [71] and LS4 [104].

**Irregular time series**   For further demonstrations of generality, we run experiments on irregularly sampled time series. We obtain 3 irregularly sampled datasets from Stocks by randomly dropping 30%, 50%, and 70% of the observations, following [45, 72]. We consider two tasks, irregular → regular: generating the full time series including the missing timesteps, and irregular → irregular: generating only the irregularly sampled time series. The former is standard in literature [45, 72], and the latter provides a proxy task of modeling informatively sampled time series where sampling can be determined by the system state. Baselines include Koopman VAE [72], state-of-the-art operator-based method for irregular time series. The results are in Table 4. Spectral mean flow achieves the best discriminative metric, supporting its generality to handle irregular time series in both settings.

**Physics-informed modeling**   Lastly, we demonstrate a benefit of being an operator method: incorporation of physics-based prior knowledge. While interpreting eigenfunctions is not direct due to our end-to-end design, this can be done via explicit spectral regularization thanks to linearity of the tensor network. We test this using a problem from [72], where we consider a nonlinear pendulum governed by an ODE of angular displacement $\theta$ from an equilibrium, $\ddot{\theta} + 9.8\sin\theta = 0, \dot{\theta}(0) = 0$. As the pendulum is stable and conservative, the physical knowledge is that eigenvalues of underlying operator have non-positive real part and some have zero real. In Koopman VAE [72], this stability constraint is built-in using a loss on matrix $A$ for hidden states $\mathbf{z}_{t+1} = A\mathbf{z}_t$, specifically $|s_1 - 1|^2 + |s_2 - 1|^2$ where $(s_1, s_2)$ are the largest eigenvalues of $A$. As our model is governed by products of matrix hidden states $O_t w(\mathbf{x}^n), S_t w(\mathbf{x}^n)$ (3.10), (3.20), we incorporate this in an end-to-end manner via the same spectral loss on the matrix states. The results are in Table 5. With stability loss, we see a clear improvement over Koopman VAE that uses the same constraint. This implies incorporating physical knowledge into our model is possible, leading to a better match with the true data-generating process.

## 6   Conclusion

We proposed a new algorithm for sequence modeling based on operator-theoretic interpretation of a hidden Markov model. Instead of implementing its stochastic recurrence directly, we considered embedded distributions in Hilbert spaces, which enabled us to use powerful linear-algebraic tools including spectral decomposition to derive a tensor-network based neural architecture, as well as a sampling procedure based on a time-dependent MMD gradient flow paired with flow matching. On synthetic setups and time-series modeling datasets, we verified our theoretical claims and observed performances competitive with the state-of-the-art.

**Acknowledgments**    This work was in part supported by the National Research Foundation of Korea (RS-2024-00351212 and RS-2024-00436165) and the Institute of Information & Communications Technology Planning & Evaluation (IITP) (RS-2024-00509279, RS-2022-II220926, and RS-2022-II220959) funded by the Korean government (MSIT). Max Beier is supported by the DAAD programme Konrad Zuse Schools of Excellence in Artificial Intelligence, sponsored by the German Federal Ministry of Education.

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

## A    Mathematical Background

We provide an overview of the necessary mathematical background, and refer the reader to Mollenhauer & Koltai (2020) [68] and Arbel et al. (2019) [4] for more details.

### A.1    Problem Setup

For a Polish space $\mathcal{S}$, we define $\mathcal{P}(\mathcal{S})$ as the set of probability distributions on $\mathcal{S}$. For any $\pi \in \mathcal{P}(\mathcal{S})$, we define $L^2(\pi)$ as the set of real-valued Lebesgue square integrable functions on $\mathcal{S}$ with respect to $\pi$. Any closed subset $\mathcal{X} \subset \mathbb{R}^d$, and its product $\mathcal{X}^N$ for a finite $N \in \mathbb{N}$, are Polish spaces.

For closed $\mathcal{X} \subset \mathbb{R}^d$ and $\mathcal{Z} \subset \mathbb{R}^m$, we consider a sequence of random variables $X^{1:N} = (X^1, ..., X^N)$ on $\mathcal{X}$ and model it as a hidden Markov model (HMM) with hidden states $Z^{1:N} = (Z^1, ..., Z^N)$ on $\mathcal{Z}$. The joint distribution of hidden states $Z^{1:N}$ is determined by a time-invariant state transition model $\mathbb{P}[Z^{n+1}|Z^n]$, and determines $X^{1:N}$ through a time-invariant local observation model $\mathbb{P}[X^n|Z^n]$.

Formally, the conditional distributions $\mathbb{P}[Z^{n+1}|Z^n]$ and $\mathbb{P}[X^n|Z^n]$ are specified by the respective Markov kernels $u$ and $o$ that define $u(\mathbf{z}, \cdot) \in \mathcal{P}(\mathcal{Z})$ and $o(\mathbf{z}, \cdot) \in \mathcal{P}(\mathcal{X})$ for every $\mathbf{z} \in \mathcal{Z}$, such that:

$$\mathbb{P}[Z^{n+1} \in \mathcal{A} \mid Z^n = \mathbf{z}] = \int_{\mathcal{A}} u(\mathbf{z}, d\mathbf{z}') = u(\mathbf{z}, \mathcal{A}), \tag{A.1}$$

$$\mathbb{P}[X^n \in \mathcal{B} \mid Z^n = \mathbf{z}] = \int_{\mathcal{B}} o(\mathbf{z}, d\mathbf{x}) = o(\mathbf{z}, \mathcal{B}), \tag{A.2}$$

for all measurable sets $\mathcal{A} \subseteq \mathcal{Z}$ and $\mathcal{B} \subseteq \mathcal{X}$.

### A.2    Reproducing Kernel Hilbert Space (RKHS)

Let $\mathbb{H}$ be an RKHS induced by a positive semi-definite kernel function $k : \mathcal{X} \times \mathcal{X} \to \mathbb{R}$. The space $\mathbb{H}$ is a vector space consisting of functions $f : \mathcal{X} \to \mathbb{R}$, equipped with an inner product $\langle \cdot, \cdot \rangle_{\mathbb{H}}$ and a corresponding norm $\| \cdot \|_{\mathbb{H}}$ given by $\|f\|_{\mathbb{H}}^2 = \langle f, f \rangle_{\mathbb{H}}$. The canonical feature map $\phi : \mathcal{X} \to \mathbb{H}$ of the RKHS is defined by $\phi(\mathbf{x}) := k(\cdot, \mathbf{x})$. The reproducing property of $\mathbb{H}$ states that $f(\mathbf{x}) = \langle \phi(\mathbf{x}), f \rangle_{\mathbb{H}}$ for any $f \in \mathbb{H}$. We denote the space of Hilbert-Schmidt operators from $\mathbb{H}$ to $\mathbb{H}$ as $S_2(\mathbb{H})$.

To handle conditional dependencies within an HMM, we also consider a second RKHS $\mathbb{G}$ induced by a kernel $l : \mathcal{Z} \times \mathcal{Z} \to \mathbb{R}$ with the corresponding canonical feature map $\varphi : \mathcal{Z} \to \mathbb{G}$. We denote the space of Hilbert-Schmidt operators from $\mathbb{G}$ to $\mathbb{H}$ as $S_2(\mathbb{G}, \mathbb{H})$.

We use some assumptions on the RKHS $\mathbb{H}$ (or $\mathbb{G}$) and the corresponding kernel $k$ (or $l$). The first five assumptions allow universal approximation of conditional distributions via linear operators on $\mathbb{H}$ [68, Section 4.5]. The sixth one allows decomposing an HMM embedded in $\mathbb{H}$ into a tensor network.

- **(A)** (Separability). The RKHS $\mathbb{H}$ is separable. This holds if $\mathcal{X}$ is Polish and $k$ is continuous.
- **(B)** (Measurability). The canonical feature map $\phi : \mathcal{X} \to \mathbb{H}$ is measurable. This holds if $k(\mathbf{x}, \cdot)$ is measurable for all $\mathbf{x} \in \mathcal{X}$.
- **(C)** (Existence of second moments). It holds that $\mathbb{E}_{\mathbf{x} \sim \pi}[\|\phi(\mathbf{x})\|_{\mathbb{H}}^2] < \infty$ for all $\pi \in \mathcal{P}(\mathcal{X})$. This is satisfied if $\sup_{\mathbf{x} \in \mathcal{X}} k(\mathbf{x}, \mathbf{x}) < \infty$.
- **(D)** ($C_0$-kernel). $\mathbb{H} \in C_0(\mathcal{X})$, where $C_0(\mathcal{X})$ is the space of continuous real functions on $\mathcal{X}$ vanishing at infinity. This holds if $\mathbf{x} \mapsto k(\mathbf{x}, \mathbf{x})$ is bounded on $\mathcal{X}$ and $k(\mathbf{x}, \cdot) \in C_0(\mathcal{X})$ for all $\mathbf{x} \in \mathcal{X}$.
- **(E)** ($L^2$-universal kernel). $\mathbb{H}$ is dense in $L^2(\pi)$ for all probability measures $\pi \in \mathcal{P}(\mathcal{X})$.
- **(F)** (Forming an algebra). $\mathbb{H}$ is closed under pointwise multiplication, $f, g \in \mathbb{H} \implies f \cdot g \in \mathbb{H}$, with the multiplication map $(f, g) \mapsto f \cdot g$ bounded.

An example RKHS satisfying all the assumptions is Sobolev space $H^s(\Omega \subset \mathbb{R}^d)$ of order $s > d/2$ [1, Theorem 4.39], motivating a natural connection to neural tangent kernels [8, 86]. Generally, a class of RKHSs satisfying **(F)** is known as reproducing kernel Hilbert algebras (RKHAs) [69, Definition 2.1]. Recent results have shown that **(F)** may hold under more abstract regularity conditions [18, 19, 69].

## A.3 Mean Embeddings of (Conditional) Distributions

Under Assumptions (A) to (C), for any $X \sim \pi \in \mathcal{P}(\mathcal{X})$, the following expectation yields an element $\mu_\pi$ in $\mathbb{H}$ which is called the mean embedding of $\pi$ [70]:

$$\mu_X := \int_{\mathcal{X}} \phi(X) \, d\pi(\mathbf{x}) = \mathbb{E}[\phi(X)] \in \mathbb{H}. \tag{A.3}$$

We write $\mu_X = \mu_\pi$ when the distribution is clear from context. We call the RKHS $\mathbb{H}$ characteristic if the mean embedding map $\pi \mapsto \mu_\pi$ is injective, that is, each mean embedding uniquely encodes a distribution. The Assumptions (D) and (E) imply that $\mathbb{H}$ is characteristic [13, 91].

With a characteristic $\mathbb{H}$, we can define a distance metric between distributions in $\mathcal{P}(\mathcal{X})$ called the maximum mean discrepancy (MMD), defined as follows [34]:

$$\mathrm{MMD}(\nu, \pi) := \sup_{f \in \mathbb{H}, \|f\|_{\mathbb{H}} \leq 1} \left| \int f(\mathbf{x}) \, d\nu(\mathbf{x}) - \int f(\mathbf{x}) \, d\pi(\mathbf{x}) \right| = \|\mu_\nu - \mu_\pi\|_{\mathbb{H}}, \tag{A.4}$$

which we can interpret as measuring differences between mean embeddings.

To handle distributions in $\mathcal{P}(\mathcal{Z})$, we analogously define mean embeddings and MMD for RKHS $\mathbb{G}$.

We can extend the concept of mean embeddings to the conditional distributions $\mathbb{P}[Z^{n+1}|Z^n]$ and $\mathbb{P}[X^n|Z^n]$ of an HMM. This is formalized by conditional mean embeddings [70, 89, 76], defined as:

$$\mu_{Z^{n+1}|Z^n=\mathbf{z}} := \int_{\mathcal{Z}} \varphi(\mathbf{z}') \, u(\mathbf{z}, d\mathbf{z}') = \mathbb{E}[\varphi(Z^{n+1}) \mid Z^n = \mathbf{z}], \tag{A.5}$$

$$\mu_{X^n|Z^n=\mathbf{z}} := \int_{\mathcal{X}} \phi(\mathbf{x}) \, o(\mathbf{z}, d\mathbf{x}) = \mathbb{E}[\phi(X^n) \mid Z^n = \mathbf{z}]. \tag{A.6}$$

In kernel-based inference, an approximation of the above is often achieved using linear operators $U : \mathbb{G} \to \mathbb{G}$ and $O : \mathbb{G} \to \mathbb{H}$ called the conditional mean embedding (CME) operators [89, 29, 68]:

$$U\varphi(\mathbf{z}) \approx \mathbb{E}[\varphi(Z^{n+1}) \mid Z^n = \mathbf{z}], \tag{A.7}$$

$$O\varphi(\mathbf{z}) \approx \mathbb{E}[\phi(X^n) \mid Z^n = \mathbf{z}]. \tag{A.8}$$

For (A.7), if the RKHS $\mathbb{H}$ satisfies Assumptions (A) to (E), it is known that the approximation in (A.7) can be made arbitrarily precise for a choice of the operator norm [68, Theorem 3.3]. Furthermore, for well-specified cases where the map $\mathbf{z} \mapsto \mu_{Z^{n+1}|Z^n=\mathbf{z}}$ is identified as an element of $\mathrm{S}_2(\mathbb{G})$, a choice of $U \in \mathrm{S}_2(\mathbb{G})$ exists which achieves zero error $U\varphi(\mathbf{z}) = \mathbb{E}[\varphi(Z^{n+1}) \mid Z^n = \mathbf{z}]$ [68, Corollary 5.5].

For (A.8), we need an extension of [68, Corollary 5.5] to different domains $\mathcal{X}$ and $\mathcal{Z}$ [74]. This is possible, since the result applies as long as there is a Markov kernel connecting the underlying measure spaces, which requires mild regularity conditions [48, Chapter 8, Theorem 8.5 and preliminaries]. Under these conditions, and if $\mathbb{H}$ and $\mathbb{G}$ satisfy Assumptions (A) to (E), we can consider well-specified cases where a choice of $O \in \mathrm{S}_2(\mathbb{G}, \mathbb{H})$ exists such that $O\varphi(\mathbf{z}) = \mathbb{E}[\phi(X^n) \mid Z^n = \mathbf{z}]$.

**Remark A.1.** *Assuming well-specified CME operators is a standard choice in kernel-based statistical learning [68, 74]. Even if the assumption is not exactly met, close approximations can still be achieved, e.g., representation learning can yield a feature space that is well-specified for the problem at hand.*

## A.4 MMD Gradient Flow

Let $\mathcal{P}_2(\mathcal{X})$ be the set of distributions on $\mathcal{X}$ with finite second moment equipped with the 2-Wasserstein metric. Under Assumptions (A) to (E), the MMD (A.4) provides a natural process for generating samples $\mathbf{x} \sim \pi \in \mathcal{P}_2(\mathcal{X})$ from an embedded distribution $\mu_\pi$ (A.3).

Specifically, the gradient flow of the MMD defines a continuous probability path $(p_t)_{t \geq 0}$ which starts at any initial distribution $p_0$ and converges towards the target $p_t \to \pi$ as $t \to \infty$. Let us fix $\pi$, and define $\mathcal{F}(p_t) := \frac{1}{2}\mathrm{MMD}^2(p_t, \pi)$, which measures distance between $p_t$ and $\pi$. It is known that the time-dependent vector field $(v_t)_{t \geq 0}$ corresponding to the gradient flow of $\mathcal{F}(p_t)$ is given by [4]:

$$v_t(\mathbf{x}) = -\nabla_{\mathbf{x}}(\mu_{p_t} - \mu_\pi)(\mathbf{x}) = -\nabla_{\mathbf{x}}\langle \phi(\mathbf{x}), \mu_{p_t} - \mu_\pi \rangle_{\mathbb{H}}, \tag{A.9}$$

which generates the path $(p_t)_{t \geq 0}$ via the continuity equation $\partial_t p_t + \mathrm{div}(p_t v_t) = 0$. If $k$ is continuously differentiable on $\mathcal{X}$ with Lipschitz gradient, for any initial distribution $p_0 \in \mathcal{P}_2(\mathcal{X})$, there exists a unique process $(X_t)_{t \geq 0}$ from $X_0 \sim p_0$ satisfying $dX_t = v_t(X_t)dt$, where the distribution $p_t$ of $X_t$ is the unique solution of the continuity equation and $\mathcal{F}(p_t)$ decreases in time [4, Proposition 1, 2]. With further regularity conditions, it can be shown that $p_t \to \pi$ asymptotically as $t \to \infty$ [4, Section 3].

# B Theoretical Results

We formally state and prove all theoretical arguments in the main text.

## B.1 Tensor Product Mean Embedding (Section 3)

In Section 3, we embedded the joint distribution $X^{1:N}$ in the tensor product $\mathbb{H}^{\otimes N} = \mathbb{H} \otimes \cdots \otimes \mathbb{H}$ of a characteristic RKHS $\mathbb{H}$. To establish that $\mathbb{H}^{\otimes N}$ is itself a characteristic RKHS, we use the following:

**Lemma B.1.** *Let Assumptions (D) and (E) be satisfied for $\mathbb{H}$. Then $\mathbb{H}^{\otimes N}$ is characteristic.*

*Proof.* From [68, Remark 4.7], Assumption (D) implies: $\mathbb{H}$ is $L^2$-universal $\iff$ $\mathbb{H}$ is $C_0$-universal, i.e., $\mathbb{H}$ is dense in $C_0(\mathcal{X})$ with respect to the supremum norm. By Assumption (E), we have that $\mathbb{H}$ is $L^2$-universal, and hence $C_0$-universal. Then, from [93, Section 4], we have that $\mathbb{H}$ is $C_0$-universal $\iff$ $\mathbb{H}^{\otimes N}$ is characteristic. Therefore, under Assumptions (D) and (E), $\mathbb{H}^{\otimes N}$ is characteristic. $\square$

## B.2 Mean Embedding Decomposition (Section 3.1)

In Section 3.1, we derived tensor network decompositions of the sequence mean embeddings $\mu_\rho$ and $\mu_{p_t}$ through (3.7) and (3.9). We provide the respective proofs in Proposition B.2 and Proposition B.6.

**Proposition B.2.** *Let Assumptions (A) to (E) be satisfied for $\mathbb{G}$ and $\mathbb{H}$. Assume that for $\mathbb{P}[X^n|Z^n]$ and $\mathbb{P}[X_t^n|Z^n] \forall t \geq 0$, well-specified CME operators $O : \mathbb{G} \to \mathbb{H}$ and $S_t : \mathbb{G} \to \mathbb{H}$ exist, that is:*

$$O\varphi(\mathbf{z}) = \mathbb{E}[\phi(X^n) \mid Z^n = \mathbf{z}] \in \mathbb{H}, \tag{B.1}$$

$$S_t\varphi(\mathbf{z}) = \mathbb{E}[\phi(X_t^n) \mid Z^n = \mathbf{z}] \in \mathbb{H}, \tag{B.2}$$

*for all $\mathbf{z} \in \mathcal{Z}$. Then the following holds:*

$$\mu_\rho = O^{\otimes N} \mu_{Z^{1:N}}, \tag{B.3}$$

$$\mu_{p_t} = S_t^{\otimes N} \mu_{Z^{1:N}}. \tag{B.4}$$

*Proof.* The proof for (B.3) is as follows. We use two properties of HMMs: conditional independence of observations $X^i \perp\!\!\!\perp X^j \mid Z^{1:N}$ for $i \neq j$ and locality of observation $\mathbb{P}[X^n|Z^{1:N}] = \mathbb{P}[X^n|Z^n]$.

$$
\begin{aligned}
\mu_\rho &:= \mathbb{E}[\phi(X^1) \otimes \cdots \otimes \phi(X^N)] && \text{(by definition)} \\
&= \mathbb{E}\left[\mathbb{E}[\phi(X^1) \otimes \cdots \otimes \phi(X^N) \mid Z^{1:N}]\right] && \text{(law of total expectation)} \\
&= \mathbb{E}\left[\mathbb{E}[\phi(X^1)|Z^{1:N}] \otimes \cdots \otimes \mathbb{E}[\phi(X^N)|Z^{1:N}]\right] && \text{(conditional independence)} \\
&= \mathbb{E}\left[\mathbb{E}[\phi(X^1)|Z^1] \otimes \cdots \otimes \mathbb{E}[\phi(X^N)|Z^N]\right] && \text{(locality)} \\
&= \mathbb{E}\left[[O\varphi(Z^1)] \otimes \cdots \otimes [O\varphi(Z^N)]\right] && \text{(CME operator (B.1))} \\
&= \mathbb{E}\left[O^{\otimes N}[\varphi(Z^1) \otimes \cdots \otimes \varphi(Z^N)]\right] && \text{(tensor product of linear operators)} \\
&= O^{\otimes N} \mathbb{E}[\varphi(Z^1) \otimes \cdots \otimes \varphi(Z^N)] && \text{(linearity)} \\
&= O^{\otimes N} \mu_{Z^{1:N}}. && \text{(by definition)}
\end{aligned}
$$

The proof for (B.4) is identical, by substituting $O$ with $S_t$. $\square$

We now prove tensor network decomposition of $\mu_{Z^{1:N}}$ (3.9). We show some useful lemmas:

**Lemma B.3.** *Let $\mathbb{G}$ be an RKHS on a set $\mathcal{Z}$ with norm $\|\cdot\|_{\mathbb{G}}$. Suppose Assumption (F) holds, so there exists a constant $C \geq 0$ such that*

$$\|f \cdot g\|_{\mathbb{G}} \leq C \|f\|_{\mathbb{G}} \|g\|_{\mathbb{G}} \quad \text{for all } f, g \in \mathbb{G}.$$

*Then the bilinear map $(f, g) \mapsto f \cdot g$ extends to a bounded linear operator*

$$T^* : \mathbb{G} \otimes \mathbb{G} \longrightarrow \mathbb{G}, \quad T^*(f \otimes g) = f \cdot g,$$

*with $\|T^*\| \leq C$. Consequently its adjoint*

$$T = (T^*)^* : \mathbb{G} \longrightarrow \mathbb{G} \otimes \mathbb{G}$$

*exists and satisfies $\|T\| = \|T^*\| \leq C$.*

*Proof.* Define $B : \mathbb{G} \times \mathbb{G} \to \mathbb{G}$ by $B(f, g) = f \cdot g$. By assumption

$$\|B(f, g)\|_{\mathbb{G}} = \|f \cdot g\|_{\mathbb{G}} \leq C \|f\|_{\mathbb{G}} \|g\|_{\mathbb{G}},$$

so $B$ is a bounded bilinear map. By the universal property of the Hilbert-space tensor product, $B$ extends uniquely to a bounded linear operator

$$T^* : \mathbb{G} \otimes \mathbb{G} \longrightarrow \mathbb{G}, \quad T^*(f \otimes g) = f \cdot g,$$

with $\|T^*\| \leq C$.

Since $T^*$ is bounded between Hilbert spaces, its adjoint $T = (T^*)^*$ exists and $\|T\| = \|T^*\| \leq C$. $\square$

**Corollary B.4.** *Under the setup of Lemma B.3, we have $T\varphi(\mathbf{z}) = \varphi(\mathbf{z}) \otimes \varphi(\mathbf{z})$ for all $\mathbf{z} \in \mathcal{Z}$.*

*Proof.* This immediately follows from the definition of $T^*$ and the reproducing property:

$$\langle T\varphi(\mathbf{z}), f \otimes g \rangle_{\mathbb{G} \otimes \mathbb{G}} = \langle \varphi(\mathbf{z}), T^*(f \otimes g) \rangle_{\mathbb{G}} = \langle \varphi(\mathbf{z}), f \cdot g \rangle_{\mathbb{G}} = (f \cdot g)(\mathbf{z}) = f(\mathbf{z})g(\mathbf{z})$$
$$= \langle \varphi(\mathbf{z}) \otimes \varphi(\mathbf{z}), f \otimes g \rangle_{\mathbb{G} \otimes \mathbb{G}} \quad \text{for all } f, g \in \mathbb{G} \text{ and } \mathbf{z} \in \mathcal{Z}. \tag{B.5}$$

Since the collection of elementary tensors $\{f \otimes g : f, g \in \mathbb{G}\}$ spans the tensor product space $\mathbb{G} \otimes \mathbb{G}$, it follows that $\varphi(\mathbf{z}) \otimes \varphi(\mathbf{z}) = T\varphi(\mathbf{z})$ for all $\mathbf{z}$. $\square$

**Lemma B.5.** *Let Assumptions (A) to (F) be satisfied for $\mathbb{G}$. Assume that for $\mathbb{P}[Z^{n+1}|Z^n]$, a well-specified CME operator $U : \mathbb{G} \to \mathbb{G}$ exists, i.e., it satisfies the following for all $\mathbf{z} \in \mathcal{Z}$:*

$$U\varphi(\mathbf{z}) = \mathbb{E}[\varphi(Z^{n+1}) \mid Z^n = \mathbf{z}] \in \mathbb{G}. \tag{B.6}$$

*Then, there exists a bounded linear operator $T : \mathbb{G} \to \mathbb{G} \otimes \mathbb{G}$ that satisfies the below for all $\mathbf{z} \in \mathcal{Z}$:*

$$TU\varphi(\mathbf{z}) = \mathbb{E}[\varphi(Z^{n+1}) \otimes \varphi(Z^{n+1}) \mid Z^n = \mathbf{z}] \in \mathbb{G} \otimes \mathbb{G}. \tag{B.7}$$

*Proof.* Let $T$ be the bounded linear operator from Lemma B.3. Then:

$$\mathbb{E}[\varphi(Z^{n+1}) \otimes \varphi(Z^{n+1}) \mid Z^n = \cdot] = \mathbb{E}[T\varphi(Z^{n+1}) \mid Z^n = \cdot] \qquad \text{(Corollary B.4)}$$
$$= T\mathbb{E}[\varphi(Z^{n+1}) \mid Z^n = \cdot] \qquad \text{(linearity)}$$
$$= TU\varphi. \qquad \text{(CME operator (B.6))}$$

This completes the proof. $\square$

We are now ready to prove the main result (3.9).

**Proposition B.6.** *Under the setup of Lemma B.5 and a choice of decomposition of the CME operator $U = \sum_i \lambda_i h_i \otimes g_i$ with $h_i, g_i \in \mathbb{G}$, the following holds:*

$$\mu_{Z^{1:N}} = \sum_{i_2, \ldots, i_N} b_{i_2} \otimes \lambda_{i_2} w_{i_2, i_3} \otimes \cdots \otimes \lambda_{i_{N-1}} w_{i_{N-1}, i_N} \otimes \lambda_{i_N} h_{i_N}, \tag{B.8}$$

*where $b_i = [T\mu_{Z^1}]g_i \in \mathbb{G}$ and $w_{i,j} = [Th_i]g_j \in \mathbb{G}$.*

*Proof.* We start with the fact that a singular value decomposition (SVD) of $U$ (B.6) always exists and is written as $U = \sum_{i \in \mathbb{N}} \lambda_i h_i \otimes g_i$ [68, Section 4.1] where $\lambda_i \in \mathbb{R}$ are singular values and $h_i, g_i \in \mathbb{G}$ are left and right singular functions, respectively. We proceed with SVD, which is sufficient for the proof, and discuss eigenvalue decomposition (EVD) at the end of this section.

Define $b_i := [T\mu_{Z^1}]g_i$ and $w_{i,j} := [Th_i]g_j$, which are elements of $\mathbb{G}$ from the standard identification of tensors as Hilbert-Schmidt operators $\mathbb{G} \otimes \mathbb{G} \simeq S_2(\mathbb{G})$ [68, Section 4.1]. The following holds:

$$\mathbb{E}[\varphi(Z^N) \mid Z^{N-1}] = \sum_{i_N} \lambda_{i_N} g_{i_N}(Z^{N-1}) h_{i_N}, \tag{B.9}$$

$$\mathbb{E}[\varphi(Z^n) g_{i_{n+1}}(Z^n) \mid Z^{n-1}] = \sum_{i_n} \lambda_{i_n} g_{i_n}(Z^{n-1}) w_{i_n, i_{n+1}}, \tag{B.10}$$

$$\mathbb{E}[\varphi(Z^1) g_{i_2}(Z^1)] = b_{i_2}. \tag{B.11}$$

To avoid clutter, we only write out the derivation of (B.10):

$$\mathbb{E}[\varphi(Z^n)g_{i_{n+1}}(Z^n) \mid Z^{n-1}] = \mathbb{E}[(\varphi(Z^n) \otimes \varphi(Z^n))g_{i_{n+1}} \mid Z^{n-1}] \qquad \text{(reproducing property)}$$
$$= \mathbb{E}[\varphi(Z^n) \otimes \varphi(Z^n) \mid Z^{n-1}]g_{i_{n+1}} \qquad \text{(linearity)}$$
$$= [TU\varphi(Z^{n-1})]g_{i_{n+1}} \qquad \text{(Lemma B.5)}$$
$$= [T\sum_{i_n} \lambda_{i_n} g_{i_n}(Z^{n-1})h_{i_n}]g_{i_{n+1}} \qquad \text{(SVD)}$$
$$= \sum_{i_n} \lambda_{i_n} g_{i_n}(Z^{n-1})[Th_{i_n}]g_{i_{n+1}}. \qquad \text{(linearity)}$$

We now construct a recurrent decomposition of $\mu_{Z^{1:N}}$ using the Markov property of $Z^{1:N}$.

The following reduction rule is useful. Let $K_n := \mathbb{E}[\varphi(Z^n)g_{i_{n+1}}(Z^n) \mid Z^{n-1}]$ for $n \geq 2$, then:

$$\mathbb{E}[\varphi(Z^{n-1}) \otimes K_n \mid Z^{n-2}] = \mathbb{E}[\varphi(Z^{n-1}) \otimes \sum_{i_n} \lambda_{i_n} g_{i_n}(Z^{n-1}) w_{i_n,i_{n+1}} \mid Z^{n-2}] \qquad \text{(by (B.10))}$$
$$= \sum_{i_n} \mathbb{E}[\varphi(Z^{n-1})g_{i_n}(Z^{n-1}) \mid Z^{n-2}] \otimes \lambda_{i_n} w_{i_n,i_{n+1}} \qquad \text{(linearity)}$$
$$= \sum_{i_n} K_{n-1} \otimes \lambda_{i_n} w_{i_n,i_{n+1}}. \qquad \text{(by definition)}$$

The full decomposition is then given as follows:

$$\mu_{Z^{1:N}} = \mathbb{E}[\varphi(Z^1) \otimes \cdots \otimes \varphi(Z^N)] \qquad \text{(by definition)}$$
$$= \mathbb{E}[\varphi(Z^1) \otimes \mathbb{E}[\varphi(Z^2) \otimes \mathbb{E}[\cdots \mathbb{E}[\varphi(Z^{N-1}) \otimes \underbrace{\mathbb{E}[\varphi(Z^N)|Z^{N-1}]}_{(B.9)}|Z^{N-2}]\cdots|Z^2]|Z^1]]$$
$$\text{(Markov property)}$$

$$= \mathbb{E}[\varphi(Z^1) \otimes \cdots \mathbb{E}[\varphi(Z^{N-1}) \otimes \sum_{i_N} \lambda_{i_N} g_{i_N}(Z^{N-1})h_{i_N}|Z^{N-2}]\cdots]$$

$$= \mathbb{E}[\varphi(Z^1) \otimes \cdots \sum_{i_N} \underbrace{\mathbb{E}[\varphi(Z^{N-1})g_{i_N}(Z^{N-1})|Z^{N-2}]}_{K_{N-1}} \otimes \lambda_{i_N} h_{i_N} \cdots] \qquad \text{(linearity)}$$

$$= \mathbb{E}[\varphi(Z^1) \otimes \cdots \mathbb{E}[\varphi(Z^{N-2}) \otimes \sum_{i_N} K_{N-1} \otimes \lambda_{i_N} h_{i_N}|Z^{N-3}]\cdots]$$

$$= \mathbb{E}[\varphi(Z^1) \otimes \cdots \sum_{i_N} \underbrace{\mathbb{E}[\varphi(Z^{N-2}) \otimes K_{N-1}|Z^{N-3}]}_{\text{reduction rule}} \otimes \lambda_{i_N} h_{i_N} \cdots] \qquad \text{(linearity)}$$

$$= \mathbb{E}[\varphi(Z^1) \otimes \cdots \sum_{i_{N-1},i_N} K_{N-2} \otimes \lambda_{i_{N-1}} w_{i_{N-1},i_N} \otimes \lambda_{i_N} h_{i_N} \cdots]$$

$$= \mathbb{E}[\varphi(Z^1) \otimes \cdots \sum_{i_{N-1},i_N} \underbrace{\mathbb{E}[\varphi(Z^{N-3}) \otimes K_{N-2}|Z^{N-4}]}_{\text{reduction rule}} \otimes \lambda_{i_{N-1}} w_{i_{N-1},i_N} \otimes \lambda_{i_N} h_{i_N} \cdots]$$
$$\text{(linearity)}$$

$$= \cdots = \sum_{i_3,\ldots,i_N} \mathbb{E}[\varphi(Z^1) \otimes \underbrace{K_2}_{(B.10)}] \otimes \lambda_{i_3} w_{i_3,i_4} \otimes \cdots \otimes \lambda_{i_N} h_{i_N} \qquad \text{(recurrent reduction)}$$

$$= \sum_{i_2,\ldots,i_N} \underbrace{\mathbb{E}[\varphi(Z^1)g_{i_2}(Z^1)]}_{(B.11)} \otimes \lambda_{i_2} w_{i_2,i_3} \otimes \lambda_{i_3} w_{i_3,i_4} \otimes \cdots \otimes \lambda_{i_N} h_{i_N}$$

$$= \sum_{i_2,\ldots,i_N} b_{i_2} \otimes \lambda_{i_2} w_{i_2,i_3} \otimes \lambda_{i_3} w_{i_3,i_4} \otimes \cdots \otimes \lambda_{i_N} h_{i_N}.$$

This completes the proof. $\qquad\square$

With Propositions B.2 and B.6, the following is obtained from linearity and reproducing property:

**Corollary B.7.** *Let Assumptions (A) to (E) be satisfied for $\mathbb{G}$ and $\mathbb{H}$, and let Assumption (F) be satisfied for $\mathbb{G}$. With well-specified CME operators in Propositions B.2 and B.6, the inner product between the feature of a data $\mathbf{x}^{1:N} \in \mathcal{X}^N$ and the mean embedding $\mu_\rho$ is given as follows:*

$$
\begin{aligned}
\langle \phi(\mathbf{x}^1) \otimes \cdots \otimes \phi(\mathbf{x}^N), \mu_\rho \rangle_{\mathbb{H}^{\otimes N}} &= \langle \phi(\mathbf{x}^1) \otimes \cdots \otimes \phi(\mathbf{x}^N), O^{\otimes N} \mu_{Z^{1:N}} \rangle_{\mathbb{H}^{\otimes N}} \\
&= \sum_{i_2,\ldots,i_N} Ob_{i_2}(\mathbf{x}^1) \cdot \lambda_{i_2} \cdot Ow_{i_2,i_3}(\mathbf{x}^2) \cdots \lambda_{i_N} \cdot Ou_{i_N}(\mathbf{x}^N).
\end{aligned}
$$

(B.12)

*For $\mu_{p_t}$, one only needs to replace $O$ by $S_t$.*

We conclude the section with a discussion on eigenvalue decomposition (EVD), which we adopt in practice. Assuming the CME operator $U$ (B.6) is a normal operator with discrete spectrum, we can invoke spectral theorem to obtain an eigendecomposition $U = \sum_i \lambda_i h_i \otimes g_i$ with eigenvalues $\lambda_i \in \mathbb{C}$ and eigenfunctions $g_i, h_i : \mathcal{X} \to \mathbb{C}$. While SVD already suffices for Proposition B.6, EVD provides a practical benefit as it efficiently captures long-range dependencies via oscillations [75, 7, 12].

**Remark B.8.** *While $U$ operates on real functions, for EVD we include complex-valued eigenfunctions to ensure sufficient expressiveness. This is standard in the analysis of dynamical systems, as operators on a Hilbert space (e.g., Sobolev spaces $H^s(\Omega \subset \mathbb{R}^n)$) with real spectra are self-adjoint and thus can only capture time-reversal invariant dynamics. Normal or bounded operators require complex spectra and eigenfunctions to construct spectral expansions, [80, Definition 12.17 and following].*

**Remark B.9.** *As the spectral expansion is only used to parameterize the operators, the results in Mollenhauer & Koltai (2020) [68] and Arbel et al. (2019) [4] underlying our work are only affected in the sense that, as an additional assumption, a spectral expansion has to exist. A complete extension of the results therein to complex-valued RKHS is out of the scope of this paper.*

## B.3 MMD Flows with Time-dependent RKHS (Section 3.2)

In Section 3.2, we introduced an extension of MMD gradient flows using time-dependent RKHS. The derivation is a direct application of existing work on gradient flows of time-dependent functionals [27].

We use the fact that, for a fixed $\pi$ in $\mathcal{P}_2(\mathbb{R}^d)$, the time-dependent functional $\mathcal{F}_t(\nu) := \frac{1}{2}\mathrm{MMD}_t^2(\nu, \pi)$ in a time interval $t \in [0, 1]$ admits the following free-energy expression [4]:

$$
\mathcal{F}_t(\nu) = \int V_t(\mathbf{x}) \, d\nu(\mathbf{x}) + \frac{1}{2} \int W_t(\mathbf{x}, \mathbf{y}) \, d\nu(\mathbf{x}) \, d\nu(\mathbf{y}) + C_t,
$$

(B.13)

where $V_t, W_t, C_t$ are time-dependent confinement potential, interaction potential, and constant:

$$
V_t(\mathbf{x}) = -\int k_t(\mathbf{x}, \mathbf{x}') \, d\pi(\mathbf{x}'), \quad W_t(\mathbf{x}, \mathbf{x}') = k_t(\mathbf{x}, \mathbf{x}'), \quad C_t = \frac{1}{2} \int k_t(\mathbf{x}, \mathbf{x}') \, d\pi(\mathbf{x}) \, d\pi(\mathbf{x}').
$$

(B.14)

Let us assume regularity conditions for $V_t, W_t$ as in [27, Theorem 6.4, Theorem 6.6 and Remark 6.7]. Then, it follows from the results that there is a continuous path $(p_t)_{t \in [0,1]}$ in $\mathcal{P}_2(\mathbb{R}^d)$ from a given $p_0$ associated to a time-dependent vector field $(v_t)_{t \in [0,1]}$ via the continuity equation $\partial_t p_t + \mathrm{div}(v_t p_t) = 0$. The vector field is obtained as $v_t(\mathbf{x}) = -\nabla_{\mathbf{x}}(V_t + W_t \star p_t)(\mathbf{x})$ [27, Theorem 6.4 and Remark 6.7] where $\star$ is classical convolution, which yields $v_t(\mathbf{x}) = -\nabla(\mu_{p_t,t} - \mu_{\pi,t})(\mathbf{x})$ as in [4, Section 2.1]. This provides a minimal derivation of the flow which is sufficient in our problem context, and we leave in-depth investigation of the regularity conditions as future work.

## B.4 Gradient Fields in Flow Matching (Section 3.3)

In Section 3.3, we used vector fields developed in flow matching [60] to define our training objectives, under the premise that they are gradient fields. We provide formal proofs for all vector fields in [60].

Consider converting an initial distribution $q_0$ to a target distribution $\pi$, by following a continuous path $(q_t)_{t \in [0,1]}$ in $\mathcal{P}_2(\mathcal{X})$ with boundary condition $q_1 = \pi$. Such a path can be identified by a time-dependent vector field $(u_t)_{t \in [0,1]}$ which satisfies the continuity equation $\partial_t u_t + \mathrm{div}(q_t u_t) = 0$ [97].

In flow matching, the probability path $q_t$ and vector field $u_t$ are defined as marginalization of the conditional counterparts $q_t(\cdot|\mathbf{x}_1)$ and $u_t(\cdot|\mathbf{x}_1)$ [60, Eq. (6), (8)], under the assumption that $q_t(\mathbf{x}) > 0$:

$$q_t(\mathbf{x}) = \int q_t(\mathbf{x}|\mathbf{x}_1)\pi(\mathbf{x}_1)\,d\mathbf{x}_1, \tag{B.15}$$

$$u_t(\mathbf{x}) = \int u_t(\mathbf{x}|\mathbf{x}_1)\frac{q_t(\mathbf{x}|\mathbf{x}_1)\pi(\mathbf{x}_1)}{q_t(\mathbf{x})}\,d\mathbf{x}_1. \tag{B.16}$$

Known choices include optimal transport (OT), variance-exploding (VE), and variance-preserving (VP) conditional vector fields [60, Section 4; Examples 1 and 2]. We show that their marginal vector fields are gradient fields. To ensure the existence of all integrals, we assume that $\pi(\mathbf{x})$ is decreasing to zero at a sufficient speed as $\|\mathbf{x}\| \to \infty$ and $v_t$ is bounded. From now on, we denote $\nabla_{\mathbf{x}}$ by $\nabla$.

We first show a useful lemma:

**Lemma B.10.** *If $q_t$ is a marginal probability path, $q_t(\cdot|\mathbf{x}_1)$ is the corresponding conditional probability path, and $\pi(\mathbf{x}_1)$ is the distribution of $\mathbf{x}_1$, then the following holds:*

$$1 = \int \frac{q_t(\mathbf{x}|\mathbf{x}_1)\pi(\mathbf{x}_1)}{q_t(\mathbf{x})}\,d\mathbf{x}_1. \tag{B.17}$$

$$\nabla \log q_t(\mathbf{x}) = \int \nabla \log q_t(\mathbf{x}|\mathbf{x}_1)\frac{q_t(\mathbf{x}|\mathbf{x}_1)\pi(\mathbf{x}_1)}{q_t(\mathbf{x})}\,d\mathbf{x}_1, \tag{B.18}$$

*Proof.* (B.17) directly follows from (B.15).

For (B.18), by noting that $\pi(\mathbf{x}_1)$ is independent of $\mathbf{x}$ in (B.15), we have:

$$\nabla q_t(\mathbf{x}) = \int \nabla q_t(\mathbf{x}|\mathbf{x}_1)\pi(\mathbf{x}_1)\,d\mathbf{x}_1. \tag{B.19}$$

Then, from $\nabla q_t(\mathbf{x}|\mathbf{x}_1) = \nabla \log q_t(\mathbf{x}|\mathbf{x}_1) \cdot q_t(\mathbf{x}|\mathbf{x}_1)$, we have:

$$\nabla \log q_t(\mathbf{x}) = \frac{\nabla q_t(\mathbf{x})}{q_t(\mathbf{x})} = \int \nabla \log q_t(\mathbf{x}|\mathbf{x}_1)\frac{q_t(\mathbf{x}|\mathbf{x}_1)\pi(\mathbf{x}_1)}{q_t(\mathbf{x})}\,d\mathbf{x}_1. \tag{B.20}$$

This completes the proof. $\qquad\square$

We now show the main results:

**Proposition B.11.** *Consider the optimal transport (OT) conditional probability path and vector field from [60, Section 4; Example 2 and Theorem 3]:*

$$q_t(\mathbf{x}|\mathbf{x}_1) = \mathcal{N}(\mathbf{x}|\mu_t(\mathbf{x}_1), \sigma_t^2(\mathbf{x}_1)I) = \mathcal{N}(\mathbf{x}|t\mathbf{x}_1, (1 - (1 - \sigma_{\min})t)^2 I), \tag{B.21}$$

$$u_t(\mathbf{x}|\mathbf{x}_1) = \frac{\mathbf{x}_1 - (1 - \sigma_{\min})\mathbf{x}}{1 - (1 - \sigma_{\min})t}. \tag{B.22}$$

*Then the marginal vector field $u_t(\mathbf{x})$ is a gradient field.*

*Proof.* By the definition of the Gaussian distribution,

$$q_t(\mathbf{x}|\mathbf{x}_1) = \mathcal{N}(\mathbf{x}|t\mathbf{x}_1, (1 - (1 - \sigma_{\min})t)^2 I) = \exp\left(-\frac{\|\mathbf{x} - t\mathbf{x}_1\|_2^2}{(1 - (1 - \sigma_{\min})t)^2}\right)/Z, \tag{B.23}$$

where $Z$ is a normalizing constant that does not depend on $\mathbf{x}$.

We separate $u_t(\mathbf{x})$ into two gradient fields: the score function of $q_t(\mathbf{x}|\mathbf{x}_1)$ and a linear map.

Let us write $\sigma_t := 1 - (1 - \sigma_{\min})t$. The score function of $q_t(\mathbf{x}|\mathbf{x}_1)$ is

$$\nabla \log q_t(\mathbf{x}|\mathbf{x}_1) = -\frac{\mathbf{x} - t\mathbf{x}_1}{(1 - (1 - \sigma_{\min})t)^2} = -\frac{\mathbf{x} - t\mathbf{x}_1}{\sigma_t^2}. \tag{B.24}$$

Next, remember that $\sigma_t = 1 - (1 - \sigma_{\min})t$ is independent of $\mathbf{x}, \mathbf{x}_1$ and note that

$$
\begin{aligned}
u_t(\mathbf{x}|\mathbf{x}_1) &= \frac{\mathbf{x}_1 - (1 - \sigma_{\min})\mathbf{x}}{1 - (1 - \sigma_{\min})t} = \frac{\mathbf{x}_1 - \frac{(1-\sigma_t)}{t}\mathbf{x}}{\sigma_t} = \frac{t\mathbf{x}_1 - (1 - \sigma_t)\mathbf{x}}{t\sigma_t} \\
&= \frac{\mathbf{x}}{t} - \frac{\mathbf{x} - t\mathbf{x}_1}{t\sigma_t} = \frac{\mathbf{x}}{t} - \frac{\sigma_t}{t} \cdot \frac{\mathbf{x} - t\mathbf{x}_1}{\sigma_t^2} \\
&= \frac{\mathbf{x}}{t} + \frac{\sigma_t}{t} \cdot \nabla \log q_t(\mathbf{x}|\mathbf{x}_1).
\end{aligned}
\tag{B.25}
$$

Let $a := \frac{\sigma_t}{t}$. Then $a$ is independent of $\mathbf{x}, \mathbf{x}_1$ and $u_t(\mathbf{x}|\mathbf{x}_1) = \frac{\mathbf{x}}{t} + a\nabla \log q_t(\mathbf{x}|\mathbf{x}_1)$.

By the definition of marginal vector field, we get

$$
\begin{aligned}
u_t(\mathbf{x}) &= \int u_t(\mathbf{x}|\mathbf{x}_1) \frac{q_t(\mathbf{x}|\mathbf{x}_1)\pi(\mathbf{x}_1)}{q_t(\mathbf{x})} d\mathbf{x}_1 \\
&= \int \left( \frac{\mathbf{x}}{t} + a\nabla \log q_t(\mathbf{x}|\mathbf{x}_1) \right) \frac{q_t(\mathbf{x}|\mathbf{x}_1)\pi(\mathbf{x}_1)}{q_t(\mathbf{x})} d\mathbf{x}_1 \\
&= \frac{\mathbf{x}}{t} \int \frac{q_t(\mathbf{x}|\mathbf{x}_1)\pi(\mathbf{x}_1)}{q_t(\mathbf{x})} d\mathbf{x}_1 + a \int \nabla \log q_t(\mathbf{x}|\mathbf{x}_1) \frac{q_t(\mathbf{x}|\mathbf{x}_1)\pi(\mathbf{x}_1)}{q_t(\mathbf{x})} d\mathbf{x}_1 \\
&= \frac{\mathbf{x}}{t} + a\nabla \log q_t(\mathbf{x}),
\end{aligned}
\tag{B.26}
$$

where the last equality holds by Lemma B.10.

Here, $\mathbf{x} \mapsto \frac{\mathbf{x}}{t}$ is a linear map on $\mathbf{x}$ so it is a gradient field, and $\mathbf{x} \mapsto a\nabla \log q_t(\mathbf{x})$ is by definition a gradient field. Since $u_t(\mathbf{x})$ is an addition of two gradient fields, it is clearly a gradient field. This completes the proof. $\qquad\square$

**Proposition B.12.** *The marginal vector fields of the variance-exploding (VE) and variance-preserving (VP) diffusion probability paths in [60, Section 4; Example 1] are gradient fields.*

*Proof.* The proof is based on a stochastic differential equations (SDE) formulation [60, Appendix D]. Consider an SDE of the standard form, defined in an interval $t \in [0, 1]$:

$$
d\mathbf{y} = f_t dt + g_t d\mathbf{w},
\tag{B.27}
$$

with drift $h_t$, diffusion coefficient $g_t$, and standard Wiener process $d\mathbf{w}$.

The corresponding marginal vector field is given as follows [60, Appendix D]:

$$
\tilde{u}_t(\mathbf{x}) = f_t(\mathbf{x}) - \frac{1}{2}g_t^2 \nabla_{\mathbf{x}} \log q_t(\mathbf{x}).
\tag{B.28}
$$

First, it is known that the SDE for the VE path is

$$
d\mathbf{y} = \sqrt{\frac{d}{dt}\sigma_t^2} d\mathbf{w},
\tag{B.29}
$$

where $\sigma_0 = 0$ and $\sigma_t \to \infty$ as $t \to 1$. This SDE moves from data at $t = 0$ to noise at $t = 1$.

Thus the SDE coefficients are $f_t(\mathbf{x}) = 0, g_t(\mathbf{x}) = \sqrt{\frac{d}{dt}\sigma_t^2}$, and the following vector field $\tilde{v}_t(\mathbf{x})$ is

$$
\tilde{u}_t(\mathbf{x}) = -\frac{1}{2}\frac{d}{dt}\sigma_t^2 \nabla \log q_t(\mathbf{x}),
\tag{B.30}
$$

and the vector field from noise at $t = 0$ to data at $t = 1$ is

$$
u_t(\mathbf{x}) = -\frac{1}{2}\frac{d}{dt}\sigma_{1-t}^2 \nabla \log q_{1-t}(\mathbf{x}).
\tag{B.31}
$$

Since $\sigma_{1-t}$ is independent of $\mathbf{x}$, $u_t(\mathbf{x})$ is clearly a gradient field.

Next, it is known that the SDE for the VP path is

$$
d\mathbf{y} = -\frac{T'(t)}{2}\mathbf{y} + \sqrt{T'(t)}d\mathbf{w},
\tag{B.32}
$$

where $T(t) = \int_0^t \beta(s)ds$ and $\beta$ is noise scale. This SDE moves from data at $t = 0$ to noise at $t = 1$. Thus the SDE coefficients are $f_t(\mathbf{x}) = -\frac{T'(t)}{2}\mathbf{x}, g_t(\mathbf{x}) = \sqrt{T'(t)}$, and the vector field $\tilde{u}_t(\mathbf{x})$ is

$$\tilde{u}_t(\mathbf{x}) = -\frac{T'(t)}{2}\mathbf{x} - \frac{1}{2}T'(t)\nabla \log q_t(\mathbf{x}), \tag{B.33}$$

and the vector field from noise at $t = 0$ to data at $t = 1$ is

$$u_t(\mathbf{x}) = -\frac{T'(1-t)}{2}\mathbf{x} - \frac{1}{2}T'(1-t)\nabla \log q_{1-t}(\mathbf{x}). \tag{B.34}$$

Here, $\mathbf{x} \mapsto -\frac{T'(1-t)}{2}\mathbf{x}$ is linear map on $\mathbf{x}$ so it is gradient field, and $-\frac{1}{2}T'(1-t)$ is independent of $\mathbf{x}$ so $\mathbf{x} \mapsto -\frac{1}{2}T'(1-t)\nabla \log q_{1-t}(\mathbf{x})$ is a gradient field. Since $u_t(\mathbf{x})$ is an addition of two gradient fields, it is clearly a gradient field. This completes the proof. $\square$

## C  Experiment Details

For all experiments, we implement spectral mean flows as in Section 3.3 in PyTorch [78]. We use 32 bit-precision and use `opt_einsum` package [17] to optimize the tensor network contractions. For flow matching training, we use the OT flow and path (Proposition B.11) with $\sigma_{\min} = 1e - 5$. Sampling is done with a `midpoint` ODE solver from `flow_matching` package [61]. Each experiment is done with a single NVIDIA RTX A6000 GPU with 48GB and Intel Xeon Gold 6330 CPU @ 2.00GHz.

### C.1  Implementation Details (Sections 3.3 and 5)

We provide the implementation details. Recall that our model is parameterized by a shared feature extractor $\mathrm{MLP}(\cdot, \cdot, t) : \mathcal{X} \times \mathbb{R}^{d_h} \to \mathbb{C}^{d_f}$ and parameters $\mathbf{o}, \mathbf{s} \in \mathbb{R}^{d_h}, \lambda \in \mathbb{C}^r, \mathbf{B}, \mathbf{L}, \mathbf{R}, \mathbf{H} \in \mathbb{C}^{d_f \times r}$.

We denote trainable linear layer from $\mathcal{X}$ to $\mathcal{Y}$ by $\mathrm{Lin}_{\mathcal{X} \to \mathcal{Y}}$, and denote $\mathrm{Lin}_{\mathcal{X}} := \mathrm{Lin}_{\mathcal{X} \to \mathcal{X}}$. We denote squared ReLU activation $\mathbf{x} \mapsto \mathrm{ReLU}(\mathbf{x})^2$ [87] by $\mathrm{ReLU}^2$, RMS normalization [103] by RMSNorm, L2 normalization by L2Norm, and $d$-dimensional sinusoidal positional encoding [96] by $\mathrm{PE}_{\mathbb{R}^d}$.

For given number of layers $L$ and hidden dimensions $d_h, d'_h$, the feature extractor $\mathrm{MLP}(\cdot, \cdot, t) : \mathcal{X} \times \mathbb{R}^{d_h} \to \mathbb{C}^{d_f}$ is designed as follows. We obtain conditioning feature of $t$ using sinusoidal PE:

$$\mathbf{c}_t = \mathrm{Lin}_{\mathbb{R}^{d_h}} \circ \mathrm{ReLU}^2 \circ \mathrm{Lin}_{\mathbb{R}^{d_h}} \circ \mathrm{ReLU}^2 \circ \mathrm{Lin}_{\mathbb{R}^{256} \to \mathbb{R}^{d_h}} \circ \mathrm{PE}_{\mathbb{R}^{256}}(t). \tag{C.1}$$

Then, the full feature extraction process for a given input $\mathbf{x} \in \mathcal{X}$ is written as follows:

$$\mathbf{z}^{(0)} = \mathrm{Lin}_{\mathcal{X} \to \mathbb{R}^{d_h}}(\mathbf{x}) + \mathbf{q} + \mathbf{c}_t, \tag{C.2}$$

$$\mathbf{z}^{(l)} = \mathbf{z}^{(l-1)} + \mathrm{Lin}_{\mathbb{R}^{d'_h} \to \mathbb{R}^{d_h}} \circ \mathrm{ReLU}^2 \circ \mathrm{Lin}_{\mathbb{R}^{d_h} \to \mathbb{R}^{d'_h}} \circ \mathrm{RMSNorm}(\mathbf{z}^{(l-1)}), \tag{C.3}$$

$$\mathbf{h}^{(l)} = \mathrm{L2Norm} \circ \mathrm{ReLU}^2 \circ \mathrm{Lin}_{\mathbb{R}^{d_h} \to \mathbb{C}^{d_h}} \circ \mathrm{ReLU}^2(\mathbf{z}^{(l)}), \tag{C.4}$$

$$\mathrm{MLP}(\mathbf{x}, \mathbf{q}, t) = \mathrm{concat}(\mathbf{h}^{(1)}, ..., \mathbf{h}^{(L)}), \quad \mathbf{q} \in \{\mathbf{o}, \mathbf{s}\}. \tag{C.5}$$

In (C.4), we apply $\mathrm{ReLU}^2$ to complex features by separately operating on real and imaginary parts. As shown in (C.5), instead of using only the output of the last layer, the MLP concatenates features from all layers to construct the output, i.e., $d_f = Ld_h$.

Then, to parameterize $\mathbf{B}, \mathbf{L}, \mathbf{R}, \mathbf{H} \in \mathbb{C}^{d_f \times r}$, we set $r = Ld_r$ for some $d_r$ and, to prevent $\mathcal{O}(L^2 d_h d_l)$ parameter complexity, impose the following block-diagonal structure on each of $\mathbf{D} \in \{\mathbf{B}, \mathbf{L}, \mathbf{R}, \mathbf{H}\}$:

$$\mathbf{D} = \mathbf{d}_1 \oplus \cdots \oplus \mathbf{d}_L = \begin{pmatrix} \mathbf{d}_1 & & \\ & \ddots & \\ & & \mathbf{d}_L \end{pmatrix} \in \mathbb{C}^{(Ld_h) \times (Ld_r)}, \quad \mathbf{d}_l \in \mathbb{C}^{d_h \times d_r}. \tag{C.6}$$

In addition to reducing the parameter count by a factor of $L$, this allows space-efficient, layerwise evaluation of tensor network contraction based on $(\mathbf{r}_1 \oplus \cdots \oplus \mathbf{r}_L)(\mathbf{l}_1 \oplus \cdots \oplus \mathbf{l}_L) = (\mathbf{r}_1\mathbf{l}_1 \oplus \cdots \oplus \mathbf{r}_L\mathbf{l}_L)$. In practice, we simply set $d_r = d_h/2$ unless stated otherwise.

Table 6: Hyperparameters for Table 1.

| Hyperparameter | Sines | Stocks | ETTh | MuJoCo | Energy | fMRI |
|---|---|---|---|---|---|---|
| Hidden dimension $d_h, d'_h$ | 64 | 64 | 64 | 96 | 96 | 96 |
| Layers $L$ | 10 | 10 | 10 | 16 | 14 | 16 |
| Batch size | 256 | 128 | 256 | 256 | 128 | 256 |
| Training steps | 12000 | 10000 | 18000 | 28000 | 25000 | 15000 |
| Sampling steps | 500 | 500 | 500 | 1000 | 1000 | 1000 |

Table 7: Wall-clock training times for Table 1.

| Method | Sines | Stocks | ETTh | MuJoCo | Energy | fMRI |
|---|---|---|---|---|---|---|
| Ours | 18min | 15min | 27min | 60min | 44min | 35min |
| Diffusion-TS | 16min | 15min | 30min | 25min | 60min | 48min |

Table 8: Wall-clock sampling times per 1,000 samples for Table 1.

| Method | Sines | Stocks | ETTh | MuJoCo | Energy | fMRI |
|---|---|---|---|---|---|---|
| Ours | 30sec | 30sec | 29sec | 109sec | 95sec | 109sec |
| Diffusion-TS | 66sec | 66sec | 67sec | 133sec | 202sec | 262sec |

Table 9: Parameter counts for Table 1, extending [101, Table 7].

| Method | Sines | Stocks | Energy |
|---|---|---|---|
| Ours | 356,736 | 356,800 | 1,086,432 |
| Diffusion-TS | 232,177 | 291,318 | 1,135,144 |
| Diffwave | 533,592 | 599,448 | 1,337,752 |
| Cot-GAN | 40,133 | 52,675 | 601,539 |
| TimeGAN | 34,026 | 48,775 | 1,043,179 |
| TimeVAE | 97,525 | 104,412 | 677,418 |

The eigenvalues $\lambda$ and nonzero blocks of $\mathbf{B}, \mathbf{L}, \mathbf{R}, \mathbf{H}$ use exponential parameterization of [75], which initializes each entry $z$ uniformly between rings of radius $r_{\min}$ and $r_{\max}$ on the complex plane:

$$z = \exp(-\nu + i\theta) \quad \nu = -\frac{1}{2}\log(u_1(r_{\max}^2 - r_{\min}^2) + r_{\min}^2) \quad \theta = 2\pi u_2 \quad u_1, u_2 \sim \mathcal{U}[0,1] \quad \text{(C.7)}$$

When initializing $z$ on the complex unit disk, we find it necessary to set $r_{\min} = \epsilon$ and $r_{\max} = 1 - \epsilon$ for a small constant $\epsilon = 10^{-12}$, to avoid numerical issues related to logarithm.

After initializing $\lambda, \mathbf{B}, \mathbf{L}, \mathbf{R}, \mathbf{H}$, we apply proper scaling to prevent the activations from exploding or vanishing. We find that scaling $\lambda$ by $\sqrt{2}$ and the rest by $\sqrt{2}d_r^{-1/4}$ provides the desired stability.

### C.2 Synthetic Experiments (Section 5.1)

We provide details of the 2D checkerboard experiment. We use $\text{MLP}(\cdot, \cdot, t) : \mathbb{R}^1 \times \mathbb{R}^h \to \mathbb{C}^{Ld_h}$ with $L = 4$ layers and hidden dimensions $d_h = d'_h = 128$. For training, we use Adam optimizer [22] with hyperparameters $(\beta_1, \beta_2) = (0.9, 0.999)$. The model is trained for 20k iterations with learning rate 1e-3 and batch size 10,000, with 1k steps of linear learning rate warmup, and gradient norm clipping at 1.0. For sampling, we keep track of exponential moving average (EMA) of the model parameters during training [50], updating every 10 iterations with decay rate 0.995.

### C.3 Time-Series Modeling (Section 5.2)

We provide details of unconditional time-series generation experiments in Section 5.2.

**Regular time series** We first detail the datasets in Table 1. Sines has 10,000 5-channel sequences where each channel is a sinusoidal signal $x(n) = \sin(2\pi\omega n + \theta)$ with independently determined

Table 10: Parameter counts for Table 2.

| Method | Sines | Stocks | MuJoCo |
|---|---|---|---|
| Ours | 32,161k | 32,161k | 32,165k |
| SDFormer-AR | 44,971k | 44,971k | 44,971k |
| ImagenTime | 57,560k | 57,562k | 14,430k |

Table 11: Parameter counts for Table 3.

| Method | FRED-MD | NN5 Daily |
|---|---|---|
| Ours | 100,603k | 100,569k |
| ImagenTime | 151,720k | 151,720k |

Table 12: Parameter counts for Tables 4 and 5.

| Method | Stocks, 0% Drop | Stocks, 30–70% Drop | Pendulum |
|---|---|---|---|
| Ours | 157,888 | 157,888 | 45,544 |
| Koopman VAE | 42,270 | 33,410 | 37,978 |

frequency $\omega \sim \mathcal{U}[0, 1]$ and phase $\theta \sim \mathcal{U}[-\pi, \pi]$. Stocks contains 3,773 6-channel sequences from daily Google stock price data from 2004 to 2019. The channels are volume and high, low, opening, closing, and adjusted closing prices. ETTh contains 17,420 7-channel sequences from electricity transformers recorded every 15 minutes from 2016 to 2018, including load and oil temperature. MuJoCo has 10,000 14-channel sequences from DeepMind `dm_control` physics simulation. Energy contains 19,711 28-channel sequences from appliances energy use in a building recorded at a noisy 10-minutes interval, including house temperature and humidity. fMRI contains 10,000 50-channel sequences from a realistic simulation of blood oxygen level dependent signal in functional MRI.

We closely follow the protocol of [101] for hyperparameter selection and training. Detailed hyperparameters are in Table 6. The ranges considered for $\mathrm{MLP}(\cdot, \cdot, t) : \mathbb{R}^d \times \mathbb{R}^{d_h} \to \mathbb{C}^{Ld_h}$ are batch size $\{128, 256\}$, layers $L \in \{10, 14, 16\}$, and hidden dimension $d_h = d'_h \in \{64, 96\}$. Other choices mostly follow [101]: For training, we use Adam optimizer [22] with $(\beta_1, \beta_2) = (0.9, 0.96)$. The model is trained for the same iterations per dataset as in [101] with learning rate 8e-4 and gradient clipping at 1.0, with 500 steps of linear warmup and then decaying by 0.5 on plateau with patience 2,000. An exception is MuJoCo, where longer training is necessary for convergence. For sampling, we use EMA of model parameters [50] following [101], updating every 10 iterations with decay rate 0.995. We use a `midpoint` ODE solver with the same sampling steps as in [101]. In Table 1, we adopt baseline scores from [100, 101] except for Diffusion-TS reproduced with the official implementation.

We provide supplementary results of Table 1. From Figure 4 to 9, we visualize real and generated data from our model in Table 1, showing that generations are visually realistic. We further perform detailed complexity analysis by measuring wall-clock training and sampling times as well as parameter counts. The results are in Tables 7, 8, and 9, showing that our models for Table 1 have reasonable computational complexity, and their generation is faster than the state-of-the-art Diffusion-TS.

We now provide the details of additional comparisons in Table 2. For this setup, we consider two recent state-of-the-art methods SDFormer [15] and ImagenTime [71] that operate in a larger parameter regime, roughly using 11–43× the parameters of the largest model Diffwave in Table 1. We use Sines, Stocks, and MuJoCo datasets, considering both resource constraints and reproducible datasets in the codebases of [15, 71]. We test larger spectral mean flows with comparable number of parameters to the baselines without other architectural changes. Specifically, we use $\mathrm{MLP}(\cdot, \cdot, t) : \mathbb{R}^d \times \mathbb{R}^{d_h} \to \mathbb{C}^{Ld_h}$ with $L = 12$ layers and dimensions $d_h = 512$, $d'_h = 1024$. The parameter counts are in Table 10. Other hyperparameters are chosen as follows. Following [15], we train our models using AdamW optimizer [63] with $(\beta_1, \beta_2) = (0.5, 0.9)$ and weight decay 1e-6 for 100k iterations and measure the discriminative score with respect to the training dataset every 2,500 iterations for validation. Without hyperparameter tuning, we use batch size 512, learning rate 1e-4, and gradient clipping at 1.0. For sampling, we use EMA of model parameters, updating every 10 iterations with decay rate 0.995. We use a `midpoint` ODE solver with 100 sampling steps for validation and 500 steps for testing. In Table 2, we reproduce SDFormer-AR and ImagenTime using the official implementations.

**Long time series** We provide details of long time-series modeling in Table 3. We use two datasets from the Monash repository [32]. FRED-MD has 107 time series of length 728 from macro-economic indicators from the Federal Reserve Bank. NN5 Daily contains 111 time series of length 728 from daily cash withdrawals from ATMs in UK. We choose these datasets as their moderate sizes allow experimentation within resource constraint. Following [71], we normalize each trajectory to adhere to a zero-centered normal distribution, and use 80% of the data for training and 20% for testing.

We design our architectures as in Appendix C.1, and equip them with time-delay observables used in operator-theoretic methods for dynamical systems [49, 95] which we found to help stabilizing training. Specifically, for time series $\mathbf{x}^{1:N}$ we define a new observable $\bar{\mathbf{x}}^{1:N'}$ as $\bar{\mathbf{x}}^i := \mathrm{concat}(\mathbf{x}^{ai}, ..., \mathbf{x}^{ai+b})$ for delay $a$ and dimension $b$. When $a \leq b$, there is no loss of information and the map $\mathbf{x} \mapsto \bar{\mathbf{x}}$ is invertible. Hence, we model $\bar{\mathbf{x}}$ using spectral mean flow and convert the result to $\mathbf{x}$ using the inverse map. We use $a = 16$ and $b = 64$ for FRED-MD, and $a = 20$ and $b = 20$ for NN5 Daily.

We include state-of-the-art ImagenTime [71] for comparison, and test spectral mean flow with a comparable number of parameters. Specifically, we use $\mathrm{MLP}(\cdot, \cdot, t) : \mathbb{R}^d \times \mathbb{R}^{d_h} \to \mathbb{C}^{L d_h}$ with $L = 12$ layers and hidden dimensions $d_h = 768$, $d'_h = 3072$. The parameter counts are in Table 11. Other hyperparameters are chosen as follows. Following [71], we train our models for 1k epochs with batch size 32 and weight decay 1e-5, measure the marginal score every 100 iterations for validation, and for sampling use EMA of model parameters with 100 warmup steps and decay rate 0.9999. We use AdamW optimizer with $(\beta_1, \beta_2) = (0.9, 0.95)$ and learning rate 3e-4 for all parameters except for $(\lambda, \mathbf{B}, \mathbf{L}, \mathbf{R}, \mathbf{H})$. For these spectral parameters, we use Muon optimizer [46] with learning rate 1e-3, which slightly improved the performances. While we leave a deeper investigation as future work, we conjecture this is because Muon is a second-order optimizer, which are expected to improve the learning of multiplicative matrix states in general [92, Section 3.3]. We use gradient clipping at 1.0. For sampling, we use a `midpoint` ODE solver with 100 sampling steps. In Table 3, we adopt the baseline scores from [71] except for ImagenTime reproduced using the official implementation.

**Irregular time series**  We provide details of irregular time-series modeling in Table 4. We recall that we consider two tasks, **(1) irregular → regular**: generating the full time series including the missing timesteps, and **(2) irregular → irregular**: generating only the irregularly sampled time series. We apply spectral mean flow to the tasks as follows. For **(1)**, we handle missing timesteps during training by simply interpolating the observations, which we find to work well in practice. For **(2)**, based on the HMM formulation we use a classical approach [105] of treating the sampling interval itself as a part of data. That is, we consider hidden states $Z^{1:N}$ under dynamics $\mathbb{P}[Z^{n+1}|Z^n]$, and a local observation model $\mathbb{P}[X^n, \Delta\tau^n|Z^n]$ that jointly determines the observation $X^n$ and sampling interval $\Delta\tau^n$ between $X^{n-1}$ and $X^n$. If expressive enough, this HMM can express any joint distribution $\mathbb{P}[(X^1, \Delta\tau^1), ..., (X^N, \Delta\tau^N)]$, i.e., general time series under irregular or informative sampling. The only change in implementation is using sampling intervals $\Delta\tau^n$ as an additional data dimension.

Detailed hyperparameters are as follows. We use $\mathrm{MLP}(\cdot, \cdot, t) : \mathbb{R}^d \times \mathbb{R}^{d_h} \to \mathbb{C}^{L d_h}$ with $L = 4$ layers and hidden dimension $d_h = d'_h = 64$. This leads to a smaller model than the previous experiments, although larger than Koopman VAE (Table 12). For other hyperparameters, we closely follow [72] and train all models using Adam optimizer with learning rate 7e-4 for 600 epochs with batch size 64. For sampling, we use EMA of model parameters, updating every 10 iterations with decay rate 0.995, and use a `midpoint` ODE solver with 100 sampling steps. In Table 4, the baseline scores are from [45, 72] except for Koopman VAE tested using the official implementation. To run Koopman VAE on task **(2)**, we adopt the hyperparameters provided in the official implementation for task **(1)**.

**Physics-informed modeling**  Lastly, we provide details of physics-informed modeling in Table 5. As discussed, we consider a nonlinear pendulum governed by $\ddot{\theta} + 9.8 \sin\theta = 0$, $\dot{\theta}(0) = 0$. To obtain a dataset, we simulate 2,000 independent trajectories with initial condition $\theta(0) \sim \mathrm{Unif}[0.5, 2.7]$ over the time interval $[0, 10]$ and sampling interval $0.25$, treating the pair $(\theta(t), \dot{\theta}(t))$ as the observation. Following [72], we add Gaussian noise with standard deviation 0.08 to simulate real-world noise and standardize each trajectory to the range $[0, 1]$. We consider Koopman VAE [72] for comparison, as it is a state-of-the-art operator-based method for time series. We use correlational score for evaluation.

To focus on the impact of spectral regularization on the matrix states described in Section 5.2, in the feature extractor we only use the last layer feature $\mathbf{h}^{(L)}$ (C.5), leading to $\mathrm{MLP}(\cdot, \cdot, t) : \mathbb{R}^d \times \mathbb{R}^{d_h} \to \mathbb{C}^{d_h}$, $\lambda \in \mathbb{C}^{d_r}$, and $\mathbf{B}, \mathbf{L}, \mathbf{R}, \mathbf{H} \in \mathbb{C}^{d_h \times d_r}$ without block-diagonal structure. We use $L = 8$ layers and dimension $d_h = d'_h = 32$. Following [72], we fix the spectral dimension as $r = d_r = 4$, as well as the latent dimension of Koopman VAE as 4. This leads to similarly sized models (Table 12) with spectral regularization on $4 \times 4$ matrices (if used). We train all models with Adam optimizer, learning rate 7e-4 for 600 epochs, with batch size 64. For sampling of our model, we use EMA of parameters updating every 10 steps with decay 0.995, and use a `midpoint` ODE solver with 100 sampling steps.

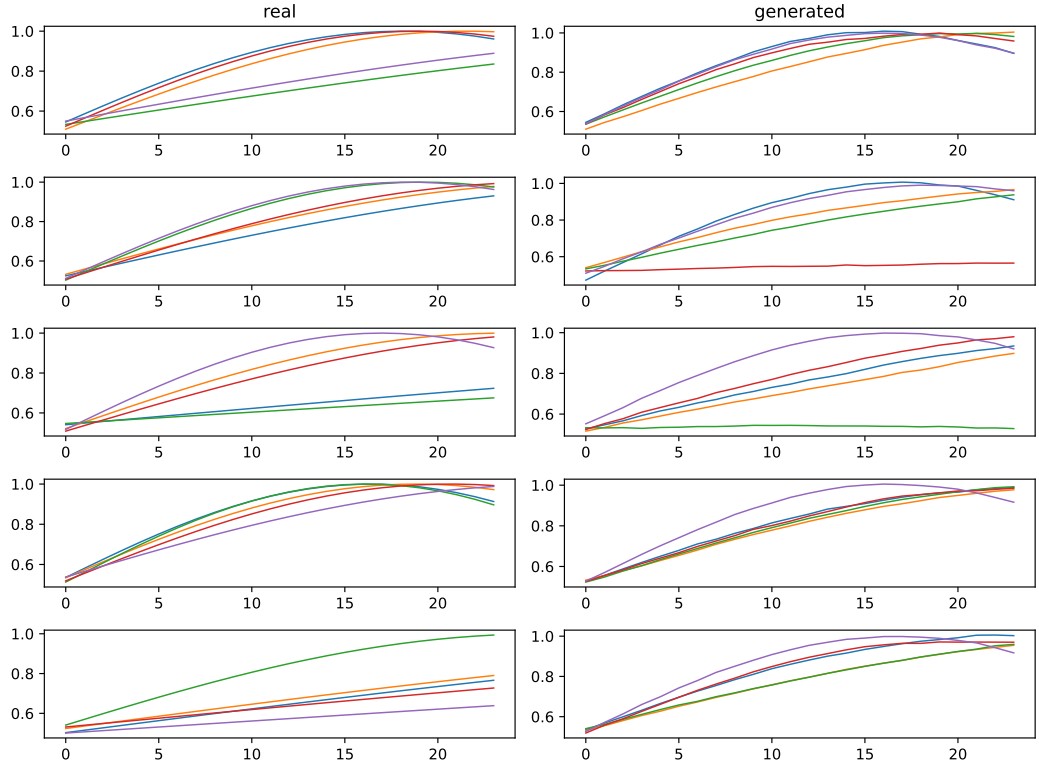

Figure 4: Examples of real and generated data for the Sines dataset.

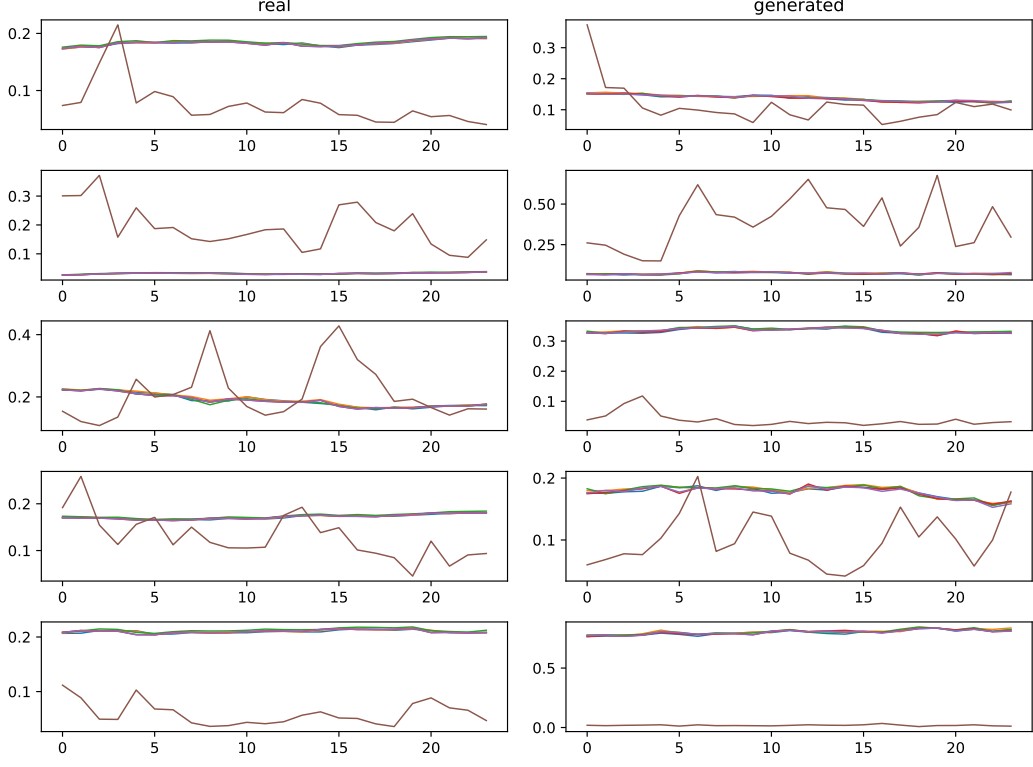

Figure 5: Examples of real and generated data for the Stocks dataset.

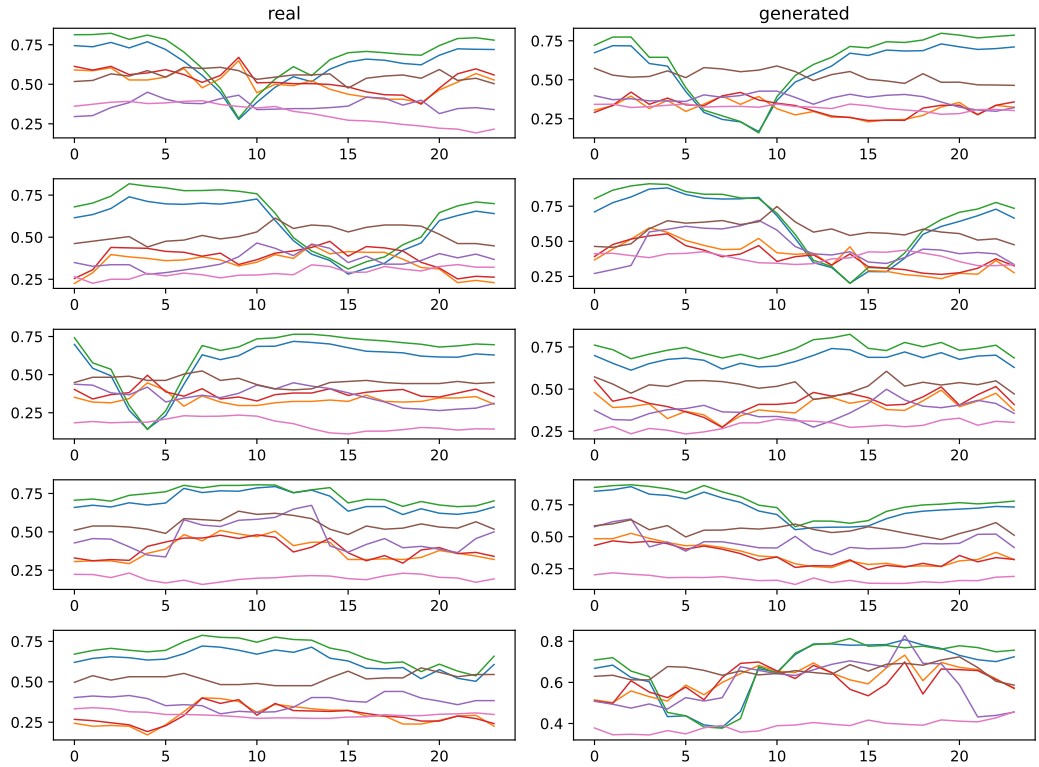

Figure 6: Examples of real and generated data for the ETTh dataset.

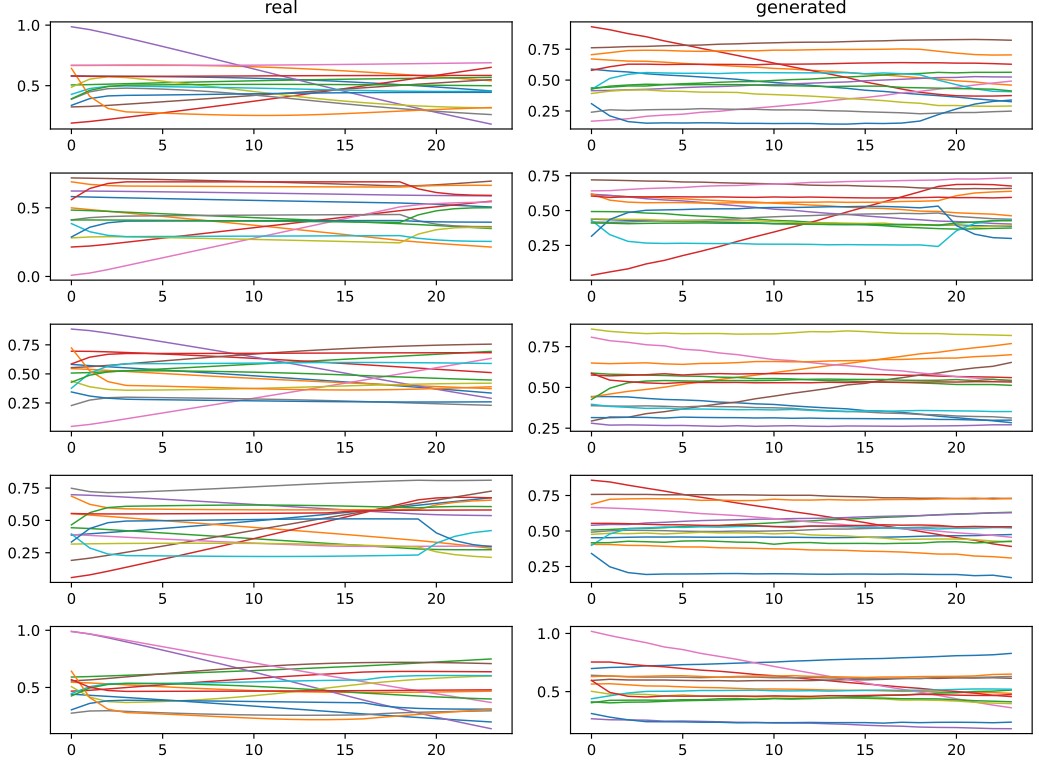

Figure 7: Examples of real and generated data for the MuJoCo dataset.

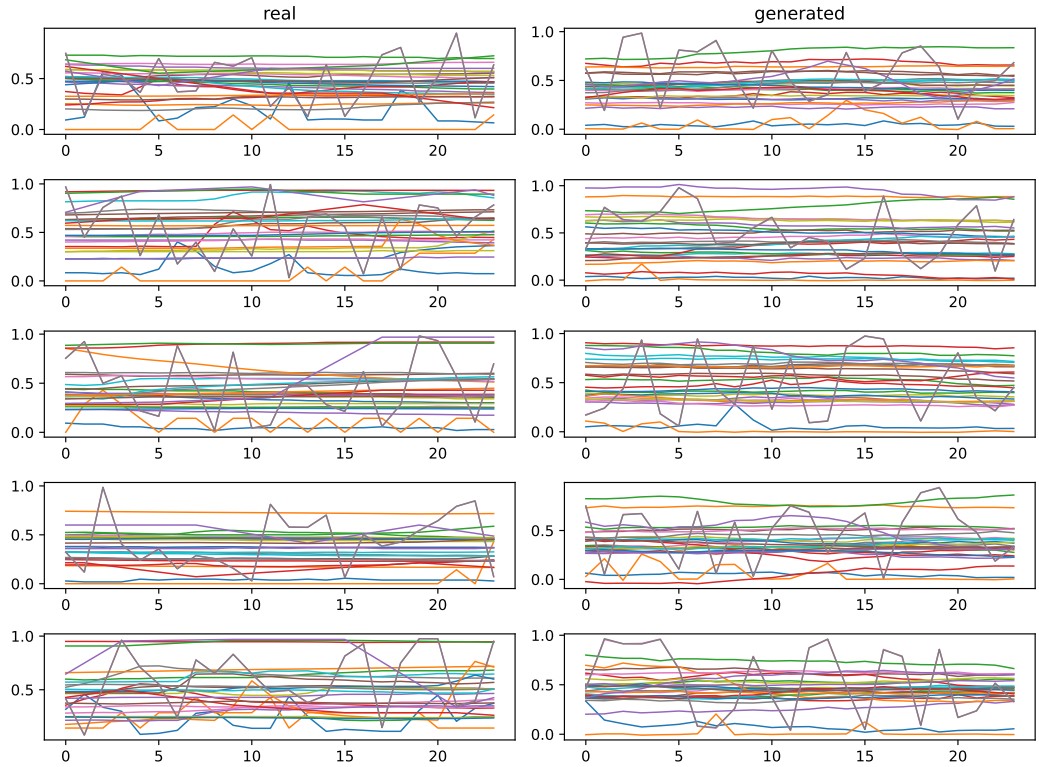

Figure 8: Examples of real and generated data for the Energy dataset.

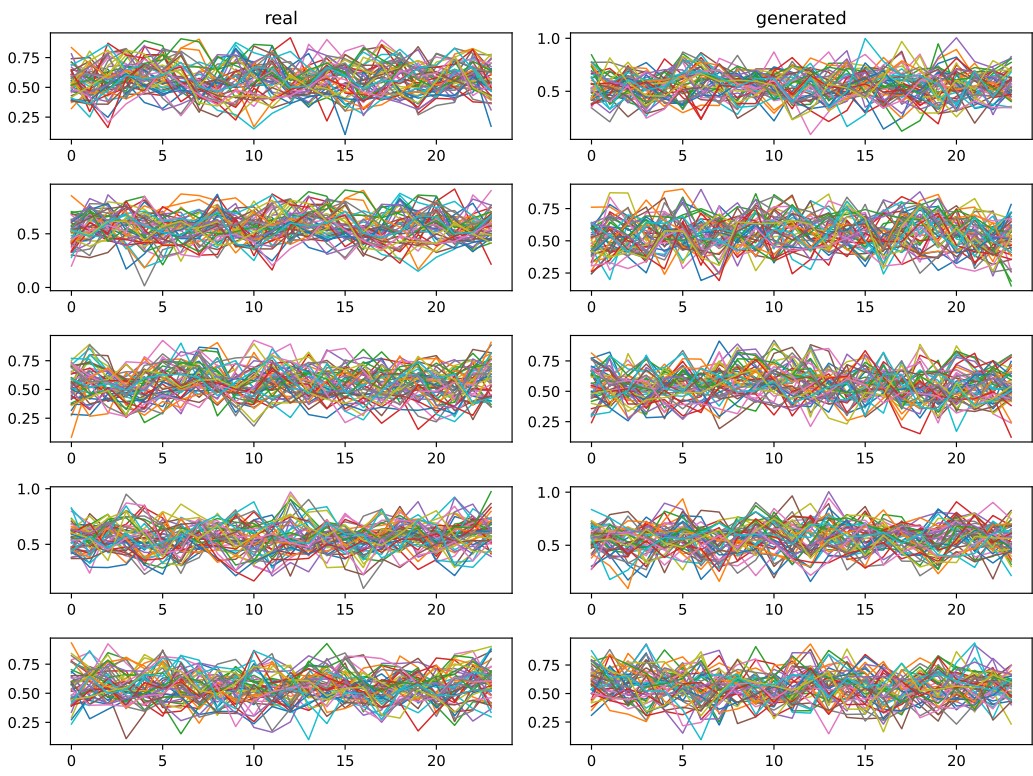

Figure 9: Examples of real and generated data for the fMRI dataset.

# D   Limitations and Future Work

Our current main limitation is the question of how to condition the generative process on partial observations in a principled manner. We believe studying kernel Bayesian inference [29] and Laplace transform of CME operators [56] in our problem context could be fruitful. The reasoning is as follows. Suppose we have a partial observation $\mathbf{x}^{1:n}$ and want to generate $\mathbf{x}^{n+1:N} \sim \mathbb{P}[X^{n+1:N}|X^{1:n} = \mathbf{x}^{1:n}]$. This can be done with MMD flow if we have conditional mean embedding $\mu_{X^{n+1:N}|X^{1:n}=\mathbf{x}^{1:n}}$. In kernel methods for HMMs, this is precisely estimated via kernel Bayesian inference [29, 73]. As an alternative, a type of Laplace transform of Koopman operator has been recently used for kernel-based forecasting $\mathbf{x}^{1:n} \mapsto \mathbf{x}^{n+1:N}$ of deterministic nonlinear dynamics [7]. Given the proximity between the two operators, we conjecture its extension to our setup may enable principled conditioning.

In addition, while in Section 5.2 we showed one possible approach to perform physics-informed modeling using our method via spectral regularization, a more direct approach of identifying the operator eigenfunctions from the learned model would strengthen the interpretability of our method.

Also, while we adopted simple strategies to handle irregularly or informatively sampled dynamical systems in Section 5.2, another possibly valid approach is using the continuous-time CME operator $U^{\Delta\tau}\varphi(\mathbf{z}) = \mathbb{E}[\varphi(Z^{\tau+\Delta\tau})|Z^\tau = \mathbf{z}]$ and its spectral decomposition $U = \sum_i \exp(s_i\Delta\tau)h_i \otimes g_i$ with continuous-time eigenvalues $s_i$ [3]. We leave further investigation of this direction as future work.

Lastly, room for future work lies in engineering aspect of the method, including better architectural components and stabilization strategies for long sequences, choices of vector fields for flow matching training, as well as empirical investigations of scaling laws.

While the above directions are exciting, we believe their proper investigation warrants separate work. Thus, we focused our effort on laying the foundations of a neural method based on operator theory.

