# OpenReview forum: "Sequence Modeling with Spectral Mean Flows"
_NeurIPS.cc/2025/Conference — NeurIPS 2025 poster_

### Official Review · Reviewer_j6aF · 2025-06-29

**Clarity:** 2
**Significance:** 2
**Originality:** 3
**Rating:** 4
**Confidence:** 4

**Summary:**

The paper proposes Spectral Mean Flow (SMF), a new sequence modeling framework that combines an operator‑theoretic view of Hidden Markov Models, a tensor‑network decomposition of the sequence mean embedding derived from the spectral decomposition of the underlying operator, and a time‑dependent Maximum Mean Discrepancy (MMD) gradient flow that is trained with the Flow‑Matching objective. The resulting model is fully differentiable, avoids simulation during training, and samples by solving an ODE. Empirically, SMF attains competitive scores on six multivariate time‑series data sets compared to diffusion, VAE, and GAN baselines.

**Questions:**

See Weaknesses

**Ethical Concerns:**

["NO or VERY MINOR ethics concerns only"]

**Final Justification:**

I appreciate the authors' response that address my primary concerns. In light of their thorough rebuttal, I have raised my score to 4.

While the reason for omitting the ImagenTime results on the Cont.-FID metric remains unclear, I have two recommendations for the final version. The authors should incorporate the work on  SDformer[1], ImagenTime [2], and KoVAE [3] into the related work section and discuss key difference with Koopman approach in KoVAE. Furthermore, the new results, which are currently only in the rebuttal, should be moved into the main paper. Their inclusion is important as they effectively position the authors' method as being on par with or competitive against other state-of-the-art techniques.

References

[1] "SDformer: Similarity-driven discrete transformer for time series generation." NeurIPS 2024

[2] "Utilizing image transforms and diffusion models for generative modeling of short and long time series." NeurIPS 2024

[3] "Generative modeling of regular and irregular time series data via Koopman VAEs." ICLR 2024.

**Limitations:**

See Weaknesses

**Quality:**

2

**Strengths And Weaknesses:**

**Strengths:**

* The paper unifies three previously disjoint ideas (operator theory, tensor networks, flow matching) into a coherent generative framework. In particular, the extension of MMD flows to time‑dependent RKHSs (Eq. 3.15) and its connection to flow matching appear original.
* The authors discuss complex‑valued parameterization, low‑rank sharing, and stability tricks (SwiGLU, RMS‑LN, sharing most of the computations), which will be valuable for practitioners.


**Weaknesses:**

* The paper does not convincingly articulate the practical advantages of this new paradigm over standard flow-matching or diffusion models. Because the operator is not modeled explicitly, potential benefits such as interpretability through mode analysis remain out of reach, and the added complexity may not translate into clear gains.
* All empirical studies are restricted to fixed-length sequences of 24 time steps. The authors do not investigate scalability to longer or variable-length sequences, nor do they discuss any practical limitations in this regard.
* A detailed complexity analysis is missing: the authors do not report wall-clock training or inference times, FLOPs, or parameter counts, nor do they compare these metrics with competing baselines.
* Important recent baselines, such as SDFormer [1], ImagenTime [2], FiDE [3], KoVAE [4] and other recent works, are omitted, making it hard to judge the proposed method’s relative performance.

[1] Chen, Zhicheng, et al. "Sdformer: Similarity-driven discrete transformer for time series generation." NeurIPS 2024

[2] Naiman, Ilan, et al. "Utilizing image transforms and diffusion models for generative modeling of short and long time series." NeurIPS 2024

[3] Galib, Asadullah Hill, Pang-Ning Tan, and Lifeng Luo. "FIDE: Frequency-Inflated Conditional Diffusion Model for Extreme-Aware Time Series Generation." NeurIPS 2024.

[4] Naiman, Ilan, et al. "Generative modeling of regular and irregular time series data via Koopman VAEs." ICLR 2024.

Other related work:

[5] Coletta, Andrea, et al. "On the constrained time-series generation problem." NeurIPS 2023.

[6] Jeon, Jinsung, et al. "GT-GAN: General purpose time series synthesis with generative adversarial networks." NeurIPS 2022

---

> ### Author Rebuttal · Authors · 2025-07-31
>
> We thank the reviewer for the constructive feedback. We address the main concerns and questions below.
>
> (note: we revised our architecture after submission, which led to improvements on high-dimensional Energy and fMRI in Table 1. Please see **Table A** for reviewer k6rf. All new experiments are done with this version. We are happy to elaborate on the details during the discussion period.)
>
> > W1. “All empirical studies are restricted to fixed 24 timesteps. The authors do not investigate scalability to longer or variable-length sequences.” “Important recent baselines are omitted, making it hard to judge the method’s relative performance.”
>
> To address the concerns jointly, we ran additional experiments on more challenging setups: (1) modeling long time series and (2) learning on irregularly sampled time series and then generating full time series, comparing with recent SoTA ImagenTime [1] and Koopman VAE (KoVAE) [2] suggested by the reviewer.
>
> (note: while the reviewer also suggested SDFormer [3] and FIDE [4], we could not find code for [3], and official code of [4] did not contain sufficient information for reproducibility, preventing us from experimenting. Still, we will cite and discuss [1-4] in our next revision and also [5, 6] as suggested by the reviewer.)
>
> For long time series, we used FRED-MD and NN5 Daily with 728- and 791-length data [7]. Their modest sizes allowed testing within the rebuttal period. Our models are designed similarly to **Table A**, equipped with time-delay observables used in operator methods [9]. We use marginal, classification, and prediction metrics [1].
>
> **Table B.** Long time series. Baselines are from [1], except for ImagenTime which was reproduced using official code.
> | | FRED-MD | | | NN5 Daily | | |
> |--|--|--|--|--|--|--|
> | | marg. ↓ | class. ↑ | pred. ↓ | marg. ↓ | class. ↑ | pred. ↓ |
> | Ours | **0.0145** | **1.547** | 0.0261 | **0.00453** | **1.07** | 0.427 |
> | ImagenTime | 0.0292 | 1.154 | 0.0279 | 0.00745 | 0.935 | 0.496 |
> | LS4 | 0.022 | 0.544 | 0.037 | 0.007 | 0.636 | **0.241** |
> | SaShiMi | 0.048 | 0.001 | 0.232 | 0.020 	| 0.045 | 0.849 |
> | SDEGAN | 0.084 | 0.501 | 0.677 | 0.085 | 0.085 | 1.01 |
> | TimeGAN | 0.081 | 0.029 | 0.058 | 0.040 | 0.001 | 1.34 |
> | Latent ODE | 0.042 | 0.327 | **0.013** | 0.107 | 0.000 | 1.04 |
> | ODE2VAE | 0.122 | 0.028 | 0.567 	| 0.211 | 0.001 | 1.19 |
> | GP-VAE | 0.152 | 0.016 	| 2.05 | 0.117 | 0.002 | 1.17 |
> | RNN-VAE | 0.132 | 0.036 | 1.47 | 0.137 | 0.000 | 0.967 |
>
> The results are in **Table B**. Our method shows strong results, achieving the best metric in 4 out of 6 cases and outperforming the previous best (ImagenTime, LS4) in 5 out of 6 cases.
>
> For irregular time series, we used 3 datasets derived from Google stock data by omitting 30%, 50%, and 70% of observations [2]. The task is generating full signal. Our models are designed similarly to **Table A**. To handle missing values at training, a simple idea of interpolating observed values worked well. Based on the code of [2], we use discriminative metric.
>
> **Table C.** Irregular time series. 30%, 50%, and 70% dropped. Baselines are from [2] except for KoVAE which was reproduced using official code.
> | | Stocks, no drop | 30% | 50% | 70% | | |
> |--|--|--|--|--|--|--|
> | disc. ↓ | | | | |
> | Ours | **0.0150** | **0.1109**| **0.0130**| **0.0628**|
> | KoVAE | 0.0203 | 0.1336 | 0.0899| 0.0740|
> | GT-GAN | 0.077| 0.251 | 0.265 | 0.230 |
> | TimeGAN | 0.102 | 0.411 | 0.477 | 0.485 |
> | RCGAN | 0.196 | 0.436 	| 0.478 | 0.381 |
> | C-RNN-GAN | 0.399 | 0.500 	| 0.500 | 0.500 |
> | T-Forcing | 0.226 | 0.305 | 0.308 | 0.365 |
> | P-Forcing | 0.257 | 0.341 | 0.388 | 0.317 |
> | GP-VAE | 0.152 | 0.016 	| 2.05 | 0.117 |
> | RNN-VAE | 0.132 | 0.036 | 1.47 | 0.137 |
>
> The results are in **Table C**. Our method achieves the best metric in all 4 cases.
>
> > W2. “The paper does not convincingly articulate the practical advantages over standard flow-matching or diffusion. Because the operator is not modeled explicitly, potential benefits such as interpretability through mode analysis remain out of reach, and the added complexity may not translate into clear gains.”
>
> While interpretation of eigenfunctions is not direct due to our end-to-end design, we believe that a closely related benefit of operator methods is retained to some degree: incorporation of physics-based prior knowledge. Thanks to linearity of tensor network, this can be done via explicit spectral regularization. We test this using a challenging physics-constrained problem from [2].
>
> We consider a nonlinear pendulum, governed by an ODE of angular displacement $\theta$ from an equilibrium [2]: $\frac{d^2}{dt^2}\theta + 9.8\sin\theta = 0, \dot{\theta}(0)=0$.
> As the pendulum is stable and conservative, the physical knowledge is that eigenvalues of underlying operator have non-positive real part and some have zero real.
>
> Since our model is governed by a linear product of matrix hidden states $Ow(x^n)$ (Eq. 3.12), we can implement this knowledge in an end-to-end manner.
> In KoVAE [2], this stability constraint is built-in using a loss on matrix $A$ for hidden states $z_{t+1} = Az_t$, specifically $|s_1 - 1|^2 + |s_2 - 1|^2$ where $(s_1, s_2)$ are the two largest eigenvalues of $A$. We compare our model against [2] using the same spectral loss for each matrix hidden state’s two largest eigenvalues.
> We use correlational score for eval. [9]
>
> **Table D.** Physics-informed nonlinear pendulum modeling.
> |  | w/o stability loss | w/ stability loss |
> |--|--|--|
> | corr. ↓ | | |
> | Ours | 2.298e-3 | 2.320e-4 |
> | KoVAE | 2.164e-3 | 6.954e-4 |
>
> The results are in **Table D**. Without physics-based loss, ours performs on par with KoVAE. When the physics-based loss is incorporated, we observe a clear improvement, outperforming KoVAE which uses the same stability constraint. This supports incorporating physical knowledge into our model is possible, resulting in a better match with the true data-generating process.
>
> > W3. “A detailed complexity analysis is missing.”
>
> We conducted detailed measurements of wall-clock training and inference times, as well as real and complex parameter counts, for the models in Table 1. The time measurements are done on a single NVIDIA RTX A6000 GPU with 48GB and Intel Xeon Gold 6330 CPU @ 2.00GHz. The results in **Tables E, F, G** show that our models for Table 1 has reasonable computational complexity, and their generation is actually faster than the previous SoTA diffusion model (Diffusion-TS).
>
> **Table E.** Wall-clock training times.
> | | Sines | Stocks | ETTh | MuJoCo | Energy | fMRI |
> |--|--|--|--|--|--|--|
> | Ours | 18min | 15min | 27min | 60min | 44min | 35min |
> | Diffusion-TS | 16min | 15min | 30min | 25min | 60min | 48min |
>
> **Table F.** Wall-clock sampling times per 1,000 samples.
> | | Sines | Stocks | ETTh | MuJoCo | Energy | fMRI |
> |--|--|--|--|--|--|--|
> | Ours | 30sec | 30sec | 29sec | 109sec | 95sec | 109sec |
> | Diffusion-TS | 66sec | 66sec | 67sec | 133sec | 202sec | 262sec |
>
> **Table G.** Numbers of parameters. The parameter counts of baselines are from Table 7 of [10].
> | | Sines | Stocks | Energy |
> |--|--|--|--|
> | Ours | 315,136 | 315,200 | 1,088,160 |
> | Diffusion-TS | 232,177 | 291,318 | 1,135,144 |
> | Diffwave | 533,592 | 599,448 | 1,337,752 |
> | Cot-GAN | 40,133 	| 52,675 | 601,539 |
> | TimeGAN | 34,026 | 48,775 | 1,043,179 |
> | TimeVAE | 97,525 	| 104,412 | 677,418 |
>
> ---
>
> [1] Utilizing image transforms and diffusion models for generative modeling of short and long time series, NeurIPS 2024
>
> [2] Generative modeling of regular and irregular time series data via Koopman VAEs, ICLR 2024
>
> [3] SDformer: Similarity-driven discrete transformer for time series generation, NeurIPS 2024
>
> [4] FIDE: Frequency-inflated conditional diffusion model for extreme-aware time series generation, NeurIPS 2024
>
> [5] On the constrained time-series generation problem, NeurIPS 2023
>
> [6] GT-GAN: General purpose time series synthesis with generative adversarial networks, NeurIPS 2022
>
> [7] Monash time series forecasting archive, NeurIPS 2021
>
> [8] Time-delay observables for Koopman: Theory and applications, SIAM 2020
>
> [9] Conditional Sig-Wasserstein GANs for time series generation, arXiv 2020
>
> [10] Diffusion-TS: Interpretable diffusion for general time series generation, ICLR 2024

---

> > ### Comment · Reviewer_j6aF · 2025-08-05
> > **Discussion**
> >
> > I appreciate the authors' response. However, I still have concerns regarding the completeness of the comparative evaluation. In particular, the authors have not included comparisons with two recent and relevant works: SDformer [1] and ImagenTime [2], both published at NeurIPS 2024. These methods demonstrate significantly stronger performance in the generation of time series data, as evidenced by Table A and comparison to the results in the cited papers. As such, the claim "We demonstrate state-of-the-art results on time-series modeling" appears overstated in the absence of comparisons to these baselines.
> >
> > Additionally, I note that the authors claim the code for SDformer is unavailable, although it has in fact been released publicly. This further undermines the justification for excluding it from the experimental comparison.
> >
> > I am also concerned by the extent of architectural changes and experimental re-runs conducted during the rebuttal phase. Substantial modifications at this stage raise questions about the maturity and stability of the proposed method, and suggest that the approach may not have been fully validated at the time of submission.
> >
> > For these reasons, I will maintain my negative score.
> >
> > References
> >
> > [1] Chen, Zhicheng, et al. "SDformer: Similarity-driven discrete transformer for time series generation." NeurIPS 2024
> >
> > [2] Naiman, Ilan, et al. "Utilizing image transforms and diffusion models for generative modeling of short and long time series." NeurIPS 2024

---

> ### Author Response · Authors · 2025-08-09
>
> We thank the reviewer for acknowledging our rebuttal on operator-based benefits (W2), more general time-series (W1), and cost analysis (W3). We address the remaining concerns.
>
> > “The authors claim the code for SDformer is unavailable, although it has been released publicly.”
>
> Thank you for pointing this out. We couldn’t locate the official repository of SDformer [1] earlier because: (1) neither the paper nor its appendix had a link to the code, and (2) searching on Google, even with keywords like “github” or “code”, did not show the repo. After the reviewer’s comment, we directly queried GitHub code search and were able to find the repo. We used it for additional comparison, as we describe next.
>
> > “The authors have not compared with SDformer [1] and ImagenTime [2], both in NeurIPS 2024. They have significantly stronger performance, as evidenced by **Table A** and a comparison to the results in [1, 2]. The claim "We demonstrate SoTA results on time-series modeling" appears overstated.”
>
> First, we clarify that the previous response exactly has a comparison with ImagenTime [2] on FRED-MD and NN5 Daily (**Table B**), where ours outperforms ImagenTime in all six cases.
>
> To further address the concern on SDformer, and ImagenTime on datasets in **Table A**, we ran an additional experiment.
>
> One thing to be cautious about the strong results of SDformer and ImagenTime is their size: respectively, at least $30\times$ and $110\times$ larger than ones in **Table A** including ours (see **Table H**).
> It is then reasonable to accordingly scale our models. We tested a larger version of ours with 35,300,864 params (still smaller than [1, 2]) with no other changes. We used Sines, Stocks, and MuJoCo, which allowed experimenting under limited time and reproducible datasets in codebase of [1, 2].
>
> The results are in **Table I**. Our method mostly outperforms ImagenTime and SDformer-M, and is competitive with SDformer-AR. We note that SDformers use sophisticated two-stage training: VQ codebook then a generative model on it, while ours is simply end-to-end trained with flow matching.
>
> Given the competitive result of ours to SDformer-AR, we will revise our writing to “competitive performances to the state-of-the-art”. We remark that, in addition to solid performance, our work hosts a list of meaningful contributions on construction of a novel model class (Sec. 3) and new theoretical results on mean embeddings and MMD flows (Sec. 3.1, 3.2).
>
> **Table H.** Size of SDformer [1] and ImagenTime [2] compared to the largest in **Table A**.
> ||Number of parameters|
> |---|---:|
> |Diffwave (largest in **Table A**)|1,337,752|
> |SDformer-AR [1]|44,971,036|
> |SDformer-M [1]|44,971,036|
> |ImagenTime [2]|151,717,761|
>
> **Table I.** More comparisons with SDformer [1] and ImagenTime [2]. Error bars omitted due to space limits. The best are **bold** and second-best are in [brackets]. ImagenTime, and SDformer-AR on Stocks/MuJoCo, were reproduced using official code.
> |  |  | Sines | Stocks | MuJoCo |
> |---|---|---|---|---|
> | Disc. ↓ | Ours (large) | **0.003** | **0.0082** | **0.0055** |
> |  | SDformer-AR | [0.006] | [0.0131] | [0.0074] |
> |  | SDformer-M | 0.008 | 0.020 | 0.025 |
> |  | ImagenTime | 0.028 | 0.0161 | 0.0223 |
> | Pred. ↓ | Ours (large) | **0.093** | **0.0367** | [0.0078] |
> |  | SDformer-AR | **0.093** | [0.0368] | 0.0080 |
> |  | SDformer-M | **0.093** | 0.037 | **0.007** |
> |  | ImagenTime | [0.094] | [0.0368] | 0.0323 |
> | Cont.-FID ↓ | Ours (large) | [0.004] | **0.0041** | [0.0121] |
> |  | SDformer-AR | **0.001** | [0.0188] | **0.0049** |
> |  | SDformer-M | 0.010 | 0.034 | 0.030 |
>
> > “I am concerned by the extent of architectural changes and experiments within rebuttal phase.”
>
> We would like to clarify that the architectural improvements made in the rebuttal are rather peripheral implementation changes, such as activation func. (SwiGLU vs. squared ReLU), time-conditioning (adaptive-norm vs. additive), and feature readout (last layer vs. all layers), all of which are standard choices and are mostly simpler than the previous version. Importantly, these improvements do not deviate from our theoretical construction, and are used consistently throughout all the results presented during the rebuttal.
>
> With the originality of our idea and framework already acknowledged by the reviewer as strengths, we believe our responses thoroughly address all of the reviewer’s remaining concerns on operator-based benefits (W2), more general time-series (W1), detailed complexity analysis (W3), and comparisons with important recent baselines (W1, W2 and this response), through the additional experiments.
>
> We thank the reviewer’s constructive comments, which helped us improve the work. We will make sure to incorporate all the updates in our next revision.
>
> ---
> [1] SDformer: Similarity-driven discrete transformer for time series generation, NeurIPS 2024
>
> [2] Utilizing image transforms and diffusion models for generative modeling of short and long time series, NeurIPS 2024

---

### Official Review · Reviewer_EcXj · 2025-07-02

**Clarity:** 3
**Significance:** 3
**Originality:** 3
**Rating:** 5
**Confidence:** 3

**Summary:**

The authors provide a novel method for generating sequences using a technique called spectral mean flows. This uses a neural network parameterization of a spectral decomposition of linear operators to define a generative model over the space of sequences. They demonstrate this on toy data sets and multiple timeseries dat sets showing good distribution match between ground truth and generated data.

**Questions:**

Following from the potential weakness above, can this method incorporate information from an informatively sampled system?

The authors comment that “G is closed under point wise multiplication“ and “in the infinite width limit, multi-layer perceptrons (MLPs) are norm equivalent to Sobolev spaces”. How well do finite width and depth MLPs approximate this space? Does this lead to unique scaling performance with MLP size?

**Ethical Concerns:**

["NO or VERY MINOR ethics concerns only"]

**Final Justification:**

The authors appropriately responded to my comments and I have kept my original rating.

**Limitations:**

The authors recognize a main limitation that methods to generate sequences conditioned on a partial sequence are not provided.

**Quality:**

3

**Strengths And Weaknesses:**

Strengths:
The authors provide adequate background grounding their extensions. This method makes no assumptions on characteristics of the timeseries being modeled (in comparison to others which use Fourier transforms to explicitly parameterize periodicity). The methods are original.
The authors recognize a main limitation that methods to generate conditioned on a partial sequence are not provided.
Weaknesses:
The authors do not comment on whether a timeseries produced through irregular or informative sampling would be handled well by this approach. For example, a system with a measurement generator function M(h) which emits measurements of value x at time t according to the value of some hidden state h such that  both the value x and the time difference between x_t and x_{t+1} are dependent on h.

---

> ### Author Rebuttal · Authors · 2025-07-31
>
> We thank the reviewer for the constructive feedback and positive assessment of our work. We address the main concerns and questions below.
>
> > W1. “The authors do not comment on whether a timeseries produced through irregular or informative sampling would be handled well by this approach.”
>
> Thank you for raising this point. Thanks to the HMM formulation, our method can be applied to time series under irregular or informative sampling using the classical approach [1] of treating the sampling interval itself as a part of the time series data. We explain the reasoning behind this and perform an experimental demonstration. There is another potential direction based on continuous-time operators, which we discuss at the end of this response.
>
> Let us consider HMM hidden states $Z^{1:N}$ under Markovian dynamics $P(Z^{n+1}|Z^n)$, and a local observation model $P(X^n, \Delta \tau^n|Z^n),$ which jointly determines (emits) the measurement of the value of $X^n$ and the sampling time interval $\Delta \tau^n$ between $X^{n-1}$ and $X^n$. The setup mentioned by the reviewer is a special case when $\Delta \tau^n$ has no randomness given $Z^n$. If the hidden Markov chain is expressive enough, this HMM can express any joint distribution $P((X^1, \Delta \tau^1), …, (X^N, \Delta \tau^N))$, i.e., general time series under irregular or informative sampling. To incorporate this into our method, the only change necessary is to use sampling time intervals $\Delta\tau^n$ as an additional data dimension.
>
> To experimentally demonstrate this, we consider three irregular datasets commonly used in literature [2] derived from Google stock price data by randomly omitting 30%, 50%, and 70% of the observations. The task is generative modeling of time series values along with their irregular sampling intervals, $P((X^1, \Delta \tau^1), …, (X^N, \Delta \tau^N))$. Along with our model, we test Koopman VAE [2], a recent SoTA method for operator-theoretic generative modeling of regular and irregular time series, using the recommended hyperparameters for the stock datasets. Based on [2], we use a discriminative metric for evaluation.
>
> **Table A.** Irregular time series modeling (joint modeling of $P((X^1, \Delta \tau^1), …, (X^N, \Delta \tau^N))$, where 30%, 50%, and 70% of the observations are dropped.
> | | Stocks, no drop | 30% | 50% | 70% |
> |--|--|--|--|--|
> | disc. ↓ | | | | |
> | Ours 	| **0.0150 ± 0.0101** 	| **0.0498 ± 0.0776** 	| **0.1453 ± 0.1390** 	| **0.0350 ± 0.0147** 	|
> | KoVAE 	| 0.0203 ± 0.0199 	| 0.3070 ± 0.0587 	| 0.2030 ± 0.0750 	| 0.1892 ± 0.0631 	|
>
> The results are in **Table A**. Our method achieves strong performances, achieving the best metric in all 4 cases. The result demonstrates the potential of scaling our method to irregular time series, and also provides a comparison against an additional baseline. We will add the results and discussion to the main text in the next revision.
>
> In addition to the above, another potential direction for handling irregular or informative sampling intervals is using the continuous-time CME operator $U^{\Delta\tau}\phi(z) = \mathbb{E}[\phi(Z^{\tau+\Delta\tau}) | Z^\tau = z]$ and its spectral decomposition $U = \sum_i \exp(s_i \Delta\tau) u_i \otimes v_i$ with continuous-time eigenvalues $s_i$ [3]. We leave further investigation of this direction as future work.
>
> > W2. “The authors recognize a main limitation that methods to generate sequences conditioned on a partial sequence are not provided.”
>
> Thank you for the comment. While we acknowledge the limitation and leave conditioning as future work, following another reviewer k6rF’s request, we discussed preliminary insights on kernel Bayesian inference or Laplace transforms of CME operators (Section 7) and would like to include the response here as well.
>
> Let’s say we have a trajectory $x^{1:n}$ and want to sample $x^{n+1:N} \sim P(X^{n+1:N}|X^{1:n} = x^{1:n})$. This can be done with MMD flow if we have conditional mean embedding $\mu\_{X^{n+1:N}|X^{1:n} = x^{1:n}}$.
>
> In kernel methods for HMMs, this is precisely what kernel Bayesian inference is used for [4, 5]. As an alternative, in deterministic settings, a type of Laplace transform of Koopman operator (a variant of CME operator in our work) was used for kernel-based forecasting $x^{1:n} \mapsto x^{n+1:N}$ of nonlinear dynamics [6]. We conjecture its extension to CME operator may give us conditional mean embedding.
>
> While the directions are exciting, we believe their proper investigation warrant a separate work. Thus, we focused our effort on laying the foundations of a neural method based on operator theory.
>
> > W3. “The authors comment that “G is closed under point wise multiplication“ and “in the infinite width limit, multi-layer perceptrons (MLPs) are norm equivalent to Sobolev spaces”. How well do finite width and depth MLPs approximate this space? Does this lead to unique scaling performance with MLP size?”
>
> Thank you for the comment. The approximation results in [7, 8] provide a theoretical footing for our final method by connecting neural networks and Sobolev spaces. Performance scaling with MLP size is analyzed in [8]. Indeed, [8, Theorem 1,3] shows that approximation of functions in Sobolev space by MLPs scales like $\inf_{f_{NN}\in NN (\theta)}\|f- f_{NN}\| \leq C\theta^{-2s/d}$, where $s$ is the smoothness if the function to be approximated, $d$ is the input dimension, and $\theta$ the number of parameters tied to a fixed, large enough layer width. As such, we (at least theoretically) expect performance to scale with MLP size. Whether this leads to unique scaling law in practice is a more complicated question, since in practice, other factors such as numerical precision and optimizers are involved. We leave the investigation of scaling laws of our model as future work.
>
> ---
>
> [1] Zucchini & MacDonald, Eruptions of the Old Faithful geyser, 2009
>
> [2] Naiman et al., Generative modeling of regular and irregular time series data via Koopman VAEs, ICLR 2024
>
> [3] Aït-Sahalia et al., Operator methods for continuous-time Markov processes, Elsevier 2010
>
> [4] Nishiyama et al., The nonparametric kernel Bayes smoother, AISTATS 2016
>
> [5] Fukumizu et al., Kernel Bayes’ rule: Bayesian inference with positive definite kernels, JMLR 2013
>
> [6] Bevanda et al., Koopman kernel regression, NeurIPS 2023
>
> [7] Bietti & Bach, Deep equals shallow for ReLU networks in kernel regimes, ICLR 2021
>
> [8] Siegel, Optimal approximation rates for deep ReLU neural networks on Sobolev and Besov spaces, JMLR 2023

---

> > ### Comment · Reviewer_EcXj · 2025-08-01
> >
> > Thank you for the thoughtful comments in response to my questions. My concerns are addressed and I will keep my original rating.

---

> > > ### Author Response · Authors · 2025-08-01
> > >
> > > Thank you very much for your warm support! We will make sure to incorporate all of the discussions in our next revision.

---

### Official Review · Reviewer_Mu8H · 2025-07-03

**Clarity:** 2
**Significance:** 2
**Originality:** 3
**Rating:** 5
**Confidence:** 2

**Summary:**

The paper utilizes an operator-theoretic interpretation of hidden Markov models to propose a novel sequence modeling algorithm. The proposed method, spectral mean flows, utilizes a neural architecture that leverages the spectral decomposition of linear operators for scalability and extends the maximum mean discrepancy (MMD) gradient flows to time-dependent Hilbert spaces. The authors validate the method on both simulated and real-world time-series datasets.

**Questions:**

Could you provide more explanation and intuition on the right panel of Figure 2? More specifically, why does the curve look as it does (initially it goes up)? Moreover, why does it take much longer for the MMD of spectral mean flow to decrease as much as TD-RBF?

Could you provide visualizations for some of the generated sequences in Section 5.2? I think that could help me understand the results from Table 1 better.

The paper makes claims on faster convergence and scalability of the spectral mean flow. Could you run ablation studies that show how each part of your method contributes to this efficiency?

**Ethical Concerns:**

["NO or VERY MINOR ethics concerns only"]

**Final Justification:**

The authors addressed my concern about the experiments, which was the main justification for my original score. The additional results that the authors shared during the rebuttal are convincing. I thus decide to increase the score to 5.

**Limitations:**

Yes.

**Paper Formatting Concerns:**

I did not notice any major formatting issues in this paper.

**Quality:**

3

**Strengths And Weaknesses:**

Quality: The submission is technically sound. The background and methods sections are especially technically sound, going into the details. However, I am not fully convinced that the spectral mean flow improves much from other baselines. For example, the predictive scores of spectral mean flow are on par or worse than other baselines across all datasets.

Clarity: The paper is mostly clearly written and well organized. It has clear methods, background, and related work sections. However, I think the figure and table captions can be improved, providing more explanations or intuition, especially for Figure 1.

Significance: Although the idea is novel, and the technical details in the paper are sound, the paper lacks experimental results to back up the proposed method. I think a more careful demonstration of the advantage of using spectral mean flow over other baseline methods will improve the significance of the paper.

Originality: The work provides new insight through an operator-theoretic view of a hidden Markov model.

---

> ### Author Rebuttal · Authors · 2025-07-31
>
> We thank the reviewer for the constructive feedback. We address the main concerns and questions below.
>
> > W1. “I am not fully convinced that the spectral mean flow improves much from other baselines. For example, the predictive scores are on par or worse than other baselines across all datasets.”
>
> We would like to point out that predictive score has saturated in many datasets in Table 1 almost to best possible value, e.g., in Sines and Energy, evaluating real data gives predictive scores 0.094 and 0.250, close to scores of our method (0.093 and 0.251). We believe it is more reasonable to assess all four metrics holistically.
>
> In addition, we did a careful revision of our model (Section 3.4) and were able to improve on high-dimensional Energy and fMRI. The key changes included dropping SwiGLU and adaLN in favor of simpler squared ReLU [1] and sinusoidal embedding [2].
> We are happy to elaborate on the details during the discussion period.
>
> Due to space limits, the updated Table 1 can be found in **Table A** for reviewer k6rf.
> Our renovated model achieves the best metric in 16 out of 24 cases.
>
> > W2. “The paper lacks experimental results to back up the proposed method. A more careful demonstration of the advantage over other baselines will improve the significance.”
>
> To address the concern, we ran additional experiments on two challenging real-world setups: (1) long time series and (2) irregularly sampled time series, comparing with recent SoTA ImagenTime [3] and KoVAE [4].
>
> For long time series, we used FRED-MD and NN5 Daily with 728- and 791-length data [5]. Their modest sizes allowed testing within the rebuttal period. Our models are designed similarly to **Table A**, equipped with time-delay observables used in operator methods [6]. We use marginal, classification, and prediction metrics [3].
>
> **Table B.** Long time series. Baselines are from [3], except for ImagenTime which was reproduced using official code.
> | | FRED-MD | | | NN5 Daily | | |
> |--|--|--|--|--|--|--|
> | | marg. ↓ | class. ↑ | pred. ↓ | marg. ↓ | class. ↑ | pred. ↓ |
> | Ours | **0.0145** | **1.547** | 0.0261 | **0.00453** | **1.07** | 0.427 |
> | ImagenTime | 0.0292 | 1.154 | 0.0279 | 0.00745 | 0.935 | 0.496 |
> | LS4 | 0.022 | 0.544 | 0.037 | 0.007 | 0.636 | **0.241** |
> | SaShiMi | 0.048 | 0.001 | 0.232 | 0.020 	| 0.045 | 0.849 |
> | SDEGAN | 0.084 | 0.501 | 0.677 | 0.085 | 0.085 | 1.01 |
> | TimeGAN | 0.081 | 0.029 | 0.058 | 0.040 | 0.001 | 1.34 |
> | Latent ODE | 0.042 | 0.327 | **0.013** | 0.107 | 0.000 | 1.04 |
> | ODE2VAE | 0.122 | 0.028 | 0.567 	| 0.211 | 0.001 | 1.19 |
> | GP-VAE | 0.152 | 0.016 	| 2.05 | 0.117 | 0.002 | 1.17 |
> | RNN-VAE | 0.132 | 0.036 | 1.47 | 0.137 | 0.000 | 0.967 |
>
> The results are in **Table B**. Our method shows strong results, achieving the best metric in 4 out of 6 cases and outperforming the previous best (ImagenTime, LS4) in 5 out of 6 cases.
>
> For irregular time series, we used 3 datasets derived from Google stock data by omitting 30%, 50%, and 70% of observations [4]. The task is generating full signal. Our models are designed similarly to **Table A**. To handle missing values at training, a simple idea of interpolating observed values worked well. Based on the code of [4], we use discriminative metric.
>
> **Table C.** Irregular time series. 30%, 50%, and 70% dropped. Baselines are from [4] except for KoVAE which was reproduced using official code.
> | | Stocks, no drop | 30% | 50% | 70% | | |
> |--|--|--|--|--|--|--|
> | disc. ↓ | | | | |
> | Ours | **0.0150** | **0.1109**| **0.0130**| **0.0628**|
> | KoVAE | 0.0203 | 0.1336 | 0.0899| 0.0740|
> | GT-GAN | 0.077| 0.251 | 0.265 | 0.230 |
> | TimeGAN | 0.102 | 0.411 | 0.477 | 0.485 |
> | RCGAN | 0.196 | 0.436 	| 0.478 | 0.381 |
> | C-RNN-GAN | 0.399 | 0.500 	| 0.500 | 0.500 |
> | T-Forcing | 0.226 | 0.305 | 0.308 | 0.365 |
> | P-Forcing | 0.257 | 0.341 | 0.388 | 0.317 |
> | GP-VAE | 0.152 | 0.016 	| 2.05 | 0.117 |
> | RNN-VAE | 0.132 | 0.036 | 1.47 | 0.137 |
>
> The results are in **Table C**. Our method achieves the best metric in all 4 cases.
>
> > W3. “I think the figure and table captions can be improved, providing more explanations or intuition, especially for Figure 1.”
>
> Thank you for the constructive feedback. We will improve the caption for Figure 1 by writing:
>
> **Figure 1.** A tensor network diagram of the inner product $\langle \phi(x^1) \otimes\cdots\otimes \phi(x^N), \mu\_\rho \rangle$ (Eq. 3.12) for $N = 5$, showing how its computation involving a higher-order tensor $\mu\_\rho$ decomposes into a series of matrix and vector multiplications.
>
> We will also add a compact guide in the Appendix on how to read tensor diagrams, and leave pointers to the related material [7, 8].
>
> > W4. “Could you provide more explanation and intuition on the right panel of Figure 2? Why does the curve look as it does (initially it goes up)? Why does it take much longer for the MMD of spectral mean flow to decrease as much as TD-RBF?”
>
> Thank you for raising this point. There are two reasons for this.
>
> One reason comes from our notion of “convergence” of the time-dependent distribution $(p_t)_{t\geq 0}$. We have a target distribution $\rho$, and, when we say “fast convergence” in our work, it means $p_t$ converges arbitrarily close to $\rho$ at a fixed time point $t = 1$ (while naïve MMD flow requires $t \to\infty$). This notion of convergence is suitable for generative modeling, where we would like to generate samples in a bounded time.
>
> As long as this boundary condition is satisfied, we have valid convergence at $t = 1$ and hence do not worry about the behavior, e.g., around $t = 0$. This convergence at $t = 1$ is what the optimal transport (OT) vector field (Eq. 3.16) guarantees, and it is precisely reflected in the right panel of Figure 2, where the MMD reaches $\approx 0$ at $t = 1$ while TD-RBF saturates around $0.1$.
>
> There might be other choices of vector fields that guarantee convergence at $t = 1$ while offering more rapid convergence at early timesteps, but in this work, we used the OT vector field since it was the simplest and easiest to use. This choice has the possibility for improvement in future work. We will make this point clear in the next revision.
>
> In addition, the technical reason TD-RBF appears to decrease rapidly in early timesteps is as follows. In the right panel of Figure 2, the y-axis MMD is measured using an RBF kernel with bandwidth $\sigma = 1.0.$ By definition, TD-RBF at each time-point t tries to minimize the MMD measured with an RBF kernel with bandwidth $\sigma(t)$. Since we defined $\sigma(t) = (1 - t) + 0.1t$, it starts with a bandwidth $\sigma(0) = 1.0$, which gradually decreases over time. This is one reason for its rapid decrease near $t = 0$: it explicitly decreases the y-axis MMD metric of Figure 2.
>
> Yet, despite its early behavior, we remind that TD-RBF stagnates around $t = 1$. This is because it does not have the convergence guarantee at $t = 1$ (unlike Ours), which we consider more important in the context of generative modeling.
>
> > W5. “Could you provide visualizations for some of the generated sequences in Section 5.2?”
>
> Thank you for the suggestion. While we can confirm that the generated sequences appear visually realistic, this year’s NeurIPS reviewing and discussion policy unfortunately prevents us from including any figures in the rebuttal [9]. We will include a side-by-side comparison of real and generated sequences for all six datasets in Table 1 in the next revision.
>
> > W6. “The paper makes claims on faster convergence and scalability of the spectral mean flow. Could you run ablation studies that show how each part of your method contributes to this efficiency?”
>
> Thank you for the constructive comment. We address each aspect below.
>
> Our claim on faster convergence is in comparison to the standard MMD gradient flow (Lines 138-139), and its naïve extension to time-dependent RKHS (Lines 148-149). Thus, the experiment in Section 5.1 and Figure 2 are precisely the ablation study testing improvements compared to these two baselines.
>
> Our claim on the scalability is in comparison to when the tensor network decomposition (Figure 1, Eq. 3.11) is not available. To illustrate how this idea is contributing to scalability under the hood, we ran a simple numerical experiment for a simplified version of Eq. 3.13: evaluating the inner product $\langle x^1 \otimes\cdots\otimes x^N, T\rangle$ between a rank-1 tensor $x^1 \otimes\cdots\otimes x^N$ of vectors $x^n\in\mathbb{R}^d$ and a higher-order tensor $T\in\mathbb{R}^{d^N}$, for $d=32$.
>
> **Table D.** Peak GPU memory usage (GB) of a simple tensor inner product operation $\langle x^1 \otimes\cdots\otimes x^N, T\rangle$ depending on the availability of tensor network decomposition of $T$ (Eq. 3.11). Measured on a single NVIDIA RTX A6000 GPU with 48GB.
> | Sequence length $N$ | w/ tensor network | w/o tensor network |
> |--|--|--|
> | 3 	| 0.00795 	| 0.00990 	|
> | 4 	| 0.00797 	| 0.13297 	|
> | 8 	| 0.00803 	| OOM 	|
> | 16 	| 0.00815 	| OOM 	|
> | 32 	| 0.00840 	| OOM 	|
> | 64 	| 0.00890 	| OOM 	|
> | 128 	| 0.00989 	| OOM 	|
> | 256 	| 0.01187 	| OOM 	|
> | 512 	| 0.01584 	| OOM 	|
> | 1024 	| 0.02381 	| OOM 	|
>
> The results are in **Table D**, indeed showing that without the tensor network decomposition, it is almost impossible to evaluate the inner product.
>
> ---
>
> [1] Primer: Searching for efficient transformers for language modeling, NeurIPS 2021
>
> [2] Attention is all you need, NeurIPS 2017
>
> [3] Utilizing image transforms and diffusion models for generative modeling of short and long time series, NeurIPS 2024
>
> [4] Generative modeling of regular and irregular time series data via Koopman VAEs, ICLR 2024
>
> [5] Monash time series forecasting archive, NeurIPS 2021
>
> [6] Time-delay observables for Koopman: Theory and applications, SIAM 2020
>
> [7] The tensor cookbook, 2024 https://tensorcookbook.com/
>
> [8] An introduction to graphical tensor notation for mechanistic interpretability, arXiv 2024
>
> [9] https://neurips.cc/Conferences/2025/PaperInformation/NeurIPS-FAQ

---

### Official Review · Reviewer_k6rF · 2025-07-03

**Clarity:** 2
**Significance:** 2
**Originality:** 2
**Rating:** 2
**Confidence:** 4

**Summary:**

This paper introduces a sequence modeling algorithm grounded in an operator-theoretic interpretation of Hidden Markov Models. Rather than directly implementing stochastic recurrence, the authors embed distributions into Hilbert spaces. The model further incorporates a time-dependent MMD gradient flow for sampling and employs a flow-matching learning algorithm trained via double backpropagation.

**Questions:**

Q1) Do the authors have any preliminary insights on how kernel Bayesian inference or Laplace transforms of conditional mean operators could be practically incorporated into their framework for conditioning on partial data?

Q2) Have the authors explored methods to recover or approximate interpretable components from the end-to-end model?

**Ethical Concerns:**

["NO or VERY MINOR ethics concerns only"]

**Final Justification:**

While some points were clarified, my concerns about the completeness of the evaluation remain. Therefore, I will be keeping my current score.

**Limitations:**

yes

**Quality:**

2

**Strengths And Weaknesses:**

Strengths: The paper is clearly written and easy to follow. The motivation behind the work is well-articulated, and the related literature is discussed. The proposed method is explained in detail, with clear examples of its integration into state-of-the-art graph learning models.

Weaknesses:
1) The paper lacks discussion and experimental analysis on the robustness of the proposed method with respect to real word datasets. Also the results in Table 1 is very marginal.
2) The framework currently lacks a principled approach for conditioning the generative process on partial observations.
3) The learned model makes it difficult to extract identifiable components like the observation operator or left eigenfunctions, which limits its interpretability.

---

> ### Author Rebuttal · Authors · 2025-07-31
>
> We thank the reviewer for the thoughtful feedback. We address the main concerns below.
>
> >  W1. “The results in Table 1 are very marginal.”
>
> We would like to remind that our gain in Table 1 is often significant, e.g., Ours improves previous best context-FID, corr., and disc. scores in ETTh by ≈50% (Lines 293-295). We believe this is notable as other methods often use specialized Fourier losses [1] while we only use standard flow matching loss.
>
> In addition, we did a careful revision of our model (Section 3.4) and were able to improve on high-dimensional Energy and fMRI. The key changes included dropping SwiGLU and adaLN in favor of simpler squared ReLU [2] and sinusoidal embedding [3].
> We are happy to elaborate on the details during the discussion period.
>
> **Table A.** Updated Table 1. Error bars omitted due to space limits. The best are **bold** and second-best are in [brackets].
> | Metric | Methods | Sines | Stocks | ETTh | MuJoCo | Energy | fMRI |
> |--|--|--|--|--|--|--|--|
> | Cont.-FID ↓ | Ours | **0.006** | **0.020** | **0.051** | [0.024] | **0.058** | [0.115] |
> | | Diffusion-TS | [0.013] | 0.209 | [0.133] | **0.014** | [0.085] | **0.106** |
> | | DiffTime | **0.006** | 0.236 | 0.299 | 0.188 | 0.279 | 0.340 |
> | | Diffwave | 0.014 | 0.232 | 0.873 | 0.393 | 1.031 | 0.244 |
> | | TimeGAN | 0.101 | [0.103] | 0.300 | 0.563 | 0.767 | 1.292 |
> | | TimeVAE | 0.307 | 0.215 | 0.805 | 0.251 | 1.631 | 14.449 |
> | | Cot-GAN | 1.337 | 0.408 | 0.980 | 1.094 | 1.039 | 7.813 |
> | Corr. ↓ | Ours | 0.025 | **0.002** | **0.032** | [0.214] | **0.746** | **0.750** |
> | | Diffusion-TS | 0.032| 0.009| [0.049] | **0.191** | [0.809] | [1.411] |
> | | DiffTime | **0.017** | [0.006] | 0.067 | 0.218 | 1.158| 1.501|
> | | Diffwave | [0.022] | 0.030| 0.175| 0.579| 5.001| 3.927|
> | | TimeGAN | 0.045 | 0.063 | 0.210 | 0.886 | 4.010 | 23.502 |
> | | TimeVAE | 0.131 | 0.095 | 0.111 | 0.388 | 1.688 | 17.296 |
> | | Cot-GAN | 0.049 | 0.087 | 0.249 | 1.042 | 3.164 | 26.824 |
> | Disc. ↓ | Ours | [0.011] | **0.015** | **0.014** | **0.010** | **0.120** | **0.108** |
> | | Diffusion-TS | **0.007** | [0.080] | [0.073] | [0.014] | [0.122] | [0.175] |
> | | DiffTime | 0.013 | 0.097 | 0.100 | 0.154 | 0.445 | 0.245 |
> | | Diffwave | 0.017 | 0.232 | 0.190 | 0.203 | 0.493 | 0.402 |
> | | TimeGAN | [0.011] | 0.102 | 0.114 | 0.238 | 0.236 | 0.484 |
> | | TimeVAE | 0.041 | 0.145 | 0.209 | 0.230 | 0.499 | 0.476 |
> | | Cot-GAN | 0.254 | 0.230 | 0.325 | 0.426 | 0.498 | 0.492 |
> | | AR-RNN | 0.495 | 0.226 | - | - | 0.483 | - |
> | Pred. ↓ | Ours | **0.093** | **0.037** | [0.124] | [0.010] | [0.251] | **0.100** |
> | | Diffusion-TS | **0.093** | **0.037** | [0.124] | **0.008** | **0.250** | **0.100** |
> | | DiffTime | **0.093** | [0.038] | **0.121** | [0.010] | 0.252 | **0.100** |
> | | Diffwave | **0.093** | 0.047 | 0.130 | 0.013 | [0.251] | [0.101] |
> | | TimeGAN | **0.093** | [0.038] | [0.124] | 0.025 | 0.273 | 0.126 |
> | | TimeVAE | **0.093** | 0.039 | 0.126 | 0.012 | 0.292 | 0.113 |
> | | Cot-GAN | [0.100] | 0.047 | 0.129 | 0.068 | 0.259 | 0.185 |
> | | AR-RNN | 0.150 | [0.038] | - | - | 0.315 | - |
> | | Original | 0.094 | 0.036 | 0.121 | 0.007 | 0.250 | 0.090 |
>
> The updated Table 1 is in **Table A**. Our renovated model achieves the best metric in 16 out of 24 cases.
>
> > W2. “The paper lacks discussion and experiments on real-world datasets.”
>
> We would like to clarify that 5 out of 6 datasets in Table 1 are real-world or realistic simulations for real-world applications (robotics, medical).
> Nevertheless, we ran additional experiments on two challenging real-world setups: (1) long time series and (2) irregularly sampled time series, comparing with recent SoTA ImagenTime [4] and KoVAE [5].
>
> For long time series, we used FRED-MD and NN5 Daily with 728- and 791-length data [6]. Their modest sizes allowed testing within the rebuttal period. Our models are designed similarly to **Table A**, equipped with time-delay observables used in operator methods [7]. We use marginal, classification, and prediction metrics [4].
>
> **Table B.** Long time series. Baselines are from [4], except for ImagenTime which was reproduced using official code.
> | | FRED-MD | | | NN5 Daily | | |
> |--|--|--|--|--|--|--|
> | | marg. ↓ | class. ↑ | pred. ↓ | marg. ↓ | class. ↑ | pred. ↓ |
> | Ours | **0.0145** | **1.547** | 0.0261 | **0.00453** | **1.07** | 0.427 |
> | ImagenTime | 0.0292 | 1.154 | 0.0279 | 0.00745 | 0.935 | 0.496 |
> | LS4 | 0.022 | 0.544 | 0.037 | 0.007 | 0.636 | **0.241** |
> | SaShiMi | 0.048 | 0.001 | 0.232 | 0.020 	| 0.045 | 0.849 |
> | SDEGAN | 0.084 | 0.501 | 0.677 | 0.085 | 0.085 | 1.01 |
> | TimeGAN | 0.081 | 0.029 | 0.058 | 0.040 | 0.001 | 1.34 |
> | Latent ODE | 0.042 | 0.327 | **0.013** | 0.107 | 0.000 | 1.04 |
> | ODE2VAE | 0.122 | 0.028 | 0.567 	| 0.211 | 0.001 | 1.19 |
> | GP-VAE | 0.152 | 0.016 	| 2.05 | 0.117 | 0.002 | 1.17 |
> | RNN-VAE | 0.132 | 0.036 | 1.47 | 0.137 | 0.000 | 0.967 |
>
> The results are in **Table B**. Our method shows strong results, achieving the best metric in 4 out of 6 cases and outperforming the previous best (ImagenTime, LS4) in 5 out of 6 cases.
>
> For irregular time series, we used 3 datasets derived from Google stock data by omitting 30%, 50%, and 70% of observations [5]. The task is generating full signal. Our models are designed similarly to **Table A**. To handle missing values at training, a simple idea of interpolating observed values worked well. Based on the code of [5], we use discriminative metric.
>
> **Table C.** Irregular time series. 30%, 50%, and 70% dropped. Baselines are from [5] except for KoVAE which was reproduced using official code.
> | | Stocks, no drop | 30% | 50% | 70% | | |
> |--|--|--|--|--|--|--|
> | disc. ↓ | | | | |
> | Ours | **0.0150** | **0.1109**| **0.0130**| **0.0628**|
> | KoVAE | 0.0203 | 0.1336 | 0.0899| 0.0740|
> | GT-GAN | 0.077| 0.251 | 0.265 | 0.230 |
> | TimeGAN | 0.102 | 0.411 | 0.477 | 0.485 |
> | RCGAN | 0.196 | 0.436 	| 0.478 | 0.381 |
> | C-RNN-GAN | 0.399 | 0.500 	| 0.500 | 0.500 |
> | T-Forcing | 0.226 | 0.305 | 0.308 | 0.365 |
> | P-Forcing | 0.257 | 0.341 | 0.388 | 0.317 |
> | GP-VAE | 0.152 | 0.016 	| 2.05 | 0.117 |
> | RNN-VAE | 0.132 | 0.036 | 1.47 | 0.137 |
>
> The results are in **Table C**. Our method achieves the best metric in all 4 cases.
>
> > W3. “It is difficult to extract identifiable components, which limits interpretability.”
>
> While interpretation of eigenfunctions is not direct due to our end-to-end design, we believe that a closely related benefit of operator methods is retained to some degree: incorporation of physics-based prior knowledge. Thanks to linearity of tensor network, this can be done via explicit spectral regularization. We test this using a challenging physics-constrained problem from [5].
>
> We consider a nonlinear pendulum, governed by an ODE of angular displacement $\theta$ from an equilibrium [5]: $\frac{d^2}{dt^2}\theta + 9.8\sin\theta = 0, \dot{\theta}(0)=0$.
> As the pendulum is stable and conservative, the physical knowledge is that eigenvalues of underlying operator have non-positive real part and some have zero real.
>
> Since our model is governed by a linear product of matrix hidden states $Ow(x^n)$ (Eq. 3.12 and Figure 2), we can implement this knowledge in an end-to-end manner.
> In KoVAE [5], this stability constraint is built-in using a loss on matrix $A$ for hidden states $z_{t+1} = Az_t$, specifically $|s_1 - 1|^2 + |s_2 - 1|^2$ where $(s_1, s_2)$ are the two largest eigenvalues of $A$. We compare our model against [5] using the same spectral loss for each matrix hidden state’s two largest eigenvalues.
> We use correlational score for eval. [8]
>
> **Table D.** Physics-informed nonlinear pendulum modeling.
> |  | w/o stability loss | w/ stability loss |
> |--|--|--|
> | corr. ↓ | | |
> | Ours | 2.298e-3 | 2.320e-4 |
> | KoVAE | 2.164e-3 | 6.954e-4 |
>
> The results are in **Table D**. Without physics-based loss, ours performs on par with KoVAE. When the physics-based loss is incorporated, we observe a clear improvement, outperforming KoVAE which uses the same stability constraint. This supports incorporating physical knowledge into our model is possible, resulting in a better match with the true data-generating process.
>
> > W4. “The framework lacks a principled approach for conditioning on partial observations.” “Do you have preliminary insights?”
>
> We do have preliminary insights. Say we have a trajectory $x^{1:n}$ and want to sample $x^{n+1:N} \sim P(X^{n+1:N}|X^{1:n} = x^{1:n})$. This can be done with MMD flow if we have conditional mean embedding $\mu\_{X^{n+1:N}|X^{1:n} = x^{1:n}}$.
>
> In kernel methods for HMMs, this is precisely what kernel Bayesian inference is used for [9, 10]. As an alternative, in deterministic settings, a type of Laplace transform of Koopman operator (a variant of CME operator in our work) was used for kernel-based forecasting $x^{1:n} \mapsto x^{n+1:N}$ of nonlinear dynamics [11]. We conjecture its extension to CME operator may give us conditional mean embedding.
>
> While the directions are exciting, we believe their proper investigation warrant a separate work. Thus, we focused our effort on laying the foundations of a neural method based on operator theory.
>
> ---
>
> [1] Diffusion-TS: Interpretable diffusion for general time series generation, ICLR 2024
>
> [2] Primer: Searching for efficient transformers for language modeling, NeurIPS 2021
>
> [3] Attention is all you need, NeurIPS 2017
>
> [4] Utilizing image transforms and diffusion models for generative modeling of short and long time series, NeurIPS 2024
>
> [5] Generative modeling of regular and irregular time series data via Koopman VAEs, ICLR 2024
>
> [6] Monash time series forecasting archive, NeurIPS 2021
>
> [7] Time-delay observables for Koopman: Theory and applications, SIAM 2020
>
> [8] Conditional Sig-Wasserstein GANs for time series generation, arXiv 2020
>
> [9] The nonparametric kernel Bayes smoother, AISTATS 2016
>
> [10] Kernel Bayes’ rule: Bayesian inference with positive definite kernels, JMLR 2013
>
> [11] Koopman kernel regression, NeurIPS 2023

---

> > ### Comment · Reviewer_k6rF · 2025-08-03
> >
> > Thank you for your response to my questions. As I mentioned, the results reported in Table 1 are marginal. Since you're modifying the model, could you please provide more details about the new results you’re reporting here?

---

> > ### Comment · Reviewer_k6rF · 2025-08-05
> >
> > Thank you to the authors for their responses. While several points were clarified, my concerns about the completeness of the evaluation remain. Therefore, I will be keeping my current score.

---

> ### Author Response · Authors · 2025-08-04
>
> Thank you for the comment and acknowledging our new results. We detail the model updates. Other parts of the experiment protocol for **Table A** are unchanged and closely follow [1].
>
> > We first give a quick, self-contained recap of our model. This applies to both previous and new versions.
>
> To generate sequences ${\bf x}^{1:N} \in \mathbb{R}^{N\times d}$, we consider a vector field $(v_t)_{t\in[0,1]}$ converting noise at $t=0$ to data at $t=1$. In our work, it is defined as the gradient of a scalar-valued neural network: $v_t({\bf x}^{1:N}) = \nabla f({\bf x}^{1:N}; t)$ where $f(\cdot, t): \mathbb{R}^{N\times d}\to \mathbb{R}$, which is trained by flow matching.
>
> The scalar-valued neural network $f(\cdot; t)$ is composed of two tensor networks with mostly shared parameters. For readability, we recap one of them. For input ${\bf x}^{1:N}$, it first computes hidden states ${\bf b}^1\in\mathbb{C}^r$, ${\bf w}^2,...,{\bf w}^{N-1}\in\mathbb{C}^{r\times r}$ and ${\bf u}^N\in\mathbb{C}^r$. This is done by the feature extractor $\mathrm{MLP}(\cdot, t): {\bf x}^n\mapsto {\bf h}^n \in\mathbb{C}^r$ followed by linear readouts ${\bf h}^1\mapsto {\bf b}^1$, ${\bf h}^n\mapsto {\bf w}^n$ and ${\bf h}^N\mapsto {\bf u}^N$ (parameterized by $\lambda, {\bf p,q,L,R}$ in Line 205).
>
> Then, tensor network evaluation simply means we multiply all states ${\bf b}^1{\bf w}^2...{\bf w}^{N-1}{\bf u}^N$, which yields a scalar (imagine a vector-matrix-...-matrix-vector multiplication). This underlies the neural network $f(\cdot, t)$, which takes ${\bf x}^{1:N}$ and outputs a scalar.
>
> > We now discuss updates in the model design.
> We emphasize that all updates are compatible with our theory (operator-theoretic HMM, linearity, tensor networks, MMD flow) and serve to improve practical results. We also remark that final model of **Table A** is still trained only with standard flow matching, not e.g. specialized Fourier loss of previous SoTA [1] (Lines 296-302).
>
> **Activation.** As our model is a gradient field $\nabla f(\cdot; t)$, we need activation functions of $\mathrm{MLP}$ to be differentiable. Previously, we used SwiGLU [2]. We found that simpler squared ReLU: ${\bf x} \mapsto {\rm ReLU}({\bf x})^2$ [3] performs better using less parameters (unlike ReLU, it is differentiable).
>
> **Time conditioning.** Previously, for $t$-conditioning in $\mathrm{MLP}(\cdot, t)$, we used adaptive normalization with per-layer learnable scales and bias [4]. We found that simpler additive sinusoidal embedding [5] at the first layer performs better using less parameters.
>
> **Features.** Previously, features ${\bf h}^n\in\mathbb{C}^r$ were taken only from the last layer of $\mathrm{MLP}(\cdot, t)$. We found that concatenating features from all layers is better. If the $\mathrm{MLP}$ has $L$ layers with feature dimension $h$, this means we have $r=Lh$. For parameter efficiency, we use a block-diagonal structure on $r\times r$ readout heads:
> e.g., ${\bf R} = {\rm diag}({\bf r}\_1, ..., {\bf r}\_L)\in\mathbb{C}^{(Lh)\times(Lh)}$ where each ${\bf r}\_l\in\mathbb{C}^{h\times h}$.
> This only uses $O(Lh^2)$ parameters, and also allows space-efficient, layerwise evaluation of tensor network thanks to linearity. That is, instead of evaluating the tensor network of $(Lh)\times (Lh)$ matrices, we can evaluate $L$ tensor networks of $h\times h$ matrices in parallel and sum their results.
>
> > It may be of question how efficient our final model is. To answer this (and from reviewer j6aF's request) we measured wall-clock training and inference times, as well as real and complex parameter counts, for models in **Table A**.
>
> We used a single NVIDIA RTX A6000 GPU with 48GB and Intel Xeon Gold 6330 CPU @ 2.00GHz.
> The results in **Tables E, F, G** show that our models for **Table A** have reasonable cost, and **their generation is actually faster** than previous SoTA diffusion model (Diffusion-TS [1]).
>
> **Table E.** Wall-clock training times.
>
> ||Sines|Stocks|ETTh|MuJoCo|Energy|fMRI|
> |---|---|---|---|---|---|---|
> |Ours|18min|15min|27min|60min|44min|35min|
> |Diffusion-TS|16min|15min|30min|25min|60min|48min|
>
> **Table F.** Wall-clock sampling times per 1,000 samples.
>
> ||Sines|Stocks|ETTh|MuJoCo|Energy|fMRI|
> |---|---|---|---|---|---|---|
> |Ours|30s|30s|29s|109s|95s|109s|
> |Diffusion-TS|66s|66s|67s|133s|202s|262s|
>
> **Table G.** Numbers of parameters. The parameter counts of baselines are from Table 7 of [1].
>
> ||Sines|Stocks|Energy|
> |---|---|---|---|
> |Ours|315,136|315,200|1,088,160|
> |Diffusion-TS|232,177|291,318|1,135,144|
> |Diffwave|533,592|599,448|1,337,752|
> |Cot-GAN|40,133|52,675|601,539|
> |TimeGAN|34,026|48,775|1,043,179|
> |TimeVAE|97,525|104,412|677,418|
>
> ---
>
> [1] Diffusion-TS: Interpretable diffusion for general time series generation, ICLR 2024
>
> [2] Glu variants improve transformer, arXiv 2020
>
> [3] Primer: Searching for efficient transformers for language modeling, arXiv 2022
>
> [4] Scalable diffusion models with transformers, ICCV 2023
>
> [5] Attention is all you need, NeurIPS 2017

---

### Note · Authors · 2025-08-14

We thank the reviewers for their constructive comments. We developed a principled generative framework for sequences with competitive results, grounded in operator theory and MMD flows. We contributed: an operator‑theoretic HMM view that yields a tractable tensor‑network factorization of the sequence mean embedding, and a time‑dependent MMD flow linked to flow matching for faster sampling.

All reviewers (k6rF/Mu8H/EcXj/j6aF) recognized novelty and technical soundness, clear exposition, and the value of unifying operator theory, tensor nets, MMF flows, and flow matching.

We addressed all major concerns:

**Table 1 (k6rF/Mu8H):** While the gain was significant (≈50% ETTh) only with flow loss, with simple updates (squared ReLU, additive time conditioning, multilayer readout) we further reach best in 16/24 entries.

**General data (k6rF/Mu8H/EcXj/j6aF), recent baselines (j6aF):** Beyond the six datasets we added: **(1)** long series (FRED‑MD  728, NN5‑Daily  791) where we are best in 4/6 metrics and beat ImagenTime/LS4 in 5/6; **(2)** irregular series (Stocks 0-70% missing) where we are best in 4/4 and beat KoVAE; **(3)** size-matched comparison to SDformer-M/-AR/ImagenTime where we beat SDformer-M and ImagenTime and are competitive with SDformer‑AR. We will revise our claim to “competitive with SoTA” and include the baselines.

**Priors/interpretability (k6rF/j6aF):** Physics‑informed spectral constraint is straightforward, thanks to operator-based linearity; on nonlinear pendulum, using stability loss reduces corr. error 2.30e‑3 → 2.32e‑4 and surpasses KoVAE with same constraint.

**Cost/ablation studies (j6aF/Mu8H):** Training time is comparable to Diffusion‑TS; generation is faster (29–109 s vs 67–262 s per 1k samples), and parameter counts are modest (~0.3–1.1 M). Ablation shows tensor-network decomposition is crucial.

**Clarity (Mu8H/EcXj):** We clarified descriptions of Figure 1/2 and Sobolev space approximations with finite MLPs.

**Conditioning (k6rF/EcXj):** We outlined a principled path via kernel Bayes’ rule and Laplace transforms; we view this as follow‑up work.

Our responses have addressed concerns of reviewers Mu8H/EcXj, who raised the score 3 → 5 or maintained to 5. Reviewer j6aF had minor remaining concerns on SDformer/ImagenTime baselines, which we further addressed in **(3)** (also, we already compared with ImagenTime in **(1)**). We again thank the reviewers’ and AC’s efforts, and hope these final remarks assist the final decision.

---

### Decision · Program_Chairs · 2025-09-17

**Decision:**

Accept (poster)

**Comment:**

The paper proposes a sequence modeling framework Spectral Mean Flow (SMF), which combines an operator‑theoretic view of Hidden Markov Models, a neural network implementation of spectral decomposition of the underlying operator, and a time‑dependent MMD gradient flow that is trained with the Flow‑Matching objective. Empirically, the method achieves good performance on several time sequence modeling tasks with reasonable time efficiency. There have been quite some discussions, additional related works, and new experiments added during the discussion period and the authors shall incorporate them into the final version.